# CLUSTERING ON SKEWED COST DISTRIBUTIONS

## ABSTRACT

In this paper, we tackle the problem of $(k, z)$-clustering, a generalization of the well-known $k$-means, $k$-medians and $k$-medoids problems that is known to be APX hard, i.e., impossible to approximate within a multiplicative factor of 1.06 in polynomial time for $n$ and $k$ unless P=NP. Due to the APX-hardness, the fastest $(1 + \varepsilon)$-approximation scheme proposed by Feldman et al. (2007), exhibits a run time with a polynomial dependency on $n$, but an exponential dependency $2^{\tilde{\mathcal{O}}(k/\varepsilon)}$ on $k$. We observe that a $(1 + \varepsilon)$-approximation in truly polynomial time is feasible if the data sets exhibit sufficiently skewed distributions. Indeed in practical scenarios, data sets often exhibit a heavy skewness, leading to the overall clustering cost disproportionately dominated by a few clusters. We propose a novel algorithm that adapts the traditional local search technique to effectively manage $(s, 1 - \varepsilon^{z+1})$-skewed datasets with a run time of $(nk/\varepsilon)^{\mathcal{O}(s+1/\varepsilon)}$ for discrete case and $\tilde{\mathcal{O}}(nk) + (k \log n)^{\tilde{\mathcal{O}}(s+1/\varepsilon)}$ for continuous case. Our method is particularly effective with Zipfian distributions with exponent $p > 1$, where $s = \mathcal{O}\left(\frac{1}{\varepsilon^{(z+1)/(p-1)}}\right)$.

## 1 INTRODUCTION

Clustering is a fundamental procedure widely used to extract structural insights from large datasets by partitioning points into groups such that similar points are grouped together. Classic clustering problems, including $k$-means, $k$-median, and $k$-medoids, have been extensively studied since the 1950s (Steinhaus et al., 1956; MacQueen et al., 1967; Rdusseeun & Kaufman, 1987). These problems are fundamental in various fields, such as bioinformatics, computational geometry, data science, and machine learning, attracting significant attention from both practical and theoretical perspectives.

The quality of a clustering solution is often measured by a cost function with the objective of minimizing that cost. Specifically, the $(k, z)$-clustering problem aims to find $k$ centers that minimize $\sum_{x \in X} \min_{c \in C} \text{dist}(x, c)^z$. In the continuous version of $(k, z)$-clustering, centers are chosen from the entire space, while in the discrete version, the centers are restricted to a specific set. Continuous $(k, z)$-clustering reduces to the well-known $k$-means problem when $z = 2$ and to $k$-median when $z = 1$. The discrete version reduces to $k$-medoids when the centers are restricted to the input data points and $z = 1$.

Numerous algorithms have been developed to tackle $(k, z)$-clustering more efficiently. Feldman et al. (2007) introduced an algorithm that approximates $k$-means with a running time of $2^{\tilde{\mathcal{O}}(k/\varepsilon)} \cdot \text{poly}(n)$, which has potentially prohibitively exponential dependencies in $k$. The core idea of the algorithm involves building a weak core set $S$ for a set of potential centers $T$, both of size $\text{poly}(k)$. A brute-force search of $(S, T)$ yields a $(1 + \varepsilon)$-approximation. This approach converts the continuous $k$-means problem into a discrete one, avoiding the exponential dependency on $n$. However, eliminating the exponential dependency on $k$ is crucial for broader applicability.

Despite advances, the $(k, z)$-clustering problem remains computationally challenging. It has been proven to be APX-hard, meaning it cannot be approximated within a fixed constant factor in polynomial time. Specifically, it cannot be approximated within a factor of 1.06 for the continuous case and 1.17 for the discrete case unless P=NP (Cohen-Addad & Lee, 2022). We discuss a number of additional related works in Appendix A.

Although eliminating the exponential dependency on $k$ for the general $(k, z)$-clustering problem is impossible due to its APX-hardness, there is hope for datasets with particular structures. In real-world applications, the datasets are often skewed, with a few clusters dominating the overall clustering cost. This observation motivates the exploration of whether $(k, z)$-clustering can be approximated within a $1 + \varepsilon$ factor in $\text{poly}(n, k)$ time for heavily skewed datasets. Our work provides a positive answer for datasets following such skewed distributions.

## 1.1 OUR CONTRIBUTIONS

Our contribution is a novel algorithm designed specifically for $(k, z)$-clustering on heavily skewed datasets. Using the intrinsic structure of these datasets, our approach achieves a run time with polynomial dependencies on $n$ and $k$, significantly improving efficiency compared to the previous $(1 + \varepsilon)$-approximation algorithms. We define a data set as being $(s, 1 - \varepsilon)$-skewed if the $s$ highest-cost clusters contribute at least a $1 - \varepsilon$ fraction of the total cost. In addition, we say a data set follows a Zipfian distribution with exponent $p$ if the $i$-th highest-cost cluster has a cost proportional to $\frac{1}{i^p}$. In fact, a Zipfian distribution with exponent $p$ is $(s, 1 - \varepsilon)$-skewed for $s > \gamma \left(\frac{1}{\varepsilon}\right)^{\frac{1}{p-1}}$ for some constant $\gamma$. We say a solution $\mathcal{P}$ is a $(1 + \varepsilon)$-approximation if $\text{Cost}(X, \mathcal{P}) \leq (1 + \varepsilon)\text{Cost}(X, \mathcal{C})$, where $\mathcal{C}$ is the optimal $(k, z)$-clustering solution.

Based on these characterizations, we propose two novel algorithms DISCRETEHEAVYSKEW and CONTINUOUSHEAVYSKEW based on local search to efficiently handle skewed data. Our DISCRETEHEAVYSKEW algorithm returns a $(1+\varepsilon)$-approximation for heavily skewed data in polynomial time for $n$ and $k$.

**Theorem 1.1.** *Let $X$ be a set of $n$ data points, and let $T$ be a set of potential centers such that $|T| = \text{poly}(n)$. There exists a deterministic algorithm that, given any $\varepsilon > 0$, for discrete $(k, z)$-clustering, in $(nk/\varepsilon)^{\mathcal{O}(s+1/\varepsilon)}$ time returns a $(1+\varepsilon)$-approximation $\mathcal{P}$ as long as $X$ is $(s, 1-\varepsilon^{z+1})$-skewed. Furthermore, for $z = 1$, $X$ only needs to be $(s, 1 - \varepsilon)$-skewed.*

Our CONTINUOUSHEAVYSKEW returns a $(1 + \varepsilon)$-approximation for heavily skewed data in even a shorter time.

**Theorem 1.2.** *Let $X$ be a set of $n$ data points. There exists an algorithm that, given any $\varepsilon > 0$, for continuous $(k, z)$-clustering, in $\tilde{\mathcal{O}}(nk) + (k \log n)^{\tilde{\mathcal{O}}(s+1/\varepsilon)}$ time returns a $(1+\varepsilon)$-approximation $\mathcal{P}$ with probability at least $0.97$ as long as $X$ is $(s, 1-\varepsilon^{z+1})$-skewed. Furthermore, for $z = 1$, $X$ only needs to be $(s, 1 - \varepsilon)$-skewed.*

If randomness is expensive, there also exists a deterministic version of CONTINUOUSHEAVYSKEW with $(nk)^{\tilde{\mathcal{O}}(s+1/\varepsilon)}$ running time. For the discussion of running time, we assume dimension $d$ as a constant. For a large $d$, a dimension reduction technique introduced by Makarychev et al. (2019) can be used to achieve a $\tilde{\mathcal{O}}(nk) + (k \log n)^{\tilde{\mathcal{O}}(\varepsilon^{-2}(s+1/\varepsilon))}$ running time.

Our DISCRETEHEAVYSKEW and CONTINUOUSHEAVYSKEW can return a $(1 + \varepsilon)$-approximation within a run-time with polynomial independence on $n$ and $k$, while the previous algorithm by Feldman et al. (2007) only has polynomial independence on $n$, but has exponential independence on $k$. The improvement of our algorithm makes the run time more feasible in the case where the input data are heavily skewed. The dependence $s$ on the exponent shows that the extent of the skewness of the data affects the run-time of our algorithm. A more heavily skewed data will induce a less $s$, which makes the run-time even shorter.

We now provide a high-level intuition behind our algorithms and analysis.

**Heavy skew local search.** We introduce an algorithm called HEAVYSKEWLOCALSEARCH, which guarantees a $(1 + \varepsilon)$-approximation for data sets following a heavily skewed distribution. The local search, originally introduced by Arya et al. (2001), seeks a local optimum where swapping up to $t$ centers no longer improves the result. The intuition behind our algorithm is that clustering costs are dominated by a few clusters. We can use brute-force search to identify the centers of these dominant clusters and employ a local search for the remaining ones. By accurately selecting the centers for the high-cost clusters, which represent more than $\left(1 - \varepsilon^{z+1}\right)$ fraction of the total cost, and using local search to achieve a constant approximation for the rest, we achieve an overall $(1 + \mathcal{O}(\varepsilon))$-approximation. At first glance, it appears the only remaining step is to directly apply a local search

to find centers with low costs: by the skewed distribution, we would get a $(1 + \mathcal{O}(\varepsilon))$-approximation for the total cost as long as we could get $\mathcal{O}(1)$-approximation for the centers with low costs.

Unfortunately, the above idea does *not* work directly, and we need more technical ideas to address the issues. In particular, although the local search returns a constant approximation for the *entire* dataset, the solution for low-cost clusters may *not* be a constant approximation. This is because we fix the location of the more expensive centers, which may adversely affect the accuracy of the local search. The returned centers for the low-cost clusters will have an extra additive error due to the influence of expensive centers. To tackle this issue, we take advantage of the multi-swap idea in the Arya et al. (2001), and we show that if we swap a sufficiently large number of centers simultaneously, the additive error is small enough to ensure that the total cost is an $(1+\varepsilon)$-approximation. Of course, we could not swap too many centers at the same time since otherwise, the running time even for a single iteration will break the limit. Fortunately, we find that the swap of $\mathcal{O}(1/\varepsilon)$ points is sufficient for $(1 + \varepsilon)$-approximation, and the efficiency for a single iteration is at least preserved.

**Fast local search.** While HEAVYSKEWLOCALSEARCH guarantees $(1 + \mathcal{O}(\varepsilon))$-approximation and single-iteration efficiency, it does not immediately imply convergence in polynomial rounds. A natural approach would be to swap centers only if the improvement exceeds $1+\varepsilon$. This strategy ensures a polynomial run time, but may overlook smaller improvements. Although individual small improvements may not alter the $(1+\varepsilon)$-approximation, a series of such small gains can accumulate, leading to deviations from the desired approximation. For example, if we ignore $(1+\varepsilon/2)$ improvements for successive $m$ swaps, the cumulative improvement could be $(1 + \varepsilon/2)^m$ factor better than our result, which means our result deviates significantly from the optimal when $m$ is very large. Fortunately, if we open the black-box of the local search, we could show that the number of accumulation is at most $\mathcal{O}(k^2)$. As such, we could rescale the parameter, so the accumulated error can still be controlled in the rate $1 + \varepsilon$. This strategy balances large and small improvements, ensuring both accuracy and efficiency.

**Construction for potential center set.** For continuous $(k, z)$-clustering, we propose an algorithm to construct a potential center set, transforming the continuous $(k, z)$-clustering problem into a discrete one. This approach restricts potential centers to a finite range, making the search computationally feasible. Feldman et al. (2007) used similar strategy to build their PTAS. However, their construction is based on the geometric property of $k$-means, where the center of each cluster is its centroid, a property that does not hold for $z \neq 2$ in general $(k, z)$-clustering. Instead, we used the $\varepsilon$-nets to construct the potential center set, which is suitable for general $z$.

**Construction for coreset.** We can further improve the speed of the algorithm by prepocessing the data into a coreset. Sensitivity sampling can generate a coreset of size $\mathsf{poly}(k)$ in $\tilde{\mathcal{O}}(nk)$ time. Unfortunately, traditional sensitivity sampling merely preserves the cost for the entire set, not individual clusters, potentially losing the skewness of the original data set. To address this, we adapt the sensitivity sampling to maintain skewness. We prove that if we sample $\mathcal{O}(k)$ times number of points in sensitivity sampling, it can preserve the cost for cluster whose cost is larger han $\frac{\varepsilon}{100k}$ fraction of the total cost, which ensures that the coreset accurately reflects the heavily skewed structure of the original dataset. We defer all proofs to the appendix.

**Empirical evaluations.** Although our contribution is primarily theoretical, we performed experiments to demonstrate its performance. We compared the precision of our algorithm with the $k$-means and $k$-mediods algorithms available in the `scikit-learn` and `scikit-learn-extra` library. These algorithms are popular in practice because of their fast execution, but they offer weaker theoretical accuracy guarantees. We chose to compare our algorithm against these fast yet lower-precision methods, rather than other $(1+\varepsilon)$-approximation algorithms, because the latter have exponential run time, making them infeasible for experiments. Our empirical evaluations show that our algorithm outperforms these widely used algorithms in terms of accuracy, serving as a proof-of-concept that complements our theoretical guarantees.

## 2 PRELIMINARIES

Given an integer $n > 0$, let $[n]$ denote the set $\{1, \cdots, n\}$. We use $\mathsf{poly}(n)$ for a fixed polynomial in $n$ and $\mathsf{polylog}(n)$ for $\mathsf{poly}(\log n)$. Since the device stores data points in bits, it is generally acceptable to rescale and assume $X \subset [\Delta]^d$, where $\Delta = \mathsf{poly}(n)$.

In this paper, we focus on Euclidean $(k, z)$-clustering. For vectors $x, y \in \mathbb{R}^d$, let $\text{dist}(x, y)$ denote the Euclidean distance $\|x - y\|_2^2 = \sum_{i=1}^d (x_i - y_i)^2$. For a point $x$ and a set $\mathcal{C}$, $\text{dist}(x, \mathcal{C}) := \min_{c \in \mathcal{C}} \text{dist}(x, c)$. For a weighted point $x$ with weight $w(x)$, $\text{Cost}(x, \mathcal{C}) := w(x) \cdot \text{dist}(x, \mathcal{C})^z$. The total cost is $\text{Cost}(X, \mathcal{C}) = \sum_{i=1}^n \text{Cost}(x_i, \mathcal{C})$. Given a weighted dataset $X = \{(x_i, w(x_i)) : i \in [n]\}$, the goal of continuous Euclidean $(k, z)$-clustering is to find $k$ centers $C = \{c_1, \cdots, c_k\} \subset \mathbb{R}^d$ that minimize the cost function $\text{Cost}(X, C)$. In discrete Euclidean $(k, z)$-clustering, $k$ centers are chosen from a finite set of potential centers $T$ with size $\text{poly}(n)$.

For a center set $\mathcal{C} = \{c_1, \cdots, c_k\}$, let $N(c_i) = \{x \in X : \text{Cost}(x, c_i) \leq \text{Cost}(x, c_j) \text{ for } j \neq i\}$ represent the set of points assigned to center $c_i$. Ties are broken arbitrarily so each $x_i$ belongs to exactly one $N(c_i)$.

**Definition 2.1** $((s, 1 - \varepsilon)$-skewed dataset). *A data set $X$ with optimal $(k, z)$-clustering centers $\mathcal{C} = \{c_1, c_2, \cdots, c_k\}$, ordered by cost such that $\text{Cost}(N(c_i), \mathcal{C}) \geq \text{Cost}(N(c_j), \mathcal{C})$ for $i < j$, is an $(s, 1 - \varepsilon)$-skewed dataset if $\sum_{i=1}^s \text{Cost}(N(c_i), \mathcal{C}) \geq (1 - \varepsilon) \sum_{i=1}^k \text{Cost}(N(c_i), \mathcal{C})$.*

**Definition 2.2** (Zipfian distribution dataset). *A data set $X$ with optimal $(k, z)$-clustering centers $\mathcal{C} = \{c_1, c_2, \cdots, c_k\}$ is a Zipfian distribution data set with exponent $p$ if there exist constants $0 < \gamma_1 < \gamma_2$ and $p > 1$ such that for any $i$, $\gamma_1 \cdot \frac{1}{i^p} \leq \text{Cost}(N(c_i), \mathcal{C}) \leq \gamma_2 \cdot \frac{1}{i^p}$.*

As a highly skewed dataset, a Zipfian distribution dataset is in fact $(s, 1 - \varepsilon)$-skewed for $s = \mathcal{O}((\frac{1}{\varepsilon})^{1/(p-1)})$.

**Lemma 2.3.** *Let $X = \{x_1, x_2, \ldots, x_n\} \subseteq [\Delta]^d$ be a Zipfian distribution dataset. There exists a constant $\gamma > 0$ such that for $s > \gamma \left(\frac{1}{\varepsilon}\right)^{\frac{1}{p-1}}$, $X$ is $(s, 1 - \varepsilon)$-skewed.*

Additionally, we introduce the concepts of $\varepsilon$-coreset and $\varepsilon$-net, which are often used to sample points and generate potential center sets to speed up clustering.

**Definition 2.4** ($\varepsilon$-coreset). *A weighted set $S$ is an $\varepsilon$-coreset of $X$ if, for any set of centers $\mathcal{C} \subset \mathbb{R}^d$ that $|\mathcal{C}| \leq k$, $(1 - \varepsilon)\text{Cost}(X, \mathcal{C}) \leq \text{Cost}(S, \mathcal{C}) \leq (1 + \varepsilon)\text{Cost}(X, \mathcal{C})$.*

**Definition 2.5** ($\varepsilon$-net). *Let $\mathcal{A} \subset \mathbb{R}^d$ be a region. $\mathcal{N}$ is an $\varepsilon$-net of $\mathcal{A}$ if for any $x \in \mathcal{A}$, there exists $y \in \mathcal{N}$ such that $\text{dist}(x, y) \leq \varepsilon$.*

# 3 CONSTRUCTION FOR CORESET AND POTENTIAL CENTER SET

In this section, we describe first describe our coreset construction, which is slightly non-standard, due to the fact that we would like the optimal clustering on the coreset to preserve the skewed distribution of costs. Note that by comparison, the general guarantees of coresets simply require that all clustering costs are preserved up to a $(1 + \varepsilon)$-factor, rather than the costs of all clusters being preserved.

## 3.1 CORESET CONSTRUCTION MAINTAINING SKEWNESS

We adapt the sensitivity sampling framework to construct a coreset that maintains the skewness of the original dataset. The sensitivity sampling framework assigns a value to each point, called sensitivity, which intuitively quantifies the "importance" of that point. Each point is then sampled with a probability proportional to its sensitivity.

First, we introduce the definition of sensitivity.

**Definition 3.1** (Sensitivity). *For $x \in X$, its sensitivity is defined as $s(x) = \sup_{\mathcal{C} \subset \mathbb{R}^d, |\mathcal{C}| \leq k} \frac{\text{Cost}(x, \mathcal{C})}{\text{Cost}(X, \mathcal{C})}$.*

We present an algorithm CORESETCONSTRUCTION that produces a weight set $S$ which is an $\varepsilon$-coreset of $X$. Furthermore, if $X$ is an $(s, 1 - \varepsilon)$-skewed dataset, then $S$ will also be an $(s, 1 - 3\varepsilon)$-skewed dataset.

**Lemma 3.2.** *Let $X$ be an $(s, 1 - \varepsilon)$-skewed dataset. There exists a constant $\gamma > 1$, such that for any $\varepsilon \in (0, \frac{1}{4}]$, CORESETCONSTRUCTION returns an $\varepsilon$-coreset $S$ for $X$ with probability at least $0.97$. Furthermore, $S$ is $(s, 1 - 3\varepsilon)$-skewed, and has a size of $\mathcal{O}(\frac{dk^2}{\varepsilon^3} \log(n\Delta))$.*

---

**Algorithm 1** CORESETCONSTRUCTION$(X, \varepsilon, n, k, \Delta)$

---

**Require:** Dataset $X$, precision parameter $\varepsilon$, size $n$, number of cluster $k$, range $\Delta$
**Ensure:** A weighted set $S$
1: $\gamma \leftarrow$ some large enough constant, $\mu \leftarrow \frac{\gamma dk}{\varepsilon^3} \log(n\Delta)$
2: $S \leftarrow \emptyset$
3: **for** $x \in X$ **do**
4: $\quad s(x) \leftarrow$ sensitivity of $x$
5: $\quad$ With probability $p_x = \min\{\mu \cdot s(x), 1\}$, $w(x) \leftarrow \frac{1}{p_x}$, $S \leftarrow S \cup \{(x, w(x))\}$
6: **return** $S$

---

Our algorithm is analogous to the conventional sensitivity sampling method, but employs a larger sampling parameter, $\mu = \mathcal{O}(\frac{dk}{\varepsilon^3} \log(n\Delta))$, in place of $\mu = \mathcal{O}(\frac{dk}{\varepsilon^2} \log(n\Delta))$ as employed in the traditional approach. With the augmented value of $\mu$, the coreset ensures preservation of both the cost of the full set and the cost for clusters whose expense exceeds $\frac{\varepsilon}{100k}$ of the total cost. This modification allows the coresets to preserve the significantly skewed structure present in the original dataset.

### 3.2 POTENTIAL CENTER SET CONSTRUCTION

In this section, we introduce an algorithm that produces a set $T$ of candidate centers for a dataset $S$, ensuring that for any $\mathcal{C} \subset [\Delta]^d$ with $|\mathcal{C}| \le k$, there exists $\mathcal{C}' \in T^k$ such that $\text{Cost}(S, \mathcal{C}') \in (1 \pm \varepsilon)\text{Cost}(S, \mathcal{C})$. The rationale for constructing such a set $T$ relies on the observation that if $\text{dist}(x, c')$ is a $(1 + \mathcal{O}(\varepsilon))$-approximation of $\text{dist}(x, c)$, then $\text{Cost}(x, c')$ will indeed be a $(1 + \varepsilon)$-approximation of $\text{Cost}(x, c)$ due to the generalized triangle inequality. Consequently, we need to ensure the existence of a center $c' \in T$ such that $\text{dist}(x, c')$ is a $(1 + \mathcal{O}(\varepsilon))$-approximation of $\text{dist}(x, c)$. This can be accomplished by constructing an $\mathcal{O}(\varepsilon)$-net for the ball $B(x, 2^i)$, where $B(x, r) = \{y \in \mathbb{R}^d : \text{dist}(x, y) < r\}$. Using this approach, we can approximate any center $c$ for which $\text{dist}(x, c) \in [2^{i-1}, 2^i]$. However, creating such nets for all possible distances would yield an excessive number of centers because $r$ can range from 0 to infinity. Thankfully, the optimal center must fall within the range $[\Delta]^d$ given that $S \subset [\Delta]^d$. Thus, we only need to construct an $\mathcal{O}(\varepsilon)$-net for balls with radii not exceeding $\Delta$. Further, even though $c$ can be exceedingly close to $x$, necessitating an $\mathcal{O}(\varepsilon)$-net for an infinite number of balls, we note that $\text{Cost}(S, \mathcal{C}) \ge \frac{1}{2^z}$ as long as the optimal clustering cost is non-zero. Hence, we can avoid building nets for very small radii. Specifically, we need to construct nets only for $B(x, 2^{i+1})$, where $i \in [\log(\frac{\varepsilon}{kW}) - 2z - 2, \log \Delta]$, with $W$ representing the upper bound of the point weights. This strategy helps maintain the size of $T$ compact.

---

**Algorithm 2** CENTERNET$(S, \varepsilon, \Delta)$

---

**Require:** Dataset $S$, precision parameter $\varepsilon$, range $\Delta$
**Ensure:** A potential center set $T$
1: $T \leftarrow S$, $W \leftarrow$ the maxium weight of $S$, $M_1 \leftarrow \log\left(\frac{\varepsilon}{kW}\right) - 2z - 2$, $M_2 \leftarrow \log \Delta$
2: **for** $i \leftarrow M_1$ to $M_2$ **do**
3: $\quad \mathcal{N}_i \leftarrow \emptyset$, $r \leftarrow 2^{i+1}$
4: $\quad$ **for** $x \in S$ **do**
5: $\quad\quad N_{i,x} \leftarrow$ an $\frac{\varepsilon r}{2^{2z+1}}$-net in $B(x, r)$
6: $\quad\quad \mathcal{N}_i \leftarrow \mathcal{N}_i \cup \mathcal{N}_{i,x}$
7: $\quad T \leftarrow T \cup \mathcal{N}_i$
8: **return** $T$

---

We prove that for any $\mathcal{C} \subset [\Delta]^d$ and $|\mathcal{C}| \le k$, there exists a set $\mathcal{C}' \in T^k$ that provides a $(1 + \varepsilon)$-approximation to $\mathcal{C}$. Furthermore, the set $T$ has a size of $\text{poly}(k, \log n)$ if $|S| = \text{poly}(k)$.

**Lemma 3.3.** *Let $S$ be a weighted set whose maximum weight is at least $1$. For $\varepsilon \in (0, 1]$, the set $T$ returned by* CENTERNET *satisfies: for any $\mathcal{C} \subset [\Delta]^d$ and $|\mathcal{C}| \le k$, there exists $\mathcal{C}' \subset T^k$ such that*

$$(1 - \varepsilon)Cost(S, \mathcal{C}) \le Cost(S, \mathcal{C}') \le (1 + \varepsilon)Cost(S, \mathcal{C}).$$

*Furthermore, $T$ has a size of $|T| = |S| \cdot 2^{\mathcal{O}(d \log \frac{1}{\varepsilon} \log \log(\frac{k\Delta}{\varepsilon}))}$.*

## 4 HEAVY SKEW LOCAL SEARCH ALGORITHM

We introduce an adapted local search algorithm designed for $(k, z)$-clustering, particularly useful for data sets exhibiting significant skewness. For simplicity, within this section, we assume that $\mathcal{C} = \{c_1, c_2, \ldots, c_k\}$ represents the optimal solution within the net $T$. The centers in $\mathcal{C}$ are arranged so that $\text{Cost}(N(c_i)) \geq \text{Cost}(N(c_j))$ for $i \leq j$. We denote $\mathcal{C}_E$ as the subset of $s$ centers corresponding to the $s$ most costly clusters.

### 4.1 HEAVY SKEW LOCAL SEARCH FOR $k$-MEDIAN

For an $(s, 1 - \varepsilon)$-skewed dataset, we can leverage the structure of the dataset to achieve efficient clustering. The intuition is to search for the $s$ most expensive clusters with high precision and then perform a quicker, lower precision search for the remaining $k - s$ cheaper clusters, aiming to achieve a $(1 + \varepsilon)$-approximation. This can be achieved by using a brute-force search for the $s$ most expensive clusters and a local search for the remaining $k - s$ cheaper centers.

In particular, for any given set of $s$ centers, we run a local search to determine the remaining $k - s$ centers. The local search procedure is used to identify a local optimum for the $(k, z)$-clustering problem. When using a local search with a swap parameter $t$, no more than $t$ existing centers are replaced with an equal number of new centers, provided that such a swap reduces the overall cost. The process continues until no further improvements can be achieved by these swaps.

Unlike the classic local search, which can swap any center, we only swap the centers for the remaining $k - s$ ones, keeping the $s$ guessed centers fixed throughout the local search. By brute-forcing all possible locations for the $s$ most expensive centers, we will eventually find the correct guess. For that correct guess, since we fix the locations of the $s$ centers and only swap the remaining $k - s$ centers, the final set returned will be the precise locations of the $s$ most expensive centers and an approximation for the remaining centers, ensuring a $(1 + \varepsilon)$-approximation.

We must consider that the presence of $s$ fixed centers may adversely affect the local search for the remaining $k - s$ centers. However, through a detailed analysis, we can show that with a carefully chosen swap parameter $t = \mathcal{O}(1/\varepsilon)$, we can mitigate such adverse effects and guarantee the $(1 + \varepsilon)$-approximation.

---

**Algorithm 3** HEAVYSKEWLOCALSEARCH$(S, T, \varepsilon, \mathcal{A}, k, s)$

---

**Require:** Dataset $S$, potential center set $T$, precision parameter $\varepsilon$, set $\mathcal{A} \subset T$ with $|\mathcal{A}| = s$, number of clusters $k$, skewness parameter $s$
**Ensure:** A center set $\mathcal{P}$ with $|\mathcal{P}| = k$
 1: $\gamma \leftarrow$ some large enough constant, $t \leftarrow \frac{\gamma}{\varepsilon}$
 2: $\mathcal{B} \leftarrow$ Arbitrary subset of $T$ with $|\mathcal{B}| = k - s$
 3: **while** $\exists \mathcal{B}' \subset T$ such that $|\mathcal{B} - \mathcal{B}'| \leq 2t$ and $\text{Cost}(S, \mathcal{A} \cup \mathcal{B}') < \text{Cost}(S, \mathcal{A} \cup \mathcal{B})$ **do**
 4: $\quad \mathcal{B} \leftarrow \mathcal{B}'$
 5: $\mathcal{P} \leftarrow \mathcal{A} \cup \mathcal{B}$
 6: **return** $\mathcal{P}$

---

We claim that HEAVYSKEWLOCALSEARCH returns a $(1 + \varepsilon)$-approximation if $S$ is $(1, 1 - \varepsilon)$-skewed and we choose the correct input set $\mathcal{A} = \mathcal{C}_E$, which is the centers of the $s$ most high-cost clusters.

**Lemma 4.1.** *Let $S$ be an $(s, 1 - \varepsilon)$-skewed dataset, $T$ be the potential center set, and $\mathcal{A} = \mathcal{C}_E$, which is the set of centers of the $s$ most high-cost clusters in optimal solution. There exists a constant $\gamma > 1$, such that for any $\varepsilon \in (0, \frac{1}{2}]$, HEAVYSKEWLOCALSEARCH returns a $(1 + \varepsilon)$-approximation $\mathcal{P}$ for the $(k, 1)$-clustering for $S$ and $T$.*

### 4.2 HEAVY SKEW LOCAL SEARCH FOR $(k, z)$-CLUSTERING

Our $(1 + \varepsilon)$-approximation guarantee extends to general $(k, z)$-clustering. The framework remains the same, but the cost function for the $(k, z)$-clustering is $\text{dist}(x, c)^z$ instead of $\text{dist}(x, c)$, as in the $k$-median case. This change affects the additivity of the cost function, requiring a more nuanced

analysis of cost distortion. However, with the generalized triangle inequality and carefully chosen parameters, an $(1 + \varepsilon)$-approximation is still achievable for $(s, 1 - \varepsilon^{z+1})$-skewed set $S$.

**Lemma 4.2.** *Let $S$ be an $(s, 1 - \varepsilon^{z+1})$-skewed dataset, $T$ be the potential center set, and $\mathcal{A} = \mathcal{C}_E$, which is the set of centers of the $s$ most high-cost clusters in optimal solution. There exists a constant $\gamma > 1$, such that for any $\varepsilon \in (0, \frac{1}{2}]$, HEAVYSKEWLOCALSEARCH returns a $(1 + \varepsilon)$-approximation $\mathcal{P}$ for the $(k, z)$-clustering for $S$ and $T$.*

Note that while $(s, 1 - \varepsilon)$-skewness is required for $z = 1$, $(s, 1 - \varepsilon^{z+1})$-skewness is needed for general $(k, z)$-clustering. This indicates that a heavier skewness is necessary for general $(k, z)$-clustering to compensate for the loss of additivity.

# 5 PTAS FOR HEAVILY SKEWED DISTRIBUTION SET

Although HEAVYSKEWLOCALSEARCH guarantees a $(1 + \varepsilon)$-approximation, it does not ensure the existence of a PTAS for $(k, z)$-clustering because it cannot guarantee to terminate in polynomial rounds. An intuitive approach might involve only swapping centers if the result improves significantly, such as an improvement in the multiplier $1 + \varepsilon'$, to ensure the polynomial iteration times. However, this method misses smaller improvements, and a series of such small improvements can accumulate, failing to maintain the desired approximation. For example, successive $m$ swaps, each improving by a factor of $1 + \frac{\varepsilon}{2}$, may result in $\left(1 + \frac{\varepsilon}{2}\right)^m$, which deviates significantly from $1 + \varepsilon$.

Through a comprehensive analysis, we demonstrate that by opting for a more precise choice of the parameter, specifically $\varepsilon' = \mathcal{O}(\frac{\varepsilon}{k^2})$, it is possible to ensure a $(1 + \varepsilon)$-approximation within polynomial iteration times.

---

**Algorithm 4** FASTLOCALSEARCH$(S, T, \varepsilon, \mathcal{A}, k, s)$

---

**Require:** Dataset $S$, potential center set $T$, precision parameter $\varepsilon$, set $\mathcal{A} \subset T$ with $|\mathcal{A}| = s$, number of clusters $k$, skewness parameter $s$
**Ensure:** A center set $\mathcal{P}$ with $|\mathcal{P}| = k$
1: $\gamma \leftarrow$ some large enough constant, $t \leftarrow \frac{\gamma}{\varepsilon}$
2: $\mathcal{B} \leftarrow$ Arbitrary subset of $T$ with $|\mathcal{B}| = k - s$
3: $\Gamma \leftarrow$ constant approximation of total cost
4: **while** $\exists \mathcal{B}' \subset T$ such that $|\mathcal{B} - \mathcal{B}'| \le 2t$ and $\text{Cost}(S, \mathcal{A} \cup \mathcal{B}') < (1 - \frac{\varepsilon}{\Gamma k^2})\text{Cost}(S, \mathcal{A} \cup \mathcal{B})$ and **do**
5: $\quad \mathcal{B} \leftarrow \mathcal{B}'$
6: $\mathcal{P} \leftarrow \mathcal{A} \cup \mathcal{B}$
7: **return** $\mathcal{P}$

---

We claim that FASTLOCALSEARCH terminates within polynomial rounds and returns a $(1 + 2\varepsilon)$-approximation for the optimal solution of clustering $(S, T)$.

**Lemma 5.1.** *Let $S$ be a dataset of $n$ points, $T$ be the potential center set, and $\mathcal{A} = \mathcal{C}_E$, which is the set of centers of the $s$ most high-cost clusters in optimal solution. There exists a constant $\gamma > 1$, such that for any $\varepsilon \in (0, \frac{1}{2}]$, FASTLOCALSEARCH terminates within $\mathcal{O}(\frac{k^2}{\varepsilon})$ swaps, and returns a $(1 + 2\varepsilon)$-approximation $\mathcal{P}$, as long as $S$ is $(s, 1 - \varepsilon^{z+1})$-skewed. Furthermore, for $z = 1$, $S$ only needs to be $(s, 1 - \varepsilon)$-skewed.*

Finally, we give DISCRETEHEAVYSKEW and CONTINUOUSHEAVYSKEW as PTASs to approximate $(k, z)$-clustering within a $(1 + \varepsilon)$ approximation. We construct DISCRETEHEAVYSKEW, the algorithm deals with the discrete $(k, z)$-clustering problem first. Assume that we have an input set $X$ and a potential center set $T$ with $|X| = n$ and $|T| = \text{poly}(n)$. We perform a brute-force search over all possible locations of the centers of the $s$ most expensive clusters and apply FASTLOCALSEARCH on each guess. Since there are $|T|^s = \text{poly}(n)$ possible choices for the $s$ centers, we only need to apply FASTLOCALSEARCH polynomial number of times. The run time of a single application of FASTLOCALSEARCH is $\text{poly}(n, k)$ because it terminates in $\text{poly}(n, k)$ swaps. As a result, we can complete all the computations in $\text{poly}(n, k)$ time.

We claim that DISCRETEHEAVYSKEW guarantee a $(1 + \varepsilon)$ approximation for $(k, z)$-clustering on $X$ and $T$ within $\text{poly}(n, k)$ run time if $X$ is $(s, 1 - \varepsilon^{z+1})$-skewed.

---

**Algorithm 5** DISCRETEHEAVYSKEW$(X, T, \varepsilon, k, s)$

---

**Require:** Dataset $S$, center set $T$, precision $\varepsilon$, number of clusters $k$, skewness parameter $s$
**Ensure:** A center set $\mathcal{P}$ with $|\mathcal{P}| = k$
 1: **if** $|X| \leq k$ and $X \subset T$ **then**
 2:     $\mathcal{P} \leftarrow X$
 3: **else**
 4:     $\mathcal{P} \leftarrow$ Arbitrary subset of $T$ with $|\mathcal{P}| = k$
 5:     **for** $\mathcal{A} \in T^s$ **do**
 6:         $\mathcal{P}' \leftarrow$ FASTLOCALSEARCH$(S, T, \frac{\varepsilon}{2}, \mathcal{A}, k, s)$
 7:         **if** Cost$(S, \mathcal{P}') <$ Cost$(S, \mathcal{P})$ **then**
 8:             $\mathcal{P} \leftarrow \mathcal{P}'$
 9: **return** $\mathcal{P}$

---

**Theorem 5.2.** *Let $X$ be a set of $n$ data points, and let $T$ be a set of potential centers such that $|T| = \mathrm{poly}(n)$. Given any $\varepsilon > 0$, DISCRETEHEAVYSKEW returns a $(1 + \varepsilon)$-approximation $\mathcal{P}$ in $(nk\varepsilon)^{\mathcal{O}(s+1/\varepsilon)}$ time for discrete $(k, z)$-clustering as long as $X$ is $(s, 1-\varepsilon^{z+1})$-skewed. Furthermore, for $z = 1$, $X$ only needs to be $(s, 1 - \varepsilon)$-skewed.*

We then construct CONTINUOUSHEAVYSKEW, the algorithm deals with the continuous $(k, z)$-clustering problem. For a data set $X$, we can use CORESETCONSTRUCTION and CENTERNET to construct the coreset $S$ and potential center set $T$, effectively transforming the continuous $(k, z)$-clustering on $X$ into the discrete $(k, z)$-clustering on $(S, T)$. As a widely used sampling technique, sensitivity sampling can be completed in $\tilde{\mathcal{O}}(nk)$ running time. Our construction of $T$ also has a run time of $\mathrm{poly}(k, \log n)$ because the construction of an individual point in $T$ requires a run time of $\mathcal{O}(1)$, and $T$ has a size of $\mathrm{poly}(k, \log n)$. Then an application of DISCRETEHEAVYSKEW solves the problem.

---

**Algorithm 6** CONTINUOUSHEAVYSKEW$(X, \varepsilon, k, s)$

---

**Require:** Dataset $X$, precision $\varepsilon$, number of clusters $k$, skewness parameter $s$,
**Ensure:** A center set $\mathcal{P}$ with $|\mathcal{P}| = k$
 1: **if** $|X| \leq k$ **then**
 2:     $\mathcal{P} \leftarrow X$
 3: **else**
 4:     $S \leftarrow$ CORESETCONSTRUCTION$(X, \varepsilon, n, k, \Delta)$
 5:     $T \leftarrow$ CENTERNET$(S, \frac{\varepsilon}{4}, \Delta)$
 6:     $\mathcal{P} \leftarrow$ DISCRETEHEAVYSKEW$(X, T, \frac{\varepsilon}{4}, k, s)$
 7: **return** $\mathcal{P}$

---

We claim that CONTINUOUSHEAVYSKEW guarantee a $(1 + \varepsilon)$ approximation for $(k, z)$-clustering on $X$ within $\mathrm{poly}(n, k)$ run time if $X$ is $(s, 1 - \varepsilon^{z+1})$-skewed.

**Theorem 5.3.** *Let $X$ be a set of $n$ data points. Given any $\varepsilon > 0$, CONTINUOUSHEAVYSKEW returns a $(1+\varepsilon)$-approximation $\mathcal{P}$ in $\tilde{\mathcal{O}}(nk) + (k \log n)^{\tilde{\mathcal{O}}(s+1/\varepsilon)}$ time for continuous $(k, z)$-clustering with probability at least $0.97$, as long as $X$ is $(s, 1 - \varepsilon^{z+1})$-skewed. Furthermore, for $z = 1$, $X$ only needs to be $(s, 1 - \varepsilon)$-skewed.*

## 6 EXPERIMENTAL EVALUATIONS

Despite our primary focus on theoretical contributions, we performed experiments to validate its efficacy. We evaluated the precision of our algorithm against the $k$-means and $k$-medoids algorithms of the `scikit-learn` and `scikit-learn-extra` libraries. These algorithms are widely favored for their quick execution times, but they have weaker theoretical accuracy assurances. We opted to benchmark our algorithm against these fast yet less precise methods rather than other $(1 + \varepsilon)$ approximation algorithms, which are infeasible for experiment due to their exponential run times. Our empirical results demonstrate that our algorithm surpasses these commonly used methods in terms

Table 1: Improvement rate for $k$-means and $k$-medoids on synthetic data

| $k$ | $k$-means (%) | | | $k$-medoids (%) | | |
|---|---|---|---|---|---|---|
| | **Avg** | **Min** | **Median** | **Avg** | **Min** | **Median** |
| **4** | 28.32 | 3.78 | 12.77 | 16.86 | 1.11 | 10.57 |
| **5** | 27.16 | 5.32 | 20.07 | 25.87 | 14.22 | 25.49 |
| **6** | 28.83 | 12.57 | 26.91 | 40.41 | 21.08 | 39.74 |
| **7** | 47.95 | 7.44 | 45.21 | 21.04 | 10.95 | 15.12 |
| **8** | 50.53 | 40.83 | 48.82 | 34.19 | 8.25 | 40.36 |
| **9** | 57.89 | 23.52 | 28.99 | 39.29 | 18.46 | 22.34 |
| **10** | 37.23 | 24.42 | 26.28 | 42.65 | 22.85 | 47.17 |

of accuracy, thereby substantiating our theoretical claims for the $(1 + \varepsilon)$-accuracy of our algorithm with practical evidence.

Our experiment is conducted using Python 3.9.6 on a 2020 MacBook Pro equipped with a 1.4 GHz Quad-Core Intel Core i5 processor. We evaluate our algorithm against `KMeans` from `scikit-learn` and `KMedoids` from `scikit-learn-extra`. For all algorithms, we generate initialization centers through uniform sampling. A maximum iteration limit is set such that each algorithm updates at most $3 \cdot k$ centers by the time they terminate.

### 6.1 SYNTHETIC DATA

Synthetic data is produced using the `datagen` function from the `coreset` library. This function creates samples from a Dirichlet Process Mixture Model (DPMM) characterized by Gaussian likelihood and fixed cluster covariance, and operates based on the Chinese restaurant process. We set $s = t = 1$ and the smallest center net scale as $0.01$ for $k$-means.

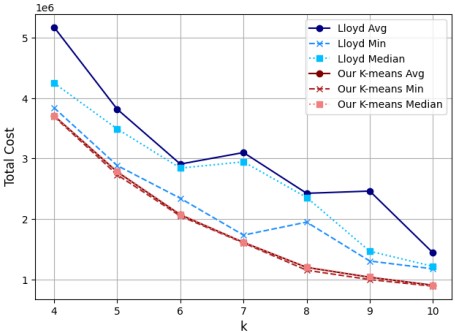

Figure 1: Comparison between Lloyd heuristic and our algorithm for $k$-means

Figure 2: Comparison between KMedoids and our algorithm for $k$-medoids

Our experiments illustrate an improvement range for $k$-means from $3.78\%$ at $k = 4$ for the minimum metric to $57.89\%$ at $k = 9$ for the average metric, and for $k$-medoids from $1.11\%$ at $k = 4$ for the minimum metric to $47.17\%$ at $k = 10$ for the median metric. This overall enhancement underscores the superior performance of our algorithm in terms of accuracy when compared to `KMeans` from `scikit-learn` and `KMedoids` from `scikit-learn-extra` across average, minimum, and median metrics. Furthermore, the notable improvement observed in the average and median metric implies a higher variability in `KMeans` and `KMedoids` when evaluated on synthetic data, whereas our algorithm demonstrates significantly lower variance.

### 6.2 REAL WORLD DATA

We also conducted the experiment using the Exasens dataset (Exa20) from the UCI Machine Learning Repository, which comprises 399 instances and 4 features. This data set includes demographic

Table 2: Improvement rate for $k$-means and $k$-medoids on real world data

| $k$ | $k$-means (%) | | | $k$-medoids (%) | | |
|---|---|---|---|---|---|---|
| | **Avg** | **Min** | **Median** | **Avg** | **Min** | **Median** |
| **4** | 82.49 | 83.53 | 82.39 | 32.50 | 17.84 | 22.68 |
| **5** | 82.58 | 5.87 | 85.69 | 24.98 | 23.69 | 24.77 |
| **6** | 86.11 | 21.90 | 88.61 | 31.01 | 29.30 | 29.02 |
| **7** | 89.94 | 39.12 | 91.79 | 30.22 | 11.48 | 32.48 |
| **8** | 84.56 | 18.04 | 36.86 | 38.04 | 35.90 | 37.24 |
| **9** | 86.69 | 28.79 | 51.80 | 41.24 | 39.89 | 41.28 |
| **10** | 88.91 | 41.94 | 38.71 | 42.02 | 40.29 | 42.67 |

information on 4 groups of saliva samples (COPD, asthma, infection, HC) collected as part of the joint research project Exasens. We utilized the `StandardScaler` from `scikit-learn`. The parameters used were identical to those used in the synthetic data experiment, with the exception of a reduced center net scale of $0.0001$, as the range of the real world data after scaling is approximately $100$ times smaller than that of the synthetic data.

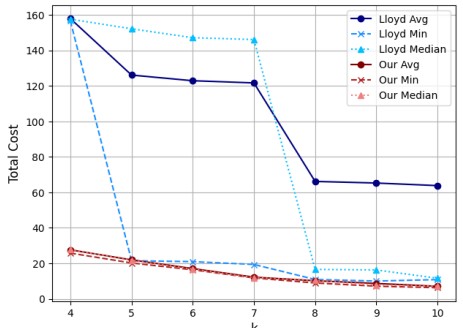

Figure 3: Comparison between Lloyd heuristic and our algorithm for $k$-means

Figure 4: Comparison between KMedoids and our algorithm for $k$-medoids

Our experimental results demonstrate an enhancement range for $k$-means from $5.87\%$ at $k = 5$ for the minimum metric up to $91.79\%$ at $k = 7$ for the median metric, and for $k$-medoids from $11.48\%$ at $k = 7$ for the minimum metric to $42.67\%$ at $k = 10$ for the median metric. This overall improvement highlights the superior accuracy performance of our algorithm relative to `KMeans` from `scikit-learn` and `KMedoids` from `scikit-learn-extra`, across various metrics including average, minimum, and median. Additionally, the observed substantial improvement in the average and median metric suggests greater variability in `KMeans` and `KMedoids` when tested on real world data, while our algorithm displays considerably lower variance. Notably, `KMeans` shows even higher variance with real-world data than with synthetic data, likely attributed to the increased skewness present in real-world datasets.

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

## A  RELATED WORK

Within this section, we present a review of related works. Initially, we discuss results studying the APX-hardness of $(k, z)$-clustering. Subsequently, we describe the progression of $(1 + \varepsilon)$-approximation algorithms. Thereafter, we briefly introduce the theoretical accuracy guarantees for popular algorithms used in practice. Additionally, we mention specific works on local search, an algorithmic paradigm integral to our approach. Lastly, we review some literature on Zipfian distributions.

**APX-hardness for $(k, z)$-clustering** The foundational work of Guha & Khuller (1999) was the first to prove that $(k, z)$-clustering is APX-hard. It established that $k$-means and $k$-median are hard to approximate within factors of $3.94$ and $1.73$, respectively, in general metric spaces. The natural question arises: Is $(k, z)$-clustering still APX-hard in more specific metrics, such as doubling or Euclidean metrics? Unfortunately, subsequent studies have confirmed that $(k, z)$-clustering remains APX-hard even under these specific metrics (Ahmadian et al., 2019; Trevisan, 2000; Guruswami & Indyk, 2003; Cohen-Addad & Karthik, 2019). According to the most recent research by Cohen-Addad & Lee (2022), the inapproximability bounds are $1.17$ and $1.06$ for discrete and continuous $k$-means, and $1.07$ and $1.015$ for discrete and continuous $k$-median in Euclidean space unless P=NP.

**Development of $(1 + \varepsilon)$-approximation algorithms** Early attempts at developing $(1 + \varepsilon)$-approximation algorithms for $k$-means clustering began with Inaba et al. (1994), who proposed an algorithm with a run time of $\mathcal{O}(n^{dk+1})$ for fixed $k$ and $\varepsilon$. Subsequent work improved the runtime

(Matoušek, 2000; Har-Peled & Mazumdar, 2004), culminating in De La Vega et al. (2003) presenting the first algorithm with a linear dependency on $n$. Kumar et al. (2004; 2005); Chen (2006) further improved the run time with a new coreset construction. Finally, Feldman et al. (2007) developed a PTAS with a run time of $\mathcal{O}(nkd + 2^{\tilde{\mathcal{O}}(k/\varepsilon)})$. However, all these PTASs assume fixed $k$ and $\varepsilon$, resulting in algorithms that are polynomial in $n$ but have exponential dependency on $k$.

**Popular practical algorithms** Lloyd (1982) introduced the Lloyd heuristic, the most widely used algorithm for $k$-means in practice. This algorithm iteratively computes the centroid of each cluster to search of a local optimum. However, despite its popularity, Inaba et al. (1994) demonstrated that the Lloyd heuristic does not guarantee a solution close to the optimal $k$-means clustering in the worst case. To address this, Arthur & Vassilvitskii (2006) proposed $k$-means++, an initialization process that provides an $\mathcal{O}(\log k)$-approximation guarantee when combined with the Lloyd heuristic. Together, these algorithms achieve a total runtime of $\tilde{\mathcal{O}}(dnk)$. For $k$-medoids, the most popular algorithm is the PAM (Partitioning Around Medoids) algorithm, proposed by Rdusseeun & Kaufman (1987). PAM can be seen as a discrete counterpart to the Lloyd heuristic. However, PAM lacks a theoretical guarantee and has a runtime of $\mathcal{O}(T \cdot k(n-k)^2)$, where $T$ is the number of iterations.

**Local search technique.** The local search technique, introduced by Arya et al. (2001), iteratively swaps $t$ centers to seek a local optimum solution. Arya et al. (2001) demonstrated that local search guarantees a $\left(3 + \frac{2}{t}\right)$-approximation for $k$-median, while Kanungo et al. (2002) showed a $(9 + \varepsilon)$-approximation for $k$-means. Cohen-Addad et al. (2019) established that local search is a PTAS for $k$-means and $k$-median in constant-dimensional Euclidean space, and Friggstad et al. (2019) demonstrated that local search is a PTAS in doubling metric spaces. Due to its simplicity, local search is frequently used as a subroutine for clustering in various computational models, such as distributed (Bateni et al., 2014), parallel (Blelloch & Tangwongsan, 2010), and streaming environments (Guha et al., 2003). In addition, numerous studies have also examined local search from a theoretical perspective (Cohen-Steiner et al., 2004; Dhillon et al., 2002; Friggstad & Zhang, 2016; Hansen & Mladenović, 2001; Yang et al., 2008). Although traditionally recognized as a constant approximation algorithm, Cohen-Addad & Schwiegelshohn (2017) explored its performance on data sets with specific properties, showing that local search can achieve a $(1 + \varepsilon)$-approximation for certain datasets, such as those with distributional stability.

**Zipfian distribution.** Zipf's law, as proposed by Zipf (1949), characterizes an empirical distribution found in numerous real-world datasets. Mandelbrot et al. (1953) refined this law by adding an exponent parameter $p$, leading to the Mandelbrot-Zipf law, which serves as a more generalized model for linguistic phenomena. In present-day network science, Zipf's law is relevant to the analysis of scale-free networks, where the degree distribution (the number of connections a node has) frequently follows a power law, akin to a Zipfian distribution. Significant advancements in understanding such networks were made by Barabási & Albert (1999) with their preferential attachment model. Halevy et al. (2009) discuss how large-scale data processing often unveils Zipfian distributions in real-world datasets, such as those pertaining to web queries and clickstream data. Additionally, the Mandelbrot-Zipf law is observed across various domains including economics (Gabaix, 1999), geography (Jiang et al., 2015), genomics (Furusawa & Kaneko, 2003), language (Ferrer i Cancho, 2005), and security (Blocki et al., 2018).

## B $(s, 1 - \varepsilon)$-SKEWED DATASET AND ZIPFIAN DISTRIBUTION

We will prove Lemma 2.3 in this section.

**Lemma B.1.** *Let $X = \{x_1, x_2, \ldots, x_n\} \subseteq [\Delta]^d$ be a Zipfian distribution dataset. There exists a constant $\gamma > 0$ such that for $s > \gamma \left(\frac{1}{\varepsilon}\right)^{\frac{1}{p-1}}$, $X$ is $(s, 1 - \varepsilon)$-skewed.*

*Proof.* Since $\frac{1}{x^p}$ is continuous and decreasing on $\mathbb{R}_{>0}$,

$$\int_i^{i+1} \frac{1}{x^p} dx \leq \frac{1}{i^p} \leq \int_{i-1}^i \frac{1}{x^p} dx.$$

Hence

$$\sum_{i=s+1}^\infty \frac{1}{i^p} \leq \int_s^\infty \frac{1}{x^p} dx = \frac{1}{p-1} \cdot \frac{1}{s^{p-1}}.$$

For $s > \gamma \left( \frac{1}{\varepsilon} \right)^{\frac{1}{p-1}}$, substituting this into the above inequality, it yields

$$\sum_{i=s+1}^{\infty} \frac{1}{i^s} \leq \frac{1}{p-1} \cdot \frac{\varepsilon}{\gamma^{p-1}}.$$

By the definition of Zipfian distribution dataset, we have

$$\gamma_1 \cdot \frac{1}{i^p} \leq \mathrm{Cost}(N(c_i), \mathcal{C}) \leq \gamma_2 \cdot \frac{1}{i^p}.$$

Hence

$$\sum_{i=s+1}^{k} \mathrm{Cost}(N(c_i), \mathcal{C}) \leq \sum_{i=s+1}^{k} \gamma_2 \cdot \frac{1}{i^p} \leq \sum_{i=s+1}^{\infty} \gamma_2 \cdot \frac{1}{i^p} \leq \frac{\gamma_2}{p-1} \cdot \frac{\varepsilon}{\gamma^{p-1}}.$$

On the other hand, we know that

$$\sum_{i=1}^{\infty} \frac{1}{i^p} = \zeta(p)$$

for $p > 1$. Thus

$$\mathrm{Cost}(X, \mathcal{C}) = \sum_{i=1}^{k} \mathrm{Cost}(N(c_i), \mathcal{C}) \geq \sum_{i=1}^{k} \gamma_1 \cdot \frac{1}{i^p} \geq \gamma_1 \cdot \zeta(p).$$

There exists a constant $\gamma > 0$ such that

$$\frac{\gamma_2}{p-1} \cdot \frac{\varepsilon}{\gamma^{p-1}} \leq \gamma_1 \cdot \zeta(p).$$

Hence for $s \geq \gamma \left( \frac{1}{\varepsilon} \right)^{\frac{1}{p-1}}$,

$$\sum_{i>s} \mathrm{Cost}(N(c_i), \mathcal{C}) \leq \varepsilon \cdot \mathrm{Cost}(X, \mathcal{C}),$$

which is equivalent to

$$\sum_{i \leq s} \mathrm{Cost}(N(c_i), \mathcal{C}) \geq (1-\varepsilon)\mathrm{Cost}(X, \mathcal{C}).$$

$\square$

Hence a Zipfian distribution is a $(s, 1-\varepsilon)$-skewed for $s = \mathcal{O}(\left( \frac{1}{\varepsilon} \right)^{\frac{1}{p-1}})$.

## C    CORESET AND CENTER NET

In Appendix C.1, we will prove Lemma 3.2. In Appendix C.2, we will also introduce an algorithm that produces a center net, providing a $(1+\varepsilon)$-approximation.

### C.1    CORESET THAT KEEPS HEAVY SKEWNESS

Before proving Lemma 3.2, we shall first revisit Bernstein's inequality, as it is essential for the subsequent proof.

**Theorem C.1** (Bernstein's inequality). *Let $Z_1, Z_2, \cdots, Z_n$ be independent random variables and $a_i \leq Z_i \leq b_i$. Let $S_n = \sum_{i=1}^{n} Z_i$, $E_n = \mathbb{E}[S_n]$, and $R \geq \max_{i \in [n]} |b_i - a_i|$. Then for any $t > 0$,*

$$\mathbf{Pr}\left[|S_n - E_n| > t\right] < 2\exp\left(-\frac{t^2/2}{V_n + R \cdot t/3}\right).$$

We first prove that under the condition of Lemma 3.2, CORESETCONSTRUCTION returns an $\varepsilon$-coreset $S$ of $X$ with probability at least $0.99$.

**Lemma C.2.** *Let $X = \{x_1, x_2, \cdots, x_n\} \subset [\Delta]^d$ be a $(s, 1 - \varepsilon)$-skewed dataset. There exists a constant $\gamma > 0$, such that for any $\varepsilon \in (0, \frac{1}{4}]$, the set $S$ returned by CORESETCONSTRUCTION$(X, \varepsilon, n, k, \Delta)$ is an $\varepsilon$-coreset of $X$ with probability at least $0.99$ if $\mu = \frac{\gamma dk}{\varepsilon^3} \log(n\Delta)$.*

*Proof.* We want to use Bernstein's inequality to bound the probability. For any $\mathcal{C} \in \left(\mathbb{R}^d\right)^k$, we define the random variable to describe the cost of $S$:

$$Z_i = \begin{cases} w(x_i) \cdot \mathrm{Cost}\left(x_i, \mathcal{C}\right), & \text{with probability } p_x, \\ 0, & \text{with probability } 1 - p_x. \end{cases}$$

Let $S_n = \sum_{i=1}^n Z_i$. Then $\mathrm{Cost}\left(S, \mathcal{C}\right) = S_n$.

Denote $E_n = \mathbb{E}\left[S_n\right]$, then

$$E_n = \sum_{i=1}^n w(x_i) \cdot \mathrm{Cost}\left(x_i, \mathcal{C}\right) \cdot p_i.$$

According to the algorithm, $w(x_i) = \frac{1}{p_x}$, so

$$E_n = \sum_{i=1}^n \frac{1}{p_x} \cdot \mathrm{Cost}\left(x_i, \mathcal{C}\right) \cdot p_i = \sum_{i=1}^n \mathrm{Cost}\left(x_i, \mathcal{C}\right) = \mathrm{Cost}\left(X, \mathcal{C}\right).$$

Next, we analyze the variance of $Z_i$. Let $V_n = \mathrm{Var}\left(S_n\right)$. Recall that the variance of a random variable is bounded by its second moment, so

$$\mathrm{Var}\left(Z_i\right) \leq \mathbb{E}\left[Z_i^2\right] = \frac{1}{p_x^2} \cdot \mathrm{Cost}\left(x_i, \mathcal{C}\right)^2 \cdot p_i = \frac{1}{p_x} \cdot \mathrm{Cost}\left(x_i, \mathcal{C}\right)^2.$$

Recall that $p_x = \min\{\mu s(x), 1\}$. For the case $\mu s(x) \leq 1$,

$$\mathrm{Var}\left(Z_i\right) \leq \frac{1}{\mu s(x)} \mathrm{Cost}\left(x_i, \mathcal{C}\right)^2.$$

Recall the definition of $s(x)$,

$$s(x) = \max_{\mathcal{C}' \in (\mathbb{R}^d)^k} \frac{\mathrm{Cost}(x, \mathcal{C}')}{\mathrm{Cost}(X, \mathcal{C}')}.$$

Therefore

$$\mathrm{Var}\left(Z_i\right) \leq \frac{1}{\mu} \frac{\mathrm{Cost}(X, \mathcal{C})}{\mathrm{Cost}(x_i, \mathcal{C})} \mathrm{Cost}(x_i, \mathcal{C})^2 = \frac{1}{\mu} \mathrm{Cost}(X, \mathcal{C}) \mathrm{Cost}(x_i, \mathcal{C}).$$

Hence

$$\sum_{\mu s(x_i) \leq 1} \mathrm{Var}\left(Z_i\right) \leq \sum_{\mu s(x_i) \leq 1} \frac{1}{\mu} \mathrm{Cost}(X, \mathcal{C}) \mathrm{Cost}(x_i, \mathcal{C})$$

$$\leq \sum_{i=1}^n \frac{1}{\mu} \mathrm{Cost}(X, \mathcal{C}) \mathrm{Cost}(x_i, \mathcal{C})$$

$$= \frac{1}{\mu} \mathrm{Cost}(X, \mathcal{C})^2.$$

For the case $\mu s(x) > 1$, we have $p_x = 1$. Hence

$$\mathrm{Var}\left(Z_i\right) = \mathbb{E}\left[Z_i^2\right] - \left(\mathbb{E}\left[Z_i\right]\right)^2 = \mathrm{Cost}(x_i, \mathcal{C})^2 \cdot 1 - \mathrm{Cost}(x_i, \mathcal{C})^2 = 0.$$

Then

$$\sum_{\mu s(x_i) > 1} \mathrm{Var}\left(Z_i\right) = 0.$$

Thus

$$V_n = \sum_{\mu s(x_i) \leq 1} \mathrm{Var}\left(Z_i\right) + \sum_{\mu s(x_i) > 1} \mathrm{Var}\left(Z_i\right) \leq \frac{1}{\mu} \mathrm{Cost}(X, \mathcal{C})^2.$$

Next we analyze the range bound $R$. For the lower bound, $0 \leq Z_i$ for any $i \in [n]$. For the upper bound, by the definition of $Z_i$, for the case $p(x_i) = \mu s(x_i)$,

$$Z_i = \frac{1}{\mu s(x_i)} \mathrm{Cost}(x_i, \mathcal{C}) \leq \frac{1}{\mu} \frac{\mathrm{Cost}(X, \mathcal{C})}{\mathrm{Cost}(x_i, \mathcal{C})} \mathrm{Cost}(x_i, \mathcal{C}) = \frac{1}{\mu} \mathrm{Cost}(X, \mathcal{C}).$$

For the case $p(x_i) = 1$,

$$Z_i = \text{Cost}(x_i, \mathcal{C}) \leq \text{Cost}(X, \mathcal{C}).$$

Hence $Z_i \leq \text{Cost}(X, \mathcal{C})$ for any $i \in [n]$.

Then by Bernstein's inequality,

$$\mathbf{Pr}\left[|S_n - E_n| > \varepsilon E_n\right] < 2\exp\left(-\frac{(\varepsilon E_n)^2/2}{V_n + R \cdot \varepsilon E_n/3}\right)$$

$$\leq 2\exp\left(-\frac{\varepsilon^2 \text{Cost}(X, \mathcal{C})^2/8}{\frac{1}{\mu}\text{Cost}(X, \mathcal{C})^2 + \varepsilon \text{Cost}(X, \mathcal{C})^2/6}\right)$$

$$= 2\exp\left(-\frac{\varepsilon^2/2}{\frac{1}{\mu} + \varepsilon/6}\right).$$

Since $\varepsilon \in (0, \frac{1}{4}]$ and $\mu = \frac{\gamma dk}{\varepsilon^3}\log(n\Delta)$, there exists $\gamma > 0$ such that $\mu \geq 1 \geq \frac{\varepsilon}{6}$. Hence

$$\mathbf{Pr}\left[|S_n - E_n| > \varepsilon E_n\right] < 2\exp\left(-\frac{\varepsilon^2/2}{\frac{1}{\mu} + \frac{1}{\mu}}\right) \leq 2\exp\left(-\frac{\mu\varepsilon^2}{4}\right).$$

By Cohen-Addad et al. (2023), there exists a collection of center set $\mathcal{F}$ that gives a good approximation for any center set, and the guarantee of $(1 + \varepsilon)$-approximation on $\mathcal{F}$ implies the $(1 + \varepsilon)$-approximation for any center set.

**Lemma C.3** (Lemma 3.2 in (Cohen-Addad et al., 2023)). *Let $X \subset [\Delta]^d$ and let $z \geq 1$ be a constant. Then there exists a set $\mathcal{F}$ of size $|\mathcal{F}| = \left(\frac{n\Delta}{\varepsilon}\right)^{\mathcal{O}(kd)}$, such that $(1 - \varepsilon)\text{Cost}(X, \mathcal{C}) \leq \text{Cost}(S, \mathcal{C}) \leq (1 + \varepsilon)\text{Cost}(X, \mathcal{C})$ for any $\mathcal{C} \in \mathcal{F}$, implies $(1 - \varepsilon)\text{Cost}(X, \mathcal{C}) \leq \text{Cost}(S, \mathcal{C}) \leq (1 + \varepsilon)\text{Cost}(X, \mathcal{C})$ for any set $\mathcal{C} \subset \mathbb{R}^d$ with $|\mathcal{C}| = k$.*

Denote $\mathcal{E}$ as the event that $(1 - \varepsilon)\text{Cost}(X, \mathcal{C}) \leq \text{Cost}(S, \mathcal{C}) \leq (1 + \varepsilon)\text{Cost}(X, \mathcal{C})$ for any $\mathcal{C} \in \mathcal{F}$. Notice that $(1 - \varepsilon)\text{Cost}(S, \mathcal{C}) \leq \text{Cost}(X, \mathcal{C}) \leq (1 + \varepsilon)\text{Cost}(S, \mathcal{C})$ is equivalent to $|\text{Cost}(S, \mathcal{C}) - \text{Cost}(X, \mathcal{C})| \leq \varepsilon\text{Cost}(X, \mathcal{C})$. By taking a union bound, we get

$$\mathbf{Pr}\left[\mathcal{E}\right] \geq 1 - |\mathcal{F}| \cdot 2\exp\left(-\frac{\mu\varepsilon^2}{4}\right).$$

Since $|\mathcal{F}| = \left(\frac{n\Delta}{\varepsilon}\right)^{\mathcal{O}(kd)}$ and $\mu = \frac{\gamma dk}{\varepsilon^3}\log(n\Delta)$, we get

$$\mathbf{Pr}\left[\mathcal{E}\right] \geq 1 - \exp\left(\mathcal{O}(dk\log\frac{n\Delta}{\varepsilon}) - \frac{\gamma dk}{4\varepsilon}\log(n\Delta)\right).$$

Thus there exists any constant $\gamma > 0$ such that $\mathbf{Pr}\left[\mathcal{E}\right] \geq 0.99$.

Then by Lemma C.3, with probability at least $0.99$, $(1 - \varepsilon)\text{Cost}(X, \mathcal{C}) \leq \text{Cost}(S, \mathcal{C}) \leq (1 + \varepsilon)\text{Cost}(X, \mathcal{C})$ for any set $\mathcal{C} \subset \mathbb{R}^d$ with $|\mathcal{C}| = k$, which is equivalent to that $S$ is an $\varepsilon$-coreset of $X$. □

Next, we prove that under the condition of Lemma 3.2, $S$ has a size of $|S| = \mathcal{O}(\frac{dk^2}{\varepsilon^3}\log(n\Delta))$.

**Lemma C.4.** *Let $X = \{x_1, x_2, \cdots, x_n\} \subset [\Delta]^d$ be a $(s, 1 - \varepsilon)$-skewed dataset. There exists a constant $\gamma > 0$, such that for any $\varepsilon \in (0, \frac{1}{4}]$, $S$ has a size of $|S| = \mathcal{O}(\frac{dk^2}{\varepsilon^3}\log(n\Delta))$ with probability at least $0.99$ if $\mu = \frac{\gamma dk}{\varepsilon^3}\log(n\Delta)$.*

*Proof.* The proof is similar to the proof of Lemma C.2. We use Bernstein's inequality to bound the probability.

Define the random variable

$$Z_i = \begin{cases} 1, & \text{with probability } p_x, \\ 0, & \text{with probability } 1 - p_x. \end{cases}$$

Denote $S_n = \sum_{i=1}^{n} Z_i$. Since $Z_i$ describe whether we sample the point $x_i$ or not, $|S| = S_n$. Let $E_n = \mathbb{E}[S_n]$ and $V_n = \text{Var}(S_n)$.

Since $p_x = \min\{\mu s(x), 1\}$, we get $E_n = \sum_{i=1}^{n} \mathbb{E}[Z_i] \leq \sum_{i=1}^{n} \mu s(x_i)$. Varadarajan & Xiao (2012) proves that for $(k, z)$-clustering, $\sum_{i=1}^{n} s(x_i) = \mathcal{O}(k)$. Hence there exists some constant $\gamma' > 0$, such that $\sum_{i=1}^{n} s(x_i) \leq \gamma' k$. Then $E_n \leq \gamma' \mu k$.

Since $Z_i$ is a Bernoulli random variable, $\text{Var}(Z_i) = p_{x_i}(1 - p_{x_i}) \leq p_{x_i}$. Hence

$$V_n = \sum_{i=1}^{n} \text{Var}(Z_i) \leq \sum_{i=1}^{n} p_{x_i} \leq \sum_{i=1}^{n} \mu s(x_i) = \gamma' \mu k.$$

For the range bound, we have $0 \leq Z_i \leq 1$ for any $i$. Then by Bernstein's inequality,

$$\mathbf{Pr}\left[|S_n - E_n| > \gamma' \mu k\right] < 2 \exp\left(-\frac{(\gamma' \mu k)^2 / 2}{V_n + R \cdot E_n / 3}\right)$$

$$\leq 2 \exp\left(-\frac{\gamma'^2 \mu^2 k^2 / 2}{\gamma' \mu k + \gamma' \mu k / 3}\right)$$

$$\leq 2 \exp\left(-\frac{\gamma' \mu k}{4}\right).$$

Since $\mu = \frac{\gamma dk}{\varepsilon^3} \log(n\Delta)$, there exists $\gamma > 0$ such that

$$\mathbf{Pr}\left[|S_n - E_n| > \gamma' \mu k\right] < 0.01.$$

Then with probability at least $0.99$,

$$|S| = S_n \leq E_n + \gamma' \mu k \leq 2\gamma' \mu k = \mathcal{O}(\frac{dk^2}{\varepsilon^3} \log(n\Delta))$$

$\square$

Finally, we demonstrate that under the assumption of Lemma 3.2, $S$ exhibits significant skewness. Our proof establishes that the coreset $S$ not only provides an accurate approximation of $X$, but also effectively approximates the expensive clusters $N_X(c_i)$. Specifically, we assert that $S$ offers a $(1 + \varepsilon)$-approximation for clusters whose cost exceeds $\frac{\varepsilon}{100k}\text{Cost}(X, \mathcal{C}_{\text{OPT}})$, with $\mathcal{C}_{\text{OPT}}$ representing the optimal solution.

**Lemma C.5.** *Let $X = \{x_1, x_2, \cdots, x_n\} \subset [\Delta]^d$ be a $(s, 1 - \varepsilon)$-skewed dataset. Let $\mathcal{C}_{\text{OPT}} = \{c_1, c_2, \cdots, c_k\}$ be the optimal solution. Let $N_X(c_i) = \{x \in X : dist(x, c_i) \leq dist(x, c_j), j \neq i\}$. Assume $N_X(c_i)$ is ordered in the way that $\text{Cost}(N_X(c_i), \mathcal{C}_{\text{OPT}}) \geq \text{Cost}(N_X(c_j), \mathcal{C}_{\text{OPT}})$ for $j > i$. There exists a constant $\gamma > 0$, such that for any $\varepsilon \in (0, 1]$, if $\mu = \frac{\gamma dk}{\varepsilon^3} \log(n\Delta)$, with probability at least $0.99$, $\text{Cost}(N_S(c_i), \mathcal{C}_{\text{OPT}}) \in (1 \pm \varepsilon)\text{Cost}(N_X(c_i), \mathcal{C}_{\text{OPT}})$ for $\text{Cost}(N_X(c_i), \mathcal{C}_{\text{OPT}}) \geq \frac{\varepsilon}{100k}\text{Cost}(X, \mathcal{C}_{\text{OPT}})$, where $N_S(c_i)$ is the set of points in $S$ that sampled from $N_X(c_i)$, and $S$ is the set returned by $\text{CORESETCONSTRUCTION}(X, \varepsilon, n, k, \Delta)$.*

*Proof.* Let $N_X(c_i) = \{x_1, x_2, \cdots, x_m\}$ be a cluster such that $\text{Cost}(N_X(c_i), \mathcal{C}_{\text{OPT}}) \geq \frac{\varepsilon}{100k}\text{Cost}(X, \mathcal{C}_{\text{OPT}})$. For $j \in [m]$, we define

$$Z_j = \begin{cases} w(x_j) \cdot \text{Cost}(x_j, \mathcal{C}_{\text{OPT}}), & \text{with probability } p_x, \\ 0, & \text{with probability } 1 - p_x. \end{cases}$$

Let $S_i = \sum_{x_j \in X_i} Z_j$, $E_i = \mathbb{E}[S_i]$ and $V_i = \text{Var}(S_i)$.

By the same proof as the one of Lemma C.2, we claim that for the case $p_{x_j} = 1$, $\text{Var}(Z_j) = 0$, and for the case $p_{x_j} = \mu s(x_j)$,

$$\text{Var}(Z_j) \leq \frac{1}{\mu}\text{Cost}(X, \mathcal{C}_{\text{OPT}})\text{Cost}(x_j, \mathcal{C}_{\text{OPT}}).$$

We also have

$$E_i = \sum_{j=1}^{m} \mathbb{E}\left[Z_j\right] = \sum_{j=1}^{m} \frac{1}{p_{x_j}} \mathrm{Cost}(x_j, \mathcal{C}_{\mathrm{OPT}}) p_{x_j} = \mathrm{Cost}(N_X(c_i), \mathcal{C}_{\mathrm{OPT}}).$$

$E_i$ is $\mathrm{Cost}(N_X(c_i), \mathcal{C}_{\mathrm{OPT}})$ here, which is different from the expextation in the proof of Lemma C.2. It is because we only add the points in $N_X(c_i)$ here, and we add all points in $X$ in the proof of Lemma C.2.

Similarly, for $V_i$, we have

$$\sum_{x_j \in N_X(c_i): \mu s(x_i) \leq 1} \mathrm{Var}\left(Z_i\right) \leq \sum_{x_j \in N_X(c_i): \mu s(x_i) \leq 1} \frac{1}{\mu} \mathrm{Cost}(X, \mathcal{C}_{\mathrm{OPT}}) \mathrm{Cost}(x_j, \mathcal{C}_{\mathrm{OPT}})$$

$$\leq \sum_{x_j \in N_X(c_i)} \frac{1}{\mu} \mathrm{Cost}(X, \mathcal{C}_{\mathrm{OPT}}) \mathrm{Cost}(x_j, \mathcal{C}_{\mathrm{OPT}})$$

$$= \frac{1}{\mu} \mathrm{Cost}(X, \mathcal{C}_{\mathrm{OPT}}) \mathrm{Cost}(N_X(c_i), \mathcal{C}_{\mathrm{OPT}}),$$

and

$$\sum_{x_j \in N_X(c_i): \mu s(x_i) < 1} \mathrm{Var}\left(Z_i\right) = 0.$$

Hence

$$V_n = \sum_{x_j \in N_X(c_i): \mu s(x_i) \leq 1} \mathrm{Var}\left(Z_i\right) + \sum_{x_j \in N_X(c_i): \mu s(x_i) > 1} \mathrm{Var}\left(Z_i\right)$$

$$\leq \frac{1}{\mu} \mathrm{Cost}(X, \mathcal{C}_{\mathrm{OPT}}) \mathrm{Cost}(N_X(c_i), \mathcal{C}_{\mathrm{OPT}}).$$

By the same proof of Lemma C.2, for the bound of $Z_j$, we have $0 \leq Z_j \leq \mathrm{Cost}(X, \mathcal{C}_{\mathrm{OPT}})$ for any $j \in [m]$.

Now we have $E_n = \mathrm{Cost}(N_X(c_i), \mathcal{C}_{\mathrm{OPT}})$, $R = \mathrm{Cost}(X, \mathcal{C}_{\mathrm{OPT}})$, and

$$V_n \leq \frac{1}{\mu} \mathrm{Cost}(X, \mathcal{C}_{\mathrm{OPT}}) \mathrm{Cost}(N_X(c_i), \mathcal{C}_{\mathrm{OPT}}) = \frac{1}{\mu} R \cdot E_n.$$

Then by Bernstein's inequality,

$$\mathbf{Pr}\left[|S_n - E_n| > \varepsilon E_n\right] < 2 \exp\left(-\frac{(\varepsilon E_n)^2/2}{V_n + R \cdot \varepsilon E_n/3}\right)$$

$$\leq 2 \exp\left(-\frac{\varepsilon^2 E_n^2/2}{\frac{1}{\mu} R \cdot E_n + \varepsilon R \cdot E_n/6}\right)$$

$$= 2 \exp\left(-\frac{\varepsilon^2 E_n/2}{\left(\frac{1}{\mu} + \frac{\varepsilon}{6}\right) R}\right).$$

Since $\varepsilon \in (0, \frac{1}{4}]$, there exists $\gamma > 0$ such that $\mu > 1$. Then $\frac{1}{\mu} + \frac{\varepsilon}{6} \leq \frac{2}{\mu}$. Thus

$$\mathbf{Pr}\left[|S_n - E_n| > \varepsilon E_n\right] < 2 \exp\left(-\frac{\varepsilon^2 \mu E_n}{4R}\right).$$

Since $\mathrm{Cost}(N_X(c_i), \mathcal{C}_{\mathrm{OPT}}) \geq \frac{\varepsilon}{100k} \mathrm{Cost}(X, \mathcal{C}_{\mathrm{OPT}})$, we get $\frac{E_n}{R} \geq \frac{\varepsilon}{100k}$. Recall that $\mu = \frac{\gamma dk}{\varepsilon^3} \log(n\Delta)$. Then

$$\mathbf{Pr}\left[|S_n - E_n| > \varepsilon E_n\right] < 2 \exp\left(-\frac{\varepsilon^3 \mu}{400k}\right)$$

$$= 2 \exp\left(-\frac{\gamma d}{400} \log(n\Delta)\right).$$

Then there exists some constant $\gamma > 0$ such that

$$\mathbf{Pr}\left[|S_n - E_n| > \varepsilon E_n\right] \leq 2\exp\left(-\frac{\gamma d}{400}\log\left(n\Delta\right)\right) \leq \frac{1}{100n}.$$

It means for a cluster $N_X(c_i)$ that $\text{Cost}(N_X(c_i), \mathcal{C}_{\text{OPT}}) \geq \frac{\varepsilon}{100k}\text{Cost}(X, \mathcal{C}_{\text{OPT}})$, with probability at least $1 - \frac{1}{100n}$, we have $|\text{Cost}(N_X(c_i), \mathcal{C}_{\text{OPT}}) - \text{Cost}(N_S(c_i), \mathcal{C}_{\text{OPT}})| \leq \varepsilon\text{Cost}(N_X(c_i), \mathcal{C}_{\text{OPT}})$.

Since we have $k$ clusters in total, the number of the clusters $N_X(c_i)$ that $\text{Cost}(X_i, \mathcal{C}_{\text{OPT}}) \geq \frac{\varepsilon}{100k}\text{Cost}(X, \mathcal{C}_{\text{OPT}})$ is at most $k$. By taking a union bound, we get that $|\text{Cost}(N_X(c_i), \mathcal{C}_{\text{OPT}}) - \text{Cost}(N_S(c_i), \mathcal{C}_{\text{OPT}})| \leq \varepsilon\text{Cost}(N_X(c_i), \mathcal{C}_{\text{OPT}})$ for any $\text{Cost}(N_X(c_i), \mathcal{C}_{\text{OPT}}) \geq \frac{\varepsilon}{100k}\text{Cost}(X, \mathcal{C}_{\text{OPT}})$ with probability at least $1 - \frac{k}{100n} \geq 0.99$. $\qquad\square$

Finally, we complete the proof of Lemma 3.2.

**Lemma C.6.** *Let $X$ be an $(s, 1 - \varepsilon)$-skewed dataset. There exists a constant $\gamma > 1$, such that for any $\varepsilon \in (0, \frac{1}{4}]$,* CORESETCONSTRUCTION *returns an $\varepsilon$-coreset $S$ for $X$ with probability at least 0.97. Furthermore, $S$ is $(s, 1 - 3\varepsilon)$-skewed, and has a size of $\mathcal{O}(\frac{dk^2}{\varepsilon^3}\log(n\Delta))$.*

*Proof.* By Lemma C.2, Lemma C.4 and Lemma C.5, we get that with probability at least 0.97, $S$ is an $\varepsilon$-coreset of $X$, $|S| = \mathcal{O}(\frac{dk^2}{\varepsilon^3}\log\left(n\Delta\right))$, and $|\text{Cost}(N_X(c_i), \mathcal{C}_{\text{OPT}}) - \text{Cost}(N_S(c_i), \mathcal{C}_{\text{OPT}})| \leq \varepsilon\text{Cost}(N_X(c_i), \mathcal{C}_{\text{OPT}})$ for any cluster $N_X(c_i)$ that $\text{Cost}(N_X(c_i), \mathcal{C}_{\text{OPT}}) \geq \frac{\varepsilon}{100k}\text{Cost}(X, \mathcal{C}_{\text{OPT}})$.

What we remain to prove is that $S$ is a $(s, 1 - \varepsilon)$-skewed set.

WLOG, we can order the clusters $N_X(c_i)$ in the way that $\text{Cost}(N_X(c_i), \mathcal{C}_{\text{OPT}}) \geq \text{Cost}(N_X(c_j), \mathcal{C}_{\text{OPT}})$ for $i < j$. We divide $\{N_X(c_i)\}_{i=1}^n$ into two part, the heavy ones $\mathcal{H} = \{1, \cdots, m\}$ and the light ones $\mathcal{L} = \{m+1, \cdots, k\}$, such that for any $i \in \mathcal{H}$, $\text{Cost}(N_X(c_i), \mathcal{C}_{\text{OPT}}) \geq \frac{\varepsilon}{100k}\text{Cost}(X, \mathcal{C}_{\text{OPT}})$, and for any $i \in \mathcal{L}$, $\text{Cost}(N_X(c_i), \mathcal{C}_{\text{OPT}}) < \frac{\varepsilon}{100k}\text{Cost}(X, \mathcal{C}_{\text{OPT}})$.

Notice that the heavy clusters $\mathcal{L}$ contribute at most $\frac{\varepsilon}{100}\text{Cost}(X, \mathcal{C}_{\text{OPT}})$. In fact, for the sum of $N_X(c_i)$ where $i \in \mathcal{L}$,

$$\sum_{i \in \mathcal{L}}\text{Cost}(N_X(c_i), \mathcal{C}_{\text{OPT}}) \leq \sum_{i \in \mathcal{L}}\frac{\varepsilon}{100k}\text{Cost}(X, \mathcal{C}_{\text{OPT}})$$

$$\leq \sum_{i=1}^k \frac{\varepsilon}{100k}\text{Cost}(X, \mathcal{C}_{\text{OPT}})$$

$$= \frac{\varepsilon}{100}\text{Cost}(X, \mathcal{C}_{\text{OPT}}).$$

We divide the sum of the $s$ most heaviest clusters into two part:

$$\sum_{i \in [s]}\text{Cost}(N_X(c_i), \mathcal{C}_{\text{OPT}}) = \sum_{i \in [s] \cap \mathcal{H}}\text{Cost}(N_X(c_i), \mathcal{C}_{\text{OPT}}) + \sum_{i \in [s] \cap \mathcal{L}}\text{Cost}(N_X(c_i), \mathcal{C}_{\text{OPT}}).$$

Since $X$ is a $(s, 1 - \varepsilon)$-skewed set,

$$\sum_{i \in [s] \cap \mathcal{H}}\text{Cost}(N_X(c_i), \mathcal{C}_{\text{OPT}}) = \sum_{i \in [s]}\text{Cost}(N_X(c_i), \mathcal{C}_{\text{OPT}}) - \sum_{i \in [s] \cap \mathcal{L}}\text{Cost}(N_X(c_i), \mathcal{C}_{\text{OPT}})$$

$$\geq (1 - \varepsilon)\text{Cost}(X, \mathcal{C}_{\text{OPT}}) - \frac{\varepsilon}{100}\text{Cost}(N_X(c_i), \mathcal{C}_{\text{OPT}})$$

$$= \left(1 - \frac{101}{100}\varepsilon\right)\text{Cost}(N_X(c_i), \mathcal{C}_{\text{OPT}}).$$

Since for $i \in \mathcal{H}$, $|\mathrm{Cost}(N_X(c_i), \mathcal{C}_{\mathrm{OPT}}) - \mathrm{Cost}(N_S(c_i), \mathcal{C}_{\mathrm{OPT}})| \leq \varepsilon\mathrm{Cost}(N_X(c_i), \mathcal{C}_{\mathrm{OPT}})$ and $|\mathrm{Cost}(X, \mathcal{C}_{\mathrm{OPT}}) - \mathrm{Cost}(S, \mathcal{C}_{\mathrm{OPT}})| \leq \varepsilon\mathrm{Cost}(X, \mathcal{C}_{\mathrm{OPT}})$, we get

$$\sum_{i \in [s] \cap \mathcal{H}} \mathrm{Cost}(N_S(c_i), \mathcal{C}_{\mathrm{OPT}}) \geq (1-\varepsilon) \sum_{i \in [i] \cap \mathcal{H}} \mathrm{Cost}(N_X(c_i), \mathcal{C}_{\mathrm{OPT}})$$

$$\geq (1-\varepsilon)\left(1 - \frac{101}{100}\varepsilon\right)\mathrm{Cost}(X, \mathcal{C}_{\mathrm{OPT}})$$

$$\geq \left(1 - \frac{201}{100}\varepsilon\right)\mathrm{Cost}(X, \mathcal{C}_{\mathrm{OPT}}).$$

We also have
$$\mathrm{Cost}(S, \mathcal{C}_{\mathrm{OPT}}) \leq (1+\varepsilon)\mathrm{Cost}(X, \mathcal{C}_{\mathrm{OPT}}).$$
Since for $\varepsilon \in (0, \frac{1}{4}]$, $(1+\varepsilon)(1 - 4\varepsilon) \leq 1 - \frac{201}{100}\varepsilon$, we get

$$\sum_{i \in [s]} \mathrm{Cost}(N_S(c_i), \mathcal{C}_{\mathrm{OPT}}) \geq \sum_{i \in [s] \cap \mathcal{H}} \mathrm{Cost}(N_S(c_i), \mathcal{C}_{\mathrm{OPT}})$$

$$\geq (1 - 4\varepsilon)(1+\varepsilon)\mathrm{Cost}(X, \mathcal{C}_{\mathrm{OPT}})$$

$$\geq (1 - 3\varepsilon)\mathrm{Cost}(S, \mathcal{C}_{\mathrm{OPT}}).$$

Therefore $S$ is $(s, 1 - 3\varepsilon)$-skewed. $\qquad\square$

## C.2 $(1 + \varepsilon)$-APPROXIMATE CENTER NET

We will prove Lemma 3.3 in this section.

First, we prove that there always exists an $\varepsilon$-net in ball $B(x, r)$ with size $2^{\mathcal{O}(d \log(r/\varepsilon))}$.

**Lemma C.7.** *There exists an $\varepsilon$-net $\mathcal{N}$ in ball $B(x, r)$, such that $|\mathcal{N}| = 2^{\mathcal{O}(d \log(r/\varepsilon))}$.*

*Proof.* Notice that a $\frac{2\varepsilon}{\sqrt{d}}$-grid is an $\varepsilon$-net. In fact, let $\mathcal{N}$ be a $\frac{2\varepsilon}{\sqrt{d}}$-grid in $B(x, r)$. Then for any $y \in B(x, r)$,

$$\mathrm{dist}(y, \mathcal{N}) \leq \sqrt{\sum_{i=1}^{d} \left(\frac{\varepsilon}{\sqrt{d}}\right)^2} = \varepsilon.$$

Hence $\mathcal{N}$ has size with

$$|\mathcal{N}| = \left(\mathcal{O}(\frac{r\sqrt{d}}{2\varepsilon})\right)^d = 2^{\mathcal{O}(d \log(r/\varepsilon))}.$$

$\qquad\square$

Next, we demonstrate the following inequality to aid in constraining the cost distortion.

**Lemma C.8.** *Let $0 < |a| < b$, $a$ can be either positive or negative. Then*

$$|(b+a)^z - b^z| \leq 2^z|a|b^{z-1}.$$

*Proof.* For $a > 0$, since $0 < a < b$, we have $a^i b^{z-i} \leq ab^{z-1}$. Hence

$$|(b+a)^z - b^z| = (b+a)^z - b^z = \sum_{i=1}^{z} \binom{z}{i} a^i b^{z-i} \leq 2^z ab^{z-1}.$$

For $a < 0$, we get

$$|(b+a)^z - b^z| = (b + a - a)^z - (b+a)^z \leq 2^z|a|(b+a)^{z-1}.$$

Since $a < 0$, we have $b + a \leq b$. Hence

$$|(b+a)^z - b^z| \leq 2^z|a|b^{z-1}.$$

$\qquad\square$

For a center $c \in [\Delta]^d$, denote $d_c$ as the distance $\text{dist}(S, c)$. We establish the theorem by categorizing $c$ into three cases based on $d_c$. The cases are: $d_c = 0$, $d_c \geq 2^{M_1}$ where $M_1 = \log\left(\frac{\varepsilon}{kW}\right) - 2z$, and $0 < d_c < 2^{M_1}$. In the first case, we set $c' = c$. We show that $\text{Cost}(x, c) = \text{Cost}(x, c')$ in this scenario, implying zero cost distortion. In the second case, we choose $c'$ so that $\text{dist}(c, c') \leq \frac{\varepsilon d_c}{2^z}$. We show that $|\text{Cost}(x, c) - \text{Cost}(x, c')| \leq \frac{\varepsilon}{2}\text{Cost}(x, c)$, which results in a minor cost distortion. In the third case, we set $c'$ as the closest $x \in S$ to $c$. We prove that $|\text{Cost}(x, c) - \text{Cost}(x, c')| \leq \frac{\varepsilon}{2}\text{Cost}(x, c)$ for $x \neq c'$, resulting in a small distortion. Furthermore, we establish that for $\text{Cost}(x, c)$ where $x = c'$, it is relatively small relative to the total cost. Ultimately, we prove that our selection of $c'$ leads to a very minor distortion and provides a good approximation of $\mathcal{C}$. We demonstrate the validity of these three cases sequentially. Initially, for $d_c = 0$, selecting $c' = c$ does not result in cost distortion.

**Lemma C.9.** *For a center $c \in [\Delta]^d$, let $d_c$ be the distance $\text{dist}(S, c)$. Suppose $d_c = 0$. Then there exists $c' \in T$ such that $\text{Cost}(x, c) = \text{Cost}(x, c')$ for any $x \in S$.*

*Proof.* In fact, $d_c = 0$ means $c \in S$. Then we can just let $c' = c$, which leads $\text{Cost}(x, c) = \text{Cost}(x, c')$ for any $x \in S$. $\qquad\square$

Second, given $d_c \in [2^{M_1}, 2^{M_2+1})$, it is possible to select some $c' \in T$ and produce a minor distortion of the cost in comparison to the initial cost.

**Lemma C.10.** *For a center $c \in [\Delta]^d$, let $d_c$ be the distance $\text{dist}(S, c)$. Suppose $d_c \in [2^{M_1}, 2^{M_2+1})$, where $M_1 = \log\left(\frac{\varepsilon}{kW}\right) - 2z - 2$ and $M_2 = \log\Delta$. Then there exists $c' \in T$ such that $|\text{Cost}(x, c) - \text{Cost}(x, c')| \leq \frac{\varepsilon}{2}\text{Cost}(x, c)$ for any $x \in S$.*

*Proof.* Assume $d_c \in [2^i, 2^{i+1})$, where $i \in [M_1, M_2]$. Define $x_c$ as the point in $S$ closest to $c$. Given $d_c \in [2^i, 2^{i+1})$, it follows that $c \in B(x_c, 2^{i+1})$. Because $i \in [M_1, M_2]$, an $\frac{\varepsilon 2^{i+1}}{2^{2z+1}}$-net has been established in $B(x_c, 2^{i+1})$, and $T$ includes such a net. Consequently, there exists some $c' \in T$ such that $\text{dist}(c, c') \leq \frac{\varepsilon 2^i}{2^{2z}}$.

For any $x \in S$, let $D_1 = \max\{\text{dist}(x, c), \text{dist}(x, c')|\}$ and let $D_2 = \min\{\text{dist}(x, c), \text{dist}(x, c')|\}$. Then by Lemma C.8, we get

$$|\text{dist}(x, c)^z - \text{dist}(x, c')^z| = |D_1^z - D_2^z| \leq 2^z |D_1 - D_2| D_1^{z-1}.$$

By triangle inequality, we get

$$|D_1 - D_2| = |\text{dist}(x, c) - \text{dist}(x, c')| \leq \text{dist}(c, c') \leq \frac{\varepsilon 2^i}{2^{2z}}.$$

If $D_1 = \text{dist}(x, c')$, we have

$$D_1 = \text{dist}(x, c) + (\text{dist}(x, c') - \text{dist}(x, c)) \leq \text{dist}(x, c) + \frac{\varepsilon 2^i}{2^{2z}}.$$

Since $d_c \in [2^i, 2^{i+1})$, for any $x \in S$, $\text{dist}(x, c) \geq d_c \geq 2^i$. Since $\varepsilon \in (0, 1]$, we get $\frac{\varepsilon 2^{i+1}}{2^{3z}} \leq \text{dist}(x, c)$ for any $x \in S$. Then

$$D_1 \leq \text{dist}(x, c) + \text{dist}(x, c) = 2\text{dist}(x, c).$$

Hence

$$|\text{dist}(x, c)^z - \text{dist}(x, c')^z| \leq 2^z \frac{\varepsilon 2^i}{2^{2z}} \left(2 \cdot \text{dist}(x, c)\right)^{z-1} = \varepsilon 2^{i-1}\text{dist}(x, c)^{z-1}.$$

Since $2^i \leq \text{dist}(x, c)$, we get

$$|\text{dist}(x, c)^z - \text{dist}(x, c')^z| \leq \frac{\varepsilon}{2}\text{dist}(x, c)^z.$$

Since $\text{Cost}(x, c) = w(x) \cdot \text{dist}(x, c)^z$, we get

$$|\text{Cost}(x, c) - \text{Cost}(x, c')| = w(x) \cdot |\text{dist}(x, c)^z - \text{dist}(x, c')^z|$$
$$\leq w(x) \cdot \frac{\varepsilon}{2}\text{dist}(x, c)^z$$
$$= \frac{\varepsilon}{2}\text{Cost}(x, c).$$

$\qquad\square$

Third, for $d_c < 2^{M_1}$, we can select a certain $c' \in T$ and produce minimal distortion in cost relative to the initial cost for $x \neq c'$, and generate minor distortion in cost relative to the overall cost for $x = c'$.

**Lemma C.11.** *For a center $c \in [\Delta]^d$, let $d_c$ be the distance $dist(S, c)$. Suppose $0 < d_c < 2^{M_1}$, where $M_1 = \log\left(\frac{\varepsilon}{kW}\right) - 2z - 2$. Let $x_c$ be the point of $S$ nearest to $c$. Let $c' = x_c$, then $Cost(x_c, c) \leq \frac{\varepsilon}{2k} \cdot \frac{1}{2^z}$, and $|Cost(x, c) - Cost(x, c')| \leq \frac{\varepsilon}{2} Cost(x, c)$ for any $x \neq x_c \in S$. Furthermore, for any $x \in S$, $x \neq x_c$ is equivalent to $dist(x, c) \geq 2^{M_1}$.*

*Proof.* Since $W \geq 1$ and $\varepsilon \in (0, 1]$, $\log\left(\frac{\varepsilon}{kW}\right) \leq 0$. Hence $M_1 \leq -2$. Then $d_c < 2^{M_1} \leq \frac{1}{4}$. Since any $x \in S$ has integer coordinates and $dist(x_c, c) = d_c \leq \frac{1}{4}$, for any $x \neq x_c \in S$, $dist(x, c) \geq \frac{1}{2} \geq 2^{M_1 + 1}$. Also, if $dist(x, c) \geq 2^{M_1} > dist(x_c, c)$, we must have $x \neq x_c$. Hence for any $x \in S$, $x \neq x_c$ is equivalent to $dist(x, c) \geq 2^{M_1}$.

For $x \neq x_c \in S$, let $D_1 = \max\{dist(x, c), dist(x, c')|\}$ and let $D_2 = \min\{dist(x, c), dist(x, c')|\}$. Then by triangle inequality,

$$|dist(x, c) - dist(x, c')| \leq dist(c, c') = dist(x_c, c) = d_c < 2^{M_1}.$$

Then for $D_1$, we have

$$D_1 \leq dist(x, c) + |dist(x, c) - dist(x, c')| = dist(x, c) + d_c.$$

Since $dist(x, c) \geq \frac{1}{4} \geq 2^{M_1} > d_c$, we get

$$D_1 \leq dist(x, c) + dist(x, c) = 2dist(x, c).$$

Then similar to the proof of Lemma C.10, by Lemma C.8, we get

$$|dist(x, c)^z - dist(x, c')^z| = |D_1^z - D_2^z|$$
$$\leq 2^z |D_1 - D_2| D_1^{z-1}$$
$$\leq 2^z d_c \left(2dist(x, c)\right)^{z-1}.$$

Since $2^{M_1} > d_c$ and $M_1 = \log\left(\frac{\varepsilon}{kW}\right) - 2z - 2$, we get

$$|dist(x, c)^z - dist(x, c')^z| \leq 2^{2z-1} 2^{M_1} dist(x, c)^{z-1}$$
$$= 2^{-3} \frac{\varepsilon}{kW} dist(x, c)^{z-1}.$$

Since $dist(x, c) \geq \frac{1}{2}$, we get

$$|dist(x, c)^z - dist(x, c')^z| \leq 2^{-2} \frac{\varepsilon}{kW} dist(x, c)^z.$$

Hence

$$|Cost(x, c) - Cost(x, c')| = w(x) \cdot |dist(x, c)^z - dist(x, c')^z|$$
$$\leq w(x) 2^{-2} \frac{\varepsilon}{kW} dist(x, c)^z.$$

Since $k \geq 1$ and $W \geq 1$, we get

$$|Cost(x, c) - Cost(x, c')| \leq w(x) \frac{\varepsilon}{2} dist(x, c)^z = \frac{\varepsilon}{2} Cost(x, c).$$

For $x_c = c'$, since $dist(x_c, c) = d_c < 2^{M_1}$, we get

$$Cost(x_c, c) = w(x_c) dist(x_c, c)^z \leq w(x_c) \left(2^{M_1}\right)^z.$$

Since $2^{M_1} \leq \frac{1}{4} < 1$, $\left(2^{M_1}\right)^z \leq 2^{M_1}$. Hence

$$Cost(x_c, c) \leq w(x_c) 2^{M_1} = w(x_c) 2^{-2z-2} \frac{\varepsilon}{kW}.$$

Since $W \geq w(x_c)$, we get

$$Cost(x_c, c) \leq \frac{\varepsilon}{2k} \frac{1}{2^z}.$$

$\square$

Now we complete the proof of Lemma 3.3.

**Lemma C.12.** *Let $S$ be a weighted set whose maximum weight is at least $1$. For $\varepsilon \in (0, 1]$, the set $T$ returned by* CENTERNET *satisfies: for any $\mathcal{C} \subset [\Delta]^d$ and $|\mathcal{C}| \leq k$, there exists $\mathcal{C}' \subset T^k$ such that*

$$(1 - \varepsilon)Cost(S, \mathcal{C}) \leq Cost(S, \mathcal{C}') \leq (1 + \varepsilon)Cost(S, \mathcal{C}).$$

*Furthermore, $T$ has a size of $|T| = |S| \cdot 2^{\mathcal{O}(d \log \frac{1}{\varepsilon} \log \log(\frac{k\Delta}{\varepsilon}))}$.*

*Proof.* We first prove the accuracy claim in the theorem.

For any $\mathcal{C} = \{c_1, c_2, \cdots, c_k\} \subset [\Delta]^d$, we will construct $\mathcal{C}' \subset T$ such that

$$(1 - \varepsilon)\text{Cost}(S, \mathcal{C}) \leq \text{Cost}(S, \mathcal{C}') \leq (1 + \varepsilon)\text{Cost}(S, \mathcal{C}).$$

For any $c_i \in \mathcal{C}$, we select the corresponding $c_i' \in T$ the way we used in Lemma C.9, Lemma C.10, and Lemma C.11. Let $\mathcal{C}' = \{c_1', c_2', \cdots, c_k'\}$.

We partition $S$ into three subsets: $S_0$, $S_1$, and $S_2$. Here, $S_0$ comprises the points that coincide with $\mathcal{C}$. The set $S_1$ consists of points whose distance from $\mathcal{C}$ is less than $2^{M_1}$ but greater than $0$. Lastly, $S_2$ contains points with a distance from $\mathcal{C}$ greater than $2^{M_1}$.

Let

$$S_0 = \{x \in S : \text{dist}(x, \mathcal{C}) = 0\},$$
$$S_1 = \{x \in S : 0 < \text{dist}(x, \mathcal{C}) < 2^{M_1}\},$$
$$S_2 = \{x \in S : \text{dist}(x, \mathcal{C}) \geq 2^{M_1}\}.$$

We will analyze the distortion of cost of $S_0$, $S_1$, and $S_2$ one by one.

For $x \in S_0$, since $\text{dist}(x, \mathcal{C}) = 0$, there exists some $c_i \in \mathcal{C}$ such that $d_{x_i} = 0$. Then by Lemma C.9, we will select $c_i' = x$. Hence we get

$$\text{Cost}(x, \mathcal{C}') = \text{Cost}(x, \mathcal{C}) = 0.$$

Then

$$|\text{Cost}(S_0, \mathcal{C}') - \text{Cost}(S_0, \mathcal{C})| = |\sum_{x \in S_0} \text{Cost}(x, \mathcal{C}') - \text{Cost}(x, \mathcal{C})| = 0.$$

For $x \in S_1$, $0 < \text{dist}(x, \mathcal{C}) < 2^{M_1}$ means there exists some $c_i \in \mathcal{C}$ such that $\text{dist}(x, c_i) = d_{c_i} \in (0, 2^{M_1})$. By Lemma C.11, we will select $c_i' = x$, which means $\text{Cost}(x, \mathcal{C}') = 0$. Also, by Lemma C.11, we have

$$\text{Cost}(x, \mathcal{C}) \leq \frac{\varepsilon}{2k} \frac{1}{2^z}.$$

Observe that $\text{Cost}(S, \mathcal{C}) \geq \frac{1}{2^z}$. Given that $|S| > k$ and each point $x \in S$ has integer coordinates, there must be some center $c_i \in \mathcal{C}$ such that at least two distinct points $x_1 \neq x_2$ are assigned to $c_i$. Since $x_1 \neq x_2$, at least one of them is at least $\frac{1}{2}$ distance away from $c_i$, which results in a cost of at least $\frac{1}{2^z}$. Therefore

$$\text{Cost}(x, \mathcal{C}) \leq \frac{\varepsilon}{2k}\text{Cost}(S, \mathcal{C}).$$

Since $x \in S$ has integer coordinators, for any $c_i \in \mathcal{C}$, there exists at most one $x \in S$ such that $\text{dist}(x, c_i) < 2^{M_1}$. Hence $|S_1|$ is at most $k$. Then

$$|\text{Cost}(S_1, \mathcal{C}') - \text{Cost}(S_1, \mathcal{C})| = |\sum_{x \in S_1} \text{Cost}(x, \mathcal{C}') - \text{Cost}(x, \mathcal{C})|$$

$$\leq \sum_{x \in S_1} \frac{\varepsilon}{2k}\text{Cost}(S, \mathcal{C})$$

$$\leq \frac{\varepsilon}{2}\text{Cost}(S, \mathcal{C}).$$

For $x \in S_2$, since $\text{dist}(x, \mathcal{C}) \geq 2^{M_1}$, we have $\text{dist}(x, c_i) \geq 2^{M_1}$ for any $c_i \in \mathcal{C}$. For $c_i \in \mathcal{C}$ that $d_{c_i} \geq 2^{M_1}$, by Lemma C.10,

$$|\text{Cost}(x, c_i) - \text{Cost}(x, c_i')| \leq \frac{\varepsilon}{2}\text{Cost}(x, c_i).$$

For $c_i \in \mathcal{C}$ that $d_{c_i} < 2^{M_1}$, since $\text{dist}(x, c_i) \geq 2^{M_1}$, by Lemma C.11, we also have

$$|\text{Cost}(x, c_i) - \text{Cost}(x, c_i')| \leq \frac{\varepsilon}{2}\text{Cost}(x, c_i).$$

Hence $|\text{Cost}(x, c_i) - \text{Cost}(x, c_i')| \leq \frac{\varepsilon}{2}\text{Cost}(x, c_i)$ is true for any $c_i \in \mathcal{C}$. Then we can claim that

$$|\text{Cost}(x, \mathcal{C}) - \text{Cost}(x, \mathcal{C}')| \leq \frac{\varepsilon}{2}\text{Cost}(x, \mathcal{C})$$

for any $x \in S_2$.

Notice that the above claim is non-trivial because it is possible that $x$ is assigned to $c_i \in \mathcal{C}$, but is assigned to $c_j' \in \mathcal{C}'$ for $i \neq j$. We may assume that $x$ is assigned to $c_i \in \mathcal{C}$, and is assigned to $c_j' \in \mathcal{C}'$, where $i$ and $j$ can be either the same, or not the same. Since $x$ is assigned to $c_i \in \mathcal{C}$, and is assigned to $c_j' \in \mathcal{C}'$, we have $\text{Cost}(x, c_j) \geq \text{Cost}(x, c_i)$, and $\text{Cost}(x, c_i') \geq \text{Cost}(x, c_j')$. Hence

$$\text{Cost}(x, \mathcal{C}') = \text{Cost}(x, c_j') \geq (1 - \frac{\varepsilon}{2})\text{Cost}(x, c_j)$$
$$\geq (1 - \frac{\varepsilon}{2})\text{Cost}(x, c_i) = (1 - \frac{\varepsilon}{2})\text{Cost}(x, \mathcal{C}),$$

and

$$\text{Cost}(x, \mathcal{C}') = \text{Cost}(x, c_j') \leq \text{Cost}(x, c_i')$$
$$\leq (1 + \frac{\varepsilon}{2})\text{Cost}(x, c_i) = (1 + \frac{\varepsilon}{2})\text{Cost}(x, \mathcal{C}).$$

Hence we get

$$|\text{Cost}(x, \mathcal{C}') - \text{Cost}(x, \mathcal{C})| \leq \frac{\varepsilon}{2}\text{Cost}(x, \mathcal{C}),$$

for any $x \in S_2$. Then

$$|\text{Cost}(S_2, \mathcal{C}') - \text{Cost}(S_2, \mathcal{C})| = |\sum_{x \in S_2} \text{Cost}(x, \mathcal{C}') - \text{Cost}(x, \mathcal{C})|$$
$$\leq \sum_{x \in S_2} \frac{\varepsilon}{2}\text{Cost}(x, \mathcal{C}).$$

Since $S_2 \subset S$, we get

$$|\text{Cost}(S_2, \mathcal{C}') - \text{Cost}(S_2, \mathcal{C})| \leq \sum_{x \in S} \frac{\varepsilon}{2}\text{Cost}(x, \mathcal{C}) = \frac{\varepsilon}{2}\text{Cost}(S, \mathcal{C}).$$

Then combining the bound of $|\text{Cost}(S_i, \mathcal{C}') - \text{Cost}(S_i, \mathcal{C})|$, we get

$$|\text{Cost}(S, \mathcal{C}') - \text{Cost}(S, \mathcal{C})| = |\sum_{i=0}^{2} (\text{Cost}(S_i, \mathcal{C}') - \text{Cost}(S_i, \mathcal{C}))|$$
$$\leq 0 + \frac{\varepsilon}{2}\text{Cost}(S, \mathcal{C}) + \frac{\varepsilon}{2}\text{Cost}(S, \mathcal{C})$$
$$= \varepsilon\text{Cost}(S, \mathcal{C}).$$

Hence we complete our proof that $\mathcal{C}' \subset T$ gives an $(1 + \varepsilon)$-approximation for $\mathcal{C}$.

Subsequently, we shall demonstrate the assertion regarding the net size within the theorem.

By the CENTERNET$(S, \varepsilon, \Delta)$, we know

$$T = S \bigcup \left( \bigcup_{i=M_1}^{M_2} \bigcup_{x \in S} \mathcal{N}_{i,x} \right).$$

Since $\mathcal{N}_{i,x}$ is an $\frac{\varepsilon r}{2^{2z+1}}$-net in $B(x,r)$, by Lemma C.7,

$$|\mathcal{N}_{i,x}| = 2^{\mathcal{O}\left(d\log\left(r\cdot\frac{2^{2z+1}}{\varepsilon r}\right)\right)} = 2^{\mathcal{O}\left(d\log\left(\frac{1}{\varepsilon}\right)\right)}.$$

Hence

$$|T| \le |S| + (M_2 - M_1)\cdot|S|\cdot 2^{\mathcal{O}\left(d\log\left(\frac{1}{\varepsilon}\right)\right)}$$
$$= |S| + \left(\log\Delta - \log\left(\frac{\varepsilon}{kW}\right) + 2z + 2\right)\cdot|S|\cdot 2^{\mathcal{O}\left(d\log\left(\frac{1}{\varepsilon}\right)\right)}$$
$$= |S|2^{\mathcal{O}\left(d\log\left(\frac{1}{\varepsilon}\right)\log\log\left(\frac{kW\Delta}{\varepsilon}\right)\right)}.$$

By CORESETCONSTRUCTION$(X, \varepsilon, n, k, \Delta)$, we know that

$$W = \max_{x\in X}\{\frac{1}{\mu s(x)}\}.$$

Notice that $s(x) \ge \frac{1}{2n}$ for any $x \in X \subset [\Delta]^d$. In fact, we can select $\mathcal{C} = \{c_1, c_2, \cdots, c_k\}$ such that $\|c_i\| = 100\sqrt{d}\Delta$ for any $c_i \in \mathcal{C}$. By the definition of sensitivity,

$$s(x) = \max_{\mathcal{C}'\in(\mathbb{R}^d)^k}\frac{\mathrm{Cost}(x,\mathcal{C}')}{\mathrm{Cost}(X,\mathcal{C}')} \ge \frac{\mathrm{Cost}(x,\mathcal{C})}{\mathrm{Cost}(X,\mathcal{C})}.$$

Since $x \in [\Delta]^d$, we have $\mathrm{dist}(x, c_i) \in [99\sqrt{d}\Delta, 101\sqrt{d}\Delta]$. Hence

$$s(x) \ge \frac{99\sqrt{d}\Delta}{n\cdot 101\sqrt{d}\Delta} \ge \frac{1}{2n}.$$

Hence we have $W \le \frac{2n}{\mu}$. Then

$$|T| = |S|2^{\mathcal{O}\left(d\log\left(\frac{1}{\varepsilon}\right)\log\log\left(\frac{k\Delta}{\varepsilon}\right)\right)}.$$

$\square$

Currently, we have $(S, T)$ where $|S| = \tilde{\mathcal{O}}(\frac{dk^2}{\varepsilon})$ and $|T| = 2^{\tilde{\mathcal{O}}\left(d\log\left(\frac{dk}{\varepsilon}\right)\right)}$. According to Lemma 3.2 and Lemma 3.3, an optimal solution for $(S, T)$ is a $(1+2\varepsilon)$-approximate solution for $X$. Therefore, using a brute force search, we can achieve a $(1 + 2\varepsilon)$-approximation within a running time of $2^{\tilde{\mathcal{O}}\left(dk\log\left(\frac{dk}{\varepsilon}\right)\right)}$. Nevertheless, this is not a PTAS for $k$ since the running time depends on $2^{\mathcal{O}(k)}$. For heavily skewed datasets, the running time can be further optimized. In Appendix D and Appendix E, we will present a PTAS utilizing this heavily skewed property.

## D LOCAL SEARCH ADAPTED FOR HEAVILY SKEWED SET

We will prove Lemma 4.1 in Appendix D.1 and Lemma 4.2 in Appendix D.2.

### D.1 HEAVY SKEW LOCAL SEARCH FOR $k$-MEDIAN

For brevity, we will consider $S$ as the data set and $T$ as a finite set of potential centers, with $S$ being a $(s, 1 - \varepsilon)$-skewed data set. We denote $\mathcal{C} = \{c_1, c_2, \cdots, c_k\}$ as the optimal solution within the net $T$, and $\mathcal{P}$ as the heuristic solution produced by the algorithm. We assume $\mathrm{Cost}(N(c_i)) \ge \mathrm{Cost}(N(c_j))$ for $i \le j$, where $N(c_i) = \{x \in S : \mathrm{Cost}(x, c_i) \le \mathrm{Cost}(x, c_j), j \ne i\}$. We define $\mathcal{C}_E = \{c_1, c_2, \cdots, c_s\}$ as the expensive centers and $\mathcal{C}_C = \mathcal{C}\backslash\mathcal{C}_E$ as the cheap centers. For $U \subset \mathcal{C}$, let $N(\mathcal{U}) = \{x \in S : \mathrm{Cost}(x, \mathcal{U}) \le \mathrm{Cost}(x, \mathcal{C}\backslash\mathcal{U})\}$ denote the points assigned to $\mathcal{U}$ in the optimal solution, and let $N^*(\mathcal{U}) = \{x \in S : \mathrm{Cost}(x, \mathcal{U}) \le \mathrm{Cost}(x, \mathcal{P}\backslash\mathcal{U})\}$ for $\mathcal{U} \subset \mathcal{P}$, representing the points allocated to $\mathcal{U}$ in the heuristic solution $\mathcal{P}$. We also denote $O_x = \mathrm{dist}(x, \mathcal{C})$ and $A_x = \mathrm{dist}(x, \mathcal{P})$.

We will establishLemma 4.1 by demonstrating that HEAVYSKEWLOCALSEARCH successfully approximates $N(\mathcal{C}_C)$.

We will employ the general framework for the analysis of local search algorithms as previously utilized by Arya et al. (2001); Kanungo et al. (2002); Gupta & Tangwongsan (2008), but with a more nuanced analysis. Within this framework, we construct a series of swaps between the heuristic centers and the optimal centers. Given that the set of heuristic centers represents a local optimum, the cost will increase after each swap. Conversely, by swapping heuristic centers to optimal centers, we can bound the cost distortion as $\gamma_1 \sum O_x - \gamma_2 \sum A_x$ if the swapping centers are chosen with precision. Consequently, we can achieve $0 \leq \gamma_1 \sum O_x - \gamma_2 \sum A_x$ for certain swaps. Ultimately, by constructing multiple such swaps and aggregating these inequalities, we derive the desired result.

Before conducting further analysis, we first present some notations and definitions to facilitate the examination of the local search algorithm. We define an optimal center $c \in \mathcal{C}_C$ as being captured by a heuristic center $b \in \mathcal{B}$ if $b$ is the closest center to $c$ within $\mathcal{B}$. Ties are resolved arbitrarily to ensure that each $c \in \mathcal{C}_C$ is captured by exactly one heuristic center. We say that a heuristic center $b$ has a degree of $m$ if it captures exactly $m$ optimal centers in $\mathcal{C}_C$.

We define $b_c$ as the heuristic center in $\mathcal{B}$ closest to $c \in \mathcal{C}$, $b_x$ as the heuristic center in $\mathcal{B}$ closest to $x \in S$, $c_x$ as the optimal center in $\mathcal{C}$ closest to $x$, and $c'_x$ as the optimal center in $\mathcal{C}_C$ closest to $x$.

We will examine the interchange between the center sets $\mathcal{F}$ and $\mathcal{R}$. Initially, we establish that the distance between $x$ and the new centers can be constrained by $O_x$ and $A_x$, provided that $\mathcal{F}$ and $\mathcal{R}$ satisfy the following condition.

**Lemma D.1.** *Suppose $\mathcal{F} \subset \mathcal{C}_C$, $\mathcal{R} \subset \mathcal{B}$, and $|\mathcal{F}| = |\mathcal{R}|$. If the heuristic centers in $\mathcal{R}$ do not capture any optimal centers in $\mathcal{C}_C \backslash \mathcal{F}$, for $x \in (N^*(\mathcal{R}) \backslash N(\mathcal{F})) \cap N(\mathcal{C}_C)$,*

$$dist(x, \mathcal{P} \backslash \mathcal{R} \cup \mathcal{F}) \leq 2O_x + A_x.$$

*Proof.* Since $x \notin N(\mathcal{F})$ and $x \in N(\mathcal{C}_C)$, $c'_x \notin \mathcal{F}$. By the condition, the centers in $\mathcal{R}$ do not capture $c'_x$, so $b_{c'_x} \in \mathcal{B} \backslash \mathcal{R} \subset \mathcal{P} \backslash \mathcal{R} \cup \mathcal{F}$. Hence

$$\text{dist}(x, \mathcal{P} \backslash \mathcal{R} \cup \mathcal{F}) \leq \text{dist}(x, b_{c'_x}).$$

By triangle inequality,

$$\text{dist}(x, b_{c'_x}) \leq \text{dist}(x, c'_x) + \text{dist}(c'_x, b_{c'_x}).$$

Since $b_{c'_x}$ is the nearest center to $c'_x$, $\text{dist}(c'_x, b_{c'_x}) \leq \text{dist}(c'_x, b_x)$, which leads

$$\text{dist}(x, b_{c'_x}) \leq \text{dist}(x, c'_x) + \text{dist}(c'_x, b_x).$$

By triangle inequality,

$$\text{dist}(x, b_{c'_x}) \leq \text{dist}(x, c'_x) + \text{dist}(c'_x, x) + \text{dist}(x, b_x)$$
$$= 2\text{dist}(x, c'_x) + \text{dist}(x, b_x).$$

Since $O_x = \text{dist}(x, c'_x)$ and $A_x = \text{dist}(x, b_x)$, it leads

$$\text{dist}(x, \mathcal{P} \backslash \mathcal{R} \cup \mathcal{F}) \leq 2O_x + A_x.$$

$\square$

Next, we design a collection of partition pairs $\{(\mathcal{F}_i, \mathcal{R}_i)\}$ that satisfy the requirement that the centers within $\mathcal{R}_i$ do not capture any center beyond $\mathcal{F}_i$.

**Lemma D.2.** *Assume $\mathcal{B}$ is the heuristic center set and $\mathcal{C}_C$ is the cheap optimal center set. There exists partition pair $\{(\mathcal{F}_i, \mathcal{R}_i)\}_{i=1}^l$ that meets the following condition:*

- *$\{\mathcal{F}_i\}$ is a partition of $\mathcal{C}_C$. In other words, $\mathcal{F}_i$ are disjoint from each other, and $\mathcal{C}_C = \cup_{i=1}^l \mathcal{F}_i$.*

- *$\{\mathcal{R}_i\}$ is a partition of $\mathcal{B}$.*

- *$|\mathcal{F}_i| = |\mathcal{R}_i|$ for $i \in [l]$.*

- *Centers in $\mathcal{R}_i$ do not capture any center $c \notin \mathcal{F}_i$ for $i \in [l]$.*

*Proof.* Recall that the degree of a heuristic center $b$ is the number of optimal centers in $\mathcal{C}_C$ that is captured by $b$. Also, every $c \in \mathcal{C}_C$ is captured by exactly one heuristic center.

WOLG, we can denote $\mathcal{B}_{>0} = \{b_1, \cdots, b_l\}$ as the set of all the centers with positive degree, and $\mathcal{B}_0 = \{b_{l+1}, \cdots, b_{k-s}\}$ as the set of centers with degree zero.

For any $b_i \in \mathcal{B}_{>0}$, we construct $\mathcal{F}_i$ as the optimal centers in $\mathcal{C}_C$ captured by $b_i$. Since every center in $\mathcal{C}_C$ is captured by exactly 1 heuristic center by definition, $\{\mathcal{F}_i\}$ is a partition of $\mathcal{C}_C$.

We construct $\mathcal{R}_i$ as the union of $b_i$ and $\deg b_i - 1$ centers with degree zero. We put centers of $\mathcal{B}_0$ into $\mathcal{R}_i$ in such way that every center in $\mathcal{B}_0$ belongs to exactly one of $\{\mathcal{R}_i\}$. Such construction is valid by the following discussion:

Since $|\mathcal{F}_i| = \deg b_i$, it leads that $|\mathcal{C}_C| = \sum_{i=1}^l |\mathcal{F}_i| = \sum_{i=1}^l \deg b_i$. Since $|\mathcal{B}| = |\mathcal{C}_C| = k - s$ and $|\mathcal{B}_{>0}| = l$, it leads that $\sum_{i=1}^l (\deg b_i - 1) = |\mathcal{B}| - l = |\mathcal{B}| - |\mathcal{B}_{>0}|$. It means we need $|\mathcal{B}| - |\mathcal{B}_{>0}|$ zero degree centers for such construction. On the other hand, we have exact $|\mathcal{B}_0| = |\mathcal{B}| - |\mathcal{B}_{>0}|$ degree zero centers. Hence we can assign every zero degree center to exact one $\mathcal{R}_i$.

Since such construction of $\{\mathcal{R}_i\}$ is valid, by the construction, $\{\mathcal{R}_i\}$ is a partition of $\mathcal{B}$. Also, by the construction, $|\mathcal{R}_i| = \deg b_i = |\mathcal{F}_i|$.

We have proven the first three conditions. For the last one, notice that $b_i$ only captures the centers in $\mathcal{F}_i$, and every other centers in $\mathcal{R}_i$ has 0 degree, which means they capture no centers. Hence $\mathcal{R}_i$ do not capture any center $c \notin \mathcal{F}_i$. $\qquad\square$

We claim that any $t$-swapping holds the following inequality if $\mathcal{R}$ do not capture $\mathcal{F}$.

**Lemma D.3.** *Let $(\mathcal{F}, \mathcal{R})$ be a pairing that $|\mathcal{F}| = |\mathcal{R}| \le t$ and $\mathcal{R}$ don't capture $\mathcal{F}$, then*

$$0 \le \sum_{x \in N(\mathcal{F})} (O_x - A_x) + \sum_{N^*(\mathcal{R}) \cap N(\mathcal{C}_E)} (O_x - A_x) + \sum_{N^*(\mathcal{R}) \cap N(\mathcal{C}_C)} 2O_x.$$

*Proof.* Since $|\mathcal{F}| \le t$, the swapping between $\mathcal{F}$ and $\mathcal{R}$ is a $t$-swapping. Since $\mathcal{P}$ returned by HEAVYSKEWLOCALSEARCHis a local optimum for $t$-swapping, the total cost of $S$ can only increase, which means

$$0 \le \mathrm{Cost}(S, \mathcal{P}\backslash\mathcal{R} \cup \mathcal{F}) - \mathrm{Cost}(S, \mathcal{P}).$$

Now we analyze the bound of $\mathrm{Cost}(S, \mathcal{P}\backslash\mathcal{R} \cup \mathcal{F}) - \mathrm{Cost}(S, \mathcal{P})$. For the sake of brevity, we will denote $\Delta_{\mathcal{U}} = \mathrm{Cost}(\mathcal{U}, \mathcal{P}\backslash\mathcal{R} \cup \mathcal{F}) - \mathrm{Cost}(\mathcal{U}, \mathcal{P})$ for any $\mathcal{U} \subset S$ in this proof. We also denote $\Delta_x = \Delta_{\{x\}}$.

Notice that $\Delta_x$ can be positive only if $x \in N^*(\mathcal{R})$. Since for $x \notin N^*(\mathcal{R})$, the center in $\mathcal{P}$ nearest to $x$ still belongs to $\mathcal{P}\backslash\mathcal{R}\cup\mathcal{F}$, which means that the new cost of $x$ can only decrease. It means $\Delta_x \le 0$ for $x \notin N^*(\mathcal{R})$. By splitting $S$ into $N^*(\mathcal{R})$ and $S\backslash N^*(\mathcal{R})$, we can express $\Delta_S$ in the following method:

$$0 \le \Delta_S = \Delta_{N^*(\mathcal{R})} + \Delta_{S\backslash N^*(\mathcal{R})}.$$

Since $N(\mathcal{F})\backslash N^*(\mathcal{R}) \subset S\backslash N^*(\mathcal{R})$ and $\Delta_x \le 0$ for $x \notin N^*(\mathcal{R})$,

$$0 \le \Delta_{N^*(\mathcal{R})} + \Delta_{N(\mathcal{F})\backslash N^*(\mathcal{R})}.$$

By splitting $N^*(\mathcal{R})$ into $N^*(\mathcal{R}) \cap N(\mathcal{F})$ and $N^*(\mathcal{R})\backslash N(\mathcal{F})$, we get

$$0 \le \Delta_{N^*(\mathcal{R})\backslash N(\mathcal{F})} + \Delta_{N^*(\mathcal{R})\cap N(\mathcal{F})} + \Delta_{N(\mathcal{F})\backslash N^*(\mathcal{R})}$$
$$= \Delta_{N^*(\mathcal{R})\backslash N(\mathcal{F})} + \Delta_{N(\mathcal{F})}.$$

For $x \in N(\mathcal{F})$, $\mathrm{Cost}(x, \mathcal{P}\backslash\mathcal{R} \cup \mathcal{F}) \le O_x$ because $c_x \in \mathcal{F} \subset \mathcal{P}\backslash\mathcal{R} \cup \mathcal{F}$. Hence $\Delta_x \le O_x - A_x$. Adding up all $x \in N(\mathcal{F})$, then

$$\Delta_{N(\mathcal{F})} \le \sum_{x \in N(\mathcal{F})} (O_x - A_x).$$

For $x \in N^*(\mathcal{R}) \backslash N(\mathcal{F})$, we split $N^*(\mathcal{R}) \backslash N(\mathcal{F})$ into $(N^*(\mathcal{R}) \backslash N(\mathcal{F})) \cap N(\mathcal{C}_E)$ and $(N^*(\mathcal{R}) \backslash N(\mathcal{F})) \cap N(\mathcal{C}_C)$.

For $x \in (N^*(\mathcal{R}) \backslash N(\mathcal{F})) \cap N(\mathcal{C}_E)$, we claim that $\text{Cost}(x, \mathcal{P} \backslash \mathcal{R} \cup \mathcal{F}) \le O_x$. In fact, $c_x \in \mathcal{C}_E$ because $x \in N(\mathcal{C}_E)$. By the HEAVYSKEWLOCALSEARCH, $\mathcal{P} = \mathcal{C}_E \cup \mathcal{B}$, which means $c_x \in \mathcal{P}$. On the other hand, since $\mathcal{R} \subset \mathcal{B} = \mathcal{P} \backslash \mathcal{C}_E$, $\mathcal{R}$ does not contain any center of $\mathcal{C}_E$. Since $c_x \in \mathcal{P}$ and we do not remove it after swapping, $c_x$ is still contained in $\mathcal{P} \backslash \mathcal{R} \cup \mathcal{F}$. Hence $\text{Cost}(x, \mathcal{P} \backslash \mathcal{R} \cup \mathcal{F}) \le O_x$.

Since $\mathcal{F} \subset \mathcal{C}_C$, $N(\mathcal{F})$ is disjoint from $N(\mathcal{C}_E)$. Hence $(N^*(\mathcal{R}) \backslash N(\mathcal{F})) \cap N(\mathcal{C}_E) = N^*(\mathcal{R}) \cap N(\mathcal{C}_E)$. It means

$$\Delta_{(N^*(\mathcal{R}) \backslash N(\mathcal{F})) \cap N(\mathcal{C}_E)} = \Delta_{N^*(\mathcal{R}) \cap N(\mathcal{C}_E)}.$$

Summing over all $x \in N^*(\mathcal{R}) \cap N(\mathcal{C}_E)$, we get

$$\Delta_{N^*(\mathcal{R}) \cap N(\mathcal{C}_E)} \le \sum_{x \in N^*(\mathcal{R}) \cap N(\mathcal{C}_E)} (O_x - A_x).$$

Hence

$$\Delta_{(N^*(\mathcal{R}) \backslash N(\mathcal{F})) \cap N(\mathcal{C}_E)} \le \sum_{x \in N^*(\mathcal{R}) \cap N(\mathcal{C}_E)} (O_x - A_x).$$

For $x \in (N^*(\mathcal{R}) \backslash N(\mathcal{F})) \cap N(\mathcal{C}_C)$, we can apply Lemma D.1 because $\mathcal{R}$ do not capture any optimal centers in $\mathcal{C}_C \backslash \mathcal{F}$. Hence

$$\Delta_x = \text{dist}(x, \mathcal{P} \backslash \mathcal{R} \cup \mathcal{F}) - A_x \le (2O_x + A_x) - A_x = 2O_x.$$

Summing over all $x \in (N^*(\mathcal{R}) \backslash N(\mathcal{F})) \cap N(\mathcal{C}_C)$, we get

$$\Delta_{(N^*(\mathcal{R}) \backslash N(\mathcal{F})) \cap N(\mathcal{C}_C)} \le \sum_{x \in (N^*(\mathcal{R}) \backslash N(\mathcal{F})) \cap N(\mathcal{C}_C)} 2O_x.$$

Since $O_x \ge 0$,

$$\sum_{x \in (N^*(\mathcal{R}) \backslash N(\mathcal{F})) \cap N(\mathcal{C}_C)} 2O_x \le \sum_{x \in N^*(\mathcal{R}) \cap N(\mathcal{C}_C)} 2O_x.$$

Hence

$$\Delta_{(N^*(\mathcal{R}) \backslash N(\mathcal{F})) \cap N(\mathcal{C}_C)} \le \sum_{x \in N^*(\mathcal{R}) \cap N(\mathcal{C}_C)} 2O_x.$$

Combining all the inequalities above, we get

$$0 \le \Delta_{N(\mathcal{F})} + \Delta_{N^*(\mathcal{R}) \backslash N(\mathcal{F})}$$
$$= \Delta_{N(\mathcal{F})} + \Delta_{(N^*(\mathcal{R}) \backslash N(\mathcal{F})) \cap N(\mathcal{C}_E)} + \Delta_{(N^*(\mathcal{R}) \backslash N(\mathcal{F})) \cap N(\mathcal{C}_C)}$$
$$\le \sum_{x \in N(\mathcal{F})} (O_x - A_x) + \sum_{N^*(\mathcal{R}) \cap N(\mathcal{C}_E)} (O_x - A_x) + \sum_{N^*(\mathcal{R}) \cap N(\mathcal{C}_C)} 2O_x.$$

$\square$

The previous lemma only holds for $t$-swapping, in other words, $|\mathcal{F}| = |\mathcal{R}| \le t$. We also claim the following inequality for the case $|\mathcal{F}| = |\mathcal{R}| > t$.

**Lemma D.4.** *If $|\mathcal{F}| = |\mathcal{R}| > t$, $\mathcal{R}$ has exactly one positive degree center, and $\mathcal{R}$ do not capture any center outside $\mathcal{F}$, the following inequality holds:*

$$0 \le \sum_{x \in N(\mathcal{F})} (O_x - A_x) + \left(1 + \frac{1}{t}\right) \left( \sum_{N^*(\mathcal{R}) \cap N(\mathcal{C}_E)} (O_x - A_x) + \sum_{N^*(\mathcal{R}) \cap N(\mathcal{C}_C)} 2O_x \right).$$

*Proof.* Since $\mathcal{R}$ has exactly one positive degree center, we just denote it as $b$. Consider a swap $(c, b') \in \mathcal{F} \times (\mathcal{R} \backslash \{b\})$. Since $b' \in \mathcal{R} \backslash \{b\}$, it is a zero degree center, which means it captures no centers. Also, $|\{c\}| = |\{b'\}| = 1 \le t$. It means the swapping pair meets the condition of Lemma D.3, which leads

$$0 \le \sum_{x \in N(c)} (O_x - A_x) + \sum_{N^*(b') \cap N(\mathcal{C}_E)} (O_x - A_x) + \sum_{N^*(b') \cap N(\mathcal{C}_C)} 2O_x.$$

Consider all the possible combination of $(c, b') \in \mathcal{F} \times (\mathcal{R} \backslash \{b\})$. Denote $|\mathcal{F}| = m$. There are $m(m-1)$ such pairs. Every center $c \in \mathcal{F}$ appears exactly $m-1$ times in these pairs, and every center $b' \in \mathcal{R} \backslash \{b\}$ appears exactly $m$ times. Every pair corresponds to one such inequality. We add all these inequalities together, and get

$$
\begin{aligned}
0 \leq & (m-1) \sum_{x \in N(\mathcal{F})} (O_x - A_x) + m \cdot \sum_{N^*(\mathcal{R}) \cap N(\mathcal{C}_E)} (O_x - A_x) \\
& + m \cdot \sum_{N^*(\mathcal{R}) \cap N(\mathcal{C}_C)} 2O_x,
\end{aligned}
$$

which is equivalent to

$$
0 \leq \sum_{x \in N(\mathcal{F})} (O_x - A_x) + \gamma \left( \sum_{N^*(\mathcal{R}) \cap N(\mathcal{C}_E)} (O_x - A_x) + \sum_{N^*(\mathcal{R}) \cap N(\mathcal{C}_C)} 2O_x \right),
$$

where $\gamma = \frac{m}{m-1}$.

Since $|\mathcal{F}| = m > t$,

$$
t = 1 + \frac{1}{m-1} \geq 1 + \frac{1}{t}.
$$

On the other hand, we demonstrated in the proof of Lemma D.3 that the second and third terms in the above inequality are non-negative. Therefore, substituting $\gamma = \frac{m}{m-1}$ with $1 + \frac{1}{t}$ does not diminish the right-hand side, leading to the desired result. $\qquad \square$

Now we have:

**Lemma D.5.**

$$
\sum_{x \in N(\mathcal{C}_C)} A_x \leq \left( 3 + \frac{2}{t} \right) \sum_{x \in N(\mathcal{C}_C)} O_x + \left( 1 + \frac{1}{t} \right) \sum_{x \in N(\mathcal{C}_E)} (O_x - A_x).
$$

*Proof.* According to Lemma D.2, there is a partition pair $\{(\mathcal{F}_i, \mathcal{R}_i)\}_{i=1}^l$ that satisfies the four conditions specified. For any pair $(\mathcal{F}_i, \mathcal{R}_i)$ within this set, if $|\mathcal{F}_i| \leq t$, Lemma D.3 can be utilized, which results

$$
0 \leq \sum_{x \in N(\mathcal{F}_i)} (O_x - A_x) + \sum_{N^*(\mathcal{R}_i) \cap N(\mathcal{C}_E)} (O_x - A_x) + \sum_{N^*(\mathcal{R}_i) \cap N(\mathcal{C}_C)} 2O_x.
$$

Since we have shown the second and third term is non-negative,

$$
0 \leq \sum_{x \in N(\mathcal{F}_i)} (O_x - A_x) + \gamma_t \left( \sum_{N^*(\mathcal{R}_i) \cap N(\mathcal{C}_E)} (O_x - A_x) + \sum_{N^*(\mathcal{R}_i) \cap N(\mathcal{C}_C)} 2O_x \right),
$$

where $\gamma_t = 1 + \frac{1}{t}$.

For any pair $(\mathcal{F}_i, \mathcal{R}_i)$ that $|\mathcal{F}_i| > t$, we can apply Lemma D.4 and get

$$
0 \leq \sum_{x \in N(\mathcal{F}_i)} (O_x - A_x) + \gamma_t \left( \sum_{N^*(\mathcal{R}_i) \cap N(\mathcal{C}_E)} (O_x - A_x) + \sum_{N^*(\mathcal{R}_i) \cap N(\mathcal{C}_C)} 2O_x \right).
$$

Each pair corresponds to an analogous inequality. Summing these inequalities from $(\mathcal{F}_1, \mathcal{R}_1)$ to $(\mathcal{F}_l, \mathcal{R}_l)$, and considering that every optimal center in $\mathcal{C}_C$ and every heuristic center in $\mathcal{B}$ appears exactly once, we obtain

$$
0 \leq \sum_{x \in N(\mathcal{C}_C)} (O_x - A_x) + \gamma_t \left( \sum_{N^*(\mathcal{B}) \cap N(\mathcal{C}_E)} (O_x - A_x) + \sum_{N^*(\mathcal{B}) \cap N(\mathcal{C}_C)} 2O_x \right).
$$

We have shown that $O_x - A_x \geq 0$ for $x \in N(\mathcal{C}_E)$ in the proof of Lemma D.3. Hence

$$\sum_{N^*(\mathcal{B}) \cap N(\mathcal{C}_E)} (O_x - A_x) \leq \sum_{N(\mathcal{C}_E)} (O_x - A_x).$$

Since $O_x$ is non-negative,

$$\sum_{N^*(\mathcal{B}) \cap N(\mathcal{C}_C)} 2O_x \leq \sum_{N(\mathcal{C}_C)} 2O_x$$

Thus

$$0 \leq \sum_{x \in N(\mathcal{C}_C)} (O_x - A_x) + \gamma_t \left( \sum_{N(\mathcal{C}_E)} (O_x - A_x) + \sum_{N(\mathcal{C}_C)} 2O_x \right),$$

where $\gamma_t = 1 + \frac{1}{t}$.

Simplifying the above inequality, we get

$$\sum_{x \in N(\mathcal{C}_C)} A_x \leq \sum_{x \in N(\mathcal{C}_C)} O_x + (1 + \frac{1}{t}) \left( \sum_{N(\mathcal{C}_E)} (O_x - A_x) + \sum_{N(\mathcal{C}_C)} 2O_x \right)$$

$$= \left( 3 + \frac{2}{t} \right) \sum_{x \in N(\mathcal{C}_C)} O_x + \left( 1 + \frac{1}{t} \right) \sum_{x \in N(\mathcal{C}_E)} (O_x - A_x).$$

$\square$

In conclusion, we demonstrate Lemma 4.1.

**Lemma D.6.** *Let $S$ be an $(s, 1 - \varepsilon)$-skewed dataset, $T$ be the potential center set, and $\mathcal{A} = \mathcal{C}_E$, which is the set of centers of the $s$ most high-cost clusters in optimal solution. There exists a constant $\gamma > 1$, such that for any $\varepsilon \in (0, \frac{1}{2}]$, HEAVYSKEWLOCALSEARCH returns a $(1 + \varepsilon)$-approximation $\mathcal{P}$ for the $(k, 1)$-clustering for $S$ and $T$.*

*Proof.* By Lemma 2.3, there exists $\gamma > 0$ such that for $s > \gamma \left( \frac{1}{\varepsilon} \right)^{\frac{1}{p-1}}$, $\text{Cost}(N(\mathcal{C}_C), \mathcal{C}) \leq \frac{\varepsilon}{100} \text{Cost}(S, \mathcal{C})$.

There also exists $\gamma > 0$ such that for $t > \frac{\gamma}{\epsilon}$, $\frac{1}{t} \leq \frac{\varepsilon}{100}$.

By Lemma D.5,

$$\text{Cost}(S, \mathcal{P}) = \sum_{x \in S} A_x = \sum_{x \in N(\mathcal{C}_C)} A_x + \sum_{x \in N(\mathcal{C}_E)} A_x$$

$$\leq \left( 3 + \frac{2}{t} \right) \sum_{x \in N(\mathcal{C}_C)} O_x + \left( 1 + \frac{1}{t} \right) \sum_{x \in N(\mathcal{C}_E)} (O_x - A_x) + \sum_{x \in N(\mathcal{C}_E)} A_x$$

$$= \left( 3 + \frac{2}{t} \right) \sum_{x \in N(\mathcal{C}_C)} O_x + \sum_{x \in N(\mathcal{C}_E)} \left( \left( 1 + \frac{1}{t} \right) O_x - \frac{1}{t} A_x \right).$$

Since $A_x \geq 0$,

$$\text{Cost}(S, \mathcal{P}) \leq (3 + \frac{2}{t}) \sum_{x \in N(\mathcal{C}_C)} O_x + \left( 1 + \frac{1}{t} \right) \sum_{x \in N(\mathcal{C}_E)} O_x.$$

Since $\frac{1}{t} \leq \frac{\varepsilon}{100} \leq \frac{1}{100}$,

$$\text{Cost}(S, \mathcal{P}) \leq 5 \sum_{x \in N(\mathcal{C}_C)} O_x + \left( 1 + \frac{\varepsilon}{100} \right) \sum_{x \in N(\mathcal{C}_E)} O_x$$

$$= 5\text{Cost}(N(\mathcal{C}_C), \mathcal{C}) + \left( 1 + \frac{\varepsilon}{100} \right) \text{Cost}(N(\mathcal{C}_E), \mathcal{C}).$$

Since $\text{Cost}(N(\mathcal{C}_C), \mathcal{C}) \leq \frac{\varepsilon}{100} \text{Cost}(S, \mathcal{C})$ and $\text{Cost}(N(\mathcal{C}_E), \mathcal{C}) \leq \text{Cost}(S, \mathcal{P})$,

$$\text{Cost}(S, \mathcal{P}) \leq \frac{\varepsilon}{20} \text{Cost}(S, \mathcal{C}) + \left(1 + \frac{\varepsilon}{100}\right) \text{Cost}(S, \mathcal{C})$$
$$\leq \left(1 + \frac{\varepsilon}{10}\right) \text{Cost}(S, \mathcal{C}).$$

Hence we complete our proof. $\qquad\square$

### D.2   Heavy skew local search for $(k, z)$-clustering

Our guarantee of the $1 + \varepsilon$-approximation can also generate to general $(k, z)$-clustering. The framework is the same, but the cost function for the $(k, z)$-clustering is $\text{dist}(x, c)^z$ rather than $\text{dist}(x, c)$ for the $k$-median case. The difference causes the cost function to lose its additivity, which requires a more subtle analysis for the distortion of cost. Fortunately, despite the loss of additivity, with the help of a generalized triangle inequality and stricter chosen parameters, an $1 + \varepsilon$-approximation is still guaranteed.

For the sake of brevity, let us consider $S$ to be a $(s, 1 - \varepsilon^{z+1})$-skewed data set. The assumptions and notations for $T, \mathcal{P}, \mathcal{C}, \mathcal{C}_E, \mathcal{C}_C, N(c_i), N(\mathcal{U}), N^*(\mathcal{U}), O_x$, and $A_x$ remain identical to those in Appendix D.1.

Observe that for the $k$-median problem, we require that $S$ be $(s, 1 - \varepsilon)$-skewed, whereas for general $(k, z)$-clustering, we stipulate that $S$ be $(s, 1 - \varepsilon^{z+1})$-skewed. This implies a greater degree of skewness is necessary for general $(k, z)$-clustering to offset the loss of additivity.

We first introduce the generalized triangle inequality by Sohler & Woodruff (2018).

**Lemma D.7** (Claim 5 in (Sohler & Woodruff, 2018)). *Suppose $z \geq 1$, $x, y \geq 0$, and $\varepsilon \in (0, 1]$. Then*

$$(x + y)^z \leq (1 + \varepsilon) \cdot x^z + \left(1 + \frac{2z}{\varepsilon}\right)^z \cdot y^z.$$

Recall that $O_x = \text{dist}(x, \mathcal{C})$ and $A_x = \text{dist}(x, \mathcal{P})$, thus our cost function in the $(k, z)$-clustering scenario becomes $\text{Cost}(x, \mathcal{C}) = O_x^z$ and $\text{Cost}(x, \mathcal{P}) = A_x^z$.

Notice that Lemma D.1 still holds for $(k, z)$-clustering, because it only analyzes the distance in its proof. $(k, z)$-clustering only has a different cost function from $k$-median, so it will not affect the validity of Lemma D.1. Notice that Lemma D.2 also holds because its analysis does not depend on cost function.

However, Lemma D.3 and Lemma D.4 no longer holds because we use the fact that $0 \leq \text{Cost}(S, \mathcal{P} \backslash \mathcal{R} \cup \mathcal{F}) - \text{Cost}(S, \mathcal{P})$ for a $t$-swapping. We will give the adapted version of these two lemmas in the $(k, z)$-clustering case.

For the sake of brevity, we denote $\mathcal{U} = N^*(\mathcal{R}) \cap N(\mathcal{C}_E)$ and $\mathcal{V} = N^*(\mathcal{R}) \cap N(\mathcal{C}_C)$. $\Delta_x$ is still the distortion of cost as we used in the previous subsection.

**Lemma D.8.** *Let $(\mathcal{F}, \mathcal{R})$ be a pairing that $|\mathcal{F}| = |\mathcal{R}| \leq t$ and $\mathcal{R}$ do not capture $\mathcal{F}$. For $\varepsilon \in (0, \frac{1}{2}]$,*

$$0 \leq \sum_{x \in N(\mathcal{F})} (O_x^z - A_x^z) + \sum_{\mathcal{U}} (O_x^z - A_x^z) + \sum_{\mathcal{V}} \left(\frac{\xi}{\varepsilon^z} O_x^z + \frac{\varepsilon}{100} A_x^z\right),$$

*where $\xi$ is a constant.*

*Proof.* It is still true in $(k, z)$-clustering that

$$0 \leq \text{Cost}(S, \mathcal{P} \backslash \mathcal{R} \cup \mathcal{F}) - \text{Cost}(S, \mathcal{P})$$

and

$$\text{Cost}(S, \mathcal{P} \backslash \mathcal{R} \cup \mathcal{F}) - \text{Cost}(S, \mathcal{P}) \leq \Delta_{N(\mathcal{F})} + \Delta_{\mathcal{U} \backslash N(\mathcal{F})} + \Delta_{\mathcal{V} \backslash N(\mathcal{F})}.$$

However, we need a new bound for $\Delta_{N(\mathcal{F})}$, $\Delta_{\mathcal{U} \backslash N(\mathcal{F})}$ and $\Delta_{\mathcal{V} \backslash N(\mathcal{F})}$ this time.

For $x \in N(\mathcal{F})$, $c_x \in \mathcal{P} \backslash \mathcal{R} \cup \mathcal{F}$, so $\text{Cost}(x, \mathcal{P} \backslash \mathcal{R} \cup \mathcal{F}) \leq O_x$. Hence

$$\Delta_{N(\mathcal{F})} = \sum_{x \in N(\mathcal{F})} \left( \text{Cost}(x, \mathcal{P} \backslash \mathcal{R} \cup \mathcal{F}) - \text{Cost}(x, \mathcal{P}) \right)$$

$$\leq \sum_{x \in N(\mathcal{F})} (O_x^z - A_x^z).$$

For $x \in \mathcal{U} \backslash N(\mathcal{F})$, $c_x \in \mathcal{C}_E \subset \mathcal{P} \backslash \mathcal{R} \cup \mathcal{F}$, so $\text{Cost}(x, \mathcal{P} \backslash \mathcal{R} \cup \mathcal{F}) \leq O_x$. Hence

$$\Delta_{\mathcal{U} \backslash N(\mathcal{F})} = \sum_{x \in \mathcal{U} \backslash N(\mathcal{F})} \left( \text{Cost}(x, \mathcal{P} \backslash \mathcal{R} \cup \mathcal{F}) - \text{Cost}(x, \mathcal{P}) \right)$$

$$\leq \sum_{x \in \mathcal{U} \backslash N(\mathcal{F})} (O_x^z - A_x^z).$$

Since $c_x \in \mathcal{P}$, $A_x \leq O_x$. Thus we further get

$$\Delta_{\mathcal{U} \backslash N(\mathcal{F})} \leq \sum_{x \in \mathcal{U}} (O_x^z - A_x^z).$$

For $x \in \mathcal{V} \backslash N(\mathcal{F})$, by Lemma D.1,

$$\text{dist}(x, \mathcal{P} \backslash \mathcal{R} \cup \mathcal{F}) \leq 2O_x + A_x.$$

Hence

$$\Delta_{\mathcal{V} \backslash N(\mathcal{F})} = \sum_{x \in \mathcal{V} \backslash N(\mathcal{F})} \left( \text{Cost}(x, \mathcal{P} \backslash \mathcal{R} \cup \mathcal{F}) - \text{Cost}(x, \mathcal{P}) \right)$$

$$\leq \sum_{x \in \mathcal{V} \backslash N(\mathcal{F})} \left( (2O_x + A_x)^z - A_x^z \right).$$

Then

$$\Delta_{\mathcal{V} \backslash N(\mathcal{F})} = \sum_{x \in \mathcal{V} \backslash N(\mathcal{F})} \left( \text{Cost}(x, \mathcal{P} \backslash \mathcal{R} \cup \mathcal{F}) - \text{Cost}(x, \mathcal{P}) \right)$$

$$\leq \sum_{x \in \mathcal{V} \backslash N(\mathcal{F})} \left( (2O_x + A_x)^z - A_x^z \right).$$

Since $\left( (2O_x + A_x)^z - A_x^z \right) \geq 0$, we get

$$\Delta_{\mathcal{V} \backslash N(\mathcal{F})} \leq \sum_{x \in \mathcal{V}} \left( (2O_x + A_x)^z - A_x^z \right).$$

Since $\varepsilon \in (0, \frac{1}{2}]$, by Lemma D.7,

$$(2O_x + A_x)^z \leq \left( 1 + \frac{\varepsilon}{100} \right) A_x^z + \left( 1 + \frac{200z}{\varepsilon} \right)^z \cdot (2O_x)^z$$

$$\leq \left( 1 + \frac{\varepsilon}{100} \right) A_x^z + \frac{\xi}{\varepsilon^z} O_x^z.$$

Hence

$$\Delta_{\mathcal{V} \backslash N(\mathcal{F})} \leq \sum_{\mathcal{V}} \left( \frac{\xi}{\varepsilon^z} O_x^z + \frac{\varepsilon}{100} A_x^z \right).$$

Summing the above result and we get

$$0 \leq \sum_{x \in N(\mathcal{F})} (O_x^z - A_x^z) + \sum_{\mathcal{U}} (O_x^z - A_x^z) + \sum_{\mathcal{V}} \left( \frac{\xi}{\varepsilon^z} O_x^z + \frac{\varepsilon}{100} A_x^z \right).$$

$\square$

**Lemma D.9.** *If $|\mathcal{F}| = |\mathcal{R}| > t$, $\mathcal{R}$ has exactly one positive degree center, and $\mathcal{R}$ don't capture any center outside $\mathcal{F}$, for $\varepsilon \in (0, \frac{1}{2}]$, the following inequality holds:*

$$0 \le \sum_{x \in N(\mathcal{F})} (O_x^z - A_x^z) + \gamma_t \left( \sum_{\mathcal{U}} (O_x^z - A_x^z) + \sum_{\mathcal{V}} \left( \frac{\xi}{\varepsilon^z} O_x^z + \frac{\varepsilon}{100} A_x^z \right) \right),$$

*where $\gamma_t = 1 + \frac{1}{t}$.*

*Proof.* The proof is just a repetition of the proof of Lemma D.4. The only difference is that we substitute Lemma D.3 with Lemma D.8. $\square$

**Lemma D.10.** *For $\varepsilon \in (0, \frac{1}{2}]$, there exists $\xi' > 0$ such that*

$$\sum_{x \in N(\mathcal{C}_C)} A_x^z \le \left( 1 + \frac{\varepsilon}{50} \right) \cdot \left( \frac{\gamma_t \xi'}{\varepsilon^z} \sum_{x \in N(\mathcal{C}_C)} O_x^z + \gamma_t \sum_{x \in N(\mathcal{C}_E)} (O_x^z - A_x^z) \right),$$

*where $\gamma_t = 1 + \frac{1}{t}$.*

*Proof.* We repeat the proof of Lemma D.5, but substitute Lemma D.3 and Lemma D.4 with Lemma D.8 and Lemma D.9. We get

$$0 \le \sum_{x \in N(\mathcal{F})} (O_x^z - A_x^z) + \gamma_t \left( \sum_{N(\mathcal{C}_E)} (O_x^z - A_x^z) + \sum_{N(\mathcal{C}_C)} \left( \frac{\xi}{\varepsilon^z} O_x^z + \frac{\varepsilon}{100} A_x^z \right) \right),$$

where $\gamma_t = 1 + \frac{1}{t}$.

Since $\varepsilon \in (0, \frac{1}{2}]$ and $\gamma_t \ge 1$, there exists $\xi' > 0$ such that $\frac{\gamma_t \xi'}{\varepsilon^z} \ge 1 + \frac{\gamma_t \xi}{\varepsilon^z}$. Simplifying the above inequality, we get

$$\left( 1 - \frac{\varepsilon}{100} \right) \sum_{x \in N(\mathcal{C}_C)} A_x^z \le \frac{\gamma_t \xi'}{\varepsilon^z} \sum_{x \in N(\mathcal{C}_C)} O_x^z + \gamma_t \sum_{x \in N(\mathcal{C}_E)} (O_x^z - A_x^z),$$

where $\gamma_t = 1 + \frac{1}{t}$.

Since $\varepsilon \in (0, \frac{1}{2}]$,

$$\left( 1 - \frac{\varepsilon}{100} \right)^{-1} = 1 + \frac{\varepsilon}{100 - \varepsilon} \le 1 + \frac{\varepsilon}{50}.$$

Hence we complete the proof. $\square$

Finally, we will demonstrate Lemma 4.2.

*Proof.* By Lemma 2.3, there exists $\gamma > 0$ such that for $s > \gamma \left( \frac{z}{\varepsilon} \right)^{\frac{1}{p-1}}$, $\mathrm{Cost}(N(\mathcal{C}_C), \mathcal{C}) \le \frac{\varepsilon^{z+1}}{100\xi'} \mathrm{Cost}(S, \mathcal{C})$.

There also exists $\gamma > 0$ such that for $t > \frac{\gamma}{\epsilon}$, $\frac{1}{t} \le \frac{\varepsilon}{100}$.

By Lemma D.10,

$$\mathrm{Cost}(S, \mathcal{P}) = \sum_{x \in N(\mathcal{C}_C)} A_x^z + \sum_{x \in N(\mathcal{C}_E)} A_x^z$$

$$\le \frac{\gamma_\varepsilon \gamma_t \xi'}{\varepsilon^z} \sum_{x \in N(\mathcal{C}_C)} O_x^z + \gamma_\varepsilon \gamma_t \sum_{x \in N(\mathcal{C}_E)} (O_x^z - A_x^z) + \sum_{x \in N(\mathcal{C}_E)} A_x^z,$$

where $\gamma_\varepsilon = 1 + \frac{\varepsilon}{50}$.

Since $\mathrm{Cost}(N(\mathcal{C}_C), \mathcal{C}) \leq \frac{\varepsilon^{z+1}}{100\xi'}\mathrm{Cost}(S, \mathcal{C})$ and $\frac{1}{t} \leq \frac{\varepsilon}{100}$,

$$\frac{\gamma_\varepsilon \gamma_t \xi'}{\varepsilon^z} \sum_{x \in N(\mathcal{C}_C)} O_x^z = \frac{\gamma_\varepsilon \gamma_t \xi'}{\varepsilon^z}\mathrm{Cost}(N(\mathcal{C}_C), \mathcal{C})$$

$$\leq \gamma_\varepsilon \gamma_t \frac{\xi'}{\varepsilon^z}\frac{\varepsilon^{z+1}}{100\xi'}\mathrm{Cost}(S, \mathcal{C})$$

$$\leq \frac{\gamma_\varepsilon \gamma_t}{100} \cdot \varepsilon \cdot \mathrm{Cost}(S, \mathcal{C}).$$

Since $\gamma_\varepsilon = 1 + \frac{\varepsilon}{50}$, $\gamma_t = 1 + \frac{1}{t}$, $\frac{1}{t} \leq \frac{\varepsilon}{100}$, and $\varepsilon \in (0, \frac{1}{2}]$, we get

$$\frac{\gamma_\varepsilon \gamma_t \xi'}{\varepsilon^z} \sum_{x \in N(\mathcal{C}_C)} O_x^z \leq \left(1 + \frac{\varepsilon}{50}\right)\left(1 + \frac{\varepsilon}{100}\right)\frac{\varepsilon}{100}\mathrm{Cost}(S, \mathcal{C})$$

$$\leq \frac{\varepsilon}{25}\mathrm{Cost}(S, \mathcal{C}).$$

Hence

$$\mathrm{Cost}(S, \mathcal{P}) \leq \frac{\varepsilon}{25}\mathrm{Cost}(S, \mathcal{C}) + \gamma_\varepsilon \gamma_t \sum_{x \in N(\mathcal{C}_E)}(O_x^z - A_x^z) + \sum_{x \in N(\mathcal{C}_E)} A_x^z.$$

For $\gamma_\varepsilon \gamma_t$, since $\varepsilon \in (0, \frac{1}{2}]$, it holds that

$$\gamma_\varepsilon \gamma_t = \left(1 + \frac{\varepsilon}{50}\right)\left(1 + \frac{\varepsilon}{100}\right)$$

$$= 1 + \frac{\varepsilon}{50} + \frac{\varepsilon}{100} + \frac{\varepsilon^2}{5000}$$

$$\leq 1 + \frac{\varepsilon}{10}.$$

Hence

$$\left(1 + \frac{\varepsilon}{10}\right)\sum_{x \in N(\mathcal{C}_E)}(O_x^z - A_x^z) + \sum_{x \in N(\mathcal{C}_E)} A_x^z \leq \left(1 + \frac{\varepsilon}{10}\right)\sum_{x \in N(\mathcal{C}_E)} O_x^z.$$

Thus we get

$$\mathrm{Cost}(S, \mathcal{P}) \leq \frac{\varepsilon}{25}\mathrm{Cost}(S, \mathcal{C}) + \left(1 + \frac{\varepsilon}{10}\right)\sum_{x \in N(\mathcal{C}_E)} O_x^z$$

$$= \frac{\varepsilon}{25}\mathrm{Cost}(S, \mathcal{C}) + \left(1 + \frac{\varepsilon}{10}\right)\mathrm{Cost}(N(\mathcal{C}_E), \mathcal{C}).$$

Since $N(\mathcal{C}_E) \subset S$, $\mathrm{Cost}(N(\mathcal{C}_E), \mathcal{C}) \leq \mathrm{Cost}(S, \mathcal{C})$, which leads

$$\mathrm{Cost}(S, \mathcal{P}) \leq (1 + \varepsilon)\mathrm{Cost}(S, \mathcal{C}).$$

Hence we complete our proof. $\qquad\square$

# E   PTAS FOR HEAVILY SKEWED SET

## E.1   FAST LOCAL SEARCH

In this subsection, we will prove Lemma 5.1.

**Lemma E.1.** *Let $S$ be a dataset of $n$ points, $T$ be the potential center set, and $\mathcal{A} = \mathcal{C}_E$, which is the set of centers of the $s$ most high-cost clusters in optimal solution. There exists a constant $\gamma > 1$, such that for any $\varepsilon \in (0, \frac{1}{2}]$, FASTLOCALSEARCH terminates within $\mathcal{O}(\frac{k^2}{\varepsilon})$ swaps, and returns a $(1 + 2\varepsilon)$-approximation $\mathcal{P}$, as long as $S$ is $(s, 1 - \varepsilon^{z+1})$-skewed. Furthermore, for $z = 1$, $S$ only needs to be $(s, 1 - \varepsilon)$-skewed.*

At first glance, this theorem may appear trivial because Lemma 4.1 guarantees a locally optimal solution $\mathcal{P}'$ which is a $(1 + \frac{\varepsilon}{2})$-approximation of the optimal solution. We might then assume that our result $\mathcal{P}$ from FASTLOCALSEARCH yields a $\mathcal{P}$ such that $\text{Cost}(S, \mathcal{P}) \leq \left(1 - \frac{\varepsilon}{\Gamma k^2}\right)^{-1} \text{Cost}(S, \mathcal{P}') \leq (1 + \varepsilon)\text{Cost}(S, \mathcal{C})$. However, this assumption is incorrect because we can only ensure that for any $\mathcal{P}''$ with no more than $t$ different centers from $\mathcal{P}$, the condition $\left(1 - \frac{\varepsilon}{k^2}\right) \text{Cost}(S, \mathcal{P}) \leq \text{Cost}(S, \mathcal{P}'')$ holds. We cannot guarantee that the locally optimal solution $\mathcal{P}'$ returned by HEAVYSKEWLOCALSEARCH is obtainable by just a single swap from our result $\mathcal{P}$.

To establish Lemma 5.1, it is necessary to replicate the proof framework used in Lemma 4.1 and Lemma 4.2. Specifically, we will demonstrate a variation of Lemma D.3, Lemma D.9, and Lemma D.5. The proofs of the corresponding variations for Lemma D.8, Lemma D.9, and Lemma D.10 will be omitted due to their similarity to the $k$-median case. The notation introduced in Appendix D will be maintained throughout.

**Lemma E.2.** *Let* $(\mathcal{F}, \mathcal{R})$ *be a pairing that* $|\mathcal{F}| = |\mathcal{R}| \leq t$ *and* $\mathcal{R}$ *don't capture* $\mathcal{F}$, *then*

$$-\frac{\varepsilon}{k^2}Cost(S,\mathcal{C}) \leq \sum_{x \in N(\mathcal{F})} (O_x - A_x) + \sum_{\mathcal{U}}(O_x - A_x) + \sum_{\mathcal{V}} 2O_x.$$

*Proof.* We prove Lemma D.3 by these two fact:

$$0 \leq \text{Cost}(S, \mathcal{P}\backslash\mathcal{R} \cup \mathcal{F}) - \text{Cost}(S, \mathcal{P})$$

and

$$\text{Cost}(S, \mathcal{P}\backslash\mathcal{R} \cup \mathcal{F}) - \text{Cost}(S, \mathcal{P}) \leq \sum_{x \in N(\mathcal{F})} (O_x - A_x) + \sum_{\mathcal{U}}(O_x - A_x) + \sum_{\mathcal{V}} 2O_x.$$

The second inequality is still true because we do not use the fact that $\mathcal{P}$ is a local optimum to prove the second inequality.

For the first inequality, it is no longer true because our $\mathcal{P}$ may not be the local optimum. However, we have

$$\text{Cost}(S, \mathcal{P}\backslash\mathcal{R} \cup \mathcal{F}) \geq (1 - \frac{\varepsilon}{\Gamma k^2})\text{Cost}(S, \mathcal{P})$$

because we only terminate local search if there does not exist $\mathcal{P}'$ such that $\text{Cost}(S, \mathcal{P}') < (1 - \frac{\varepsilon}{\Gamma k^2})\text{Cost}(S, \mathcal{P})$.

Since $\Gamma \geq \text{Cost}(S, \mathcal{P})$, we get

$$\text{Cost}(S, \mathcal{P}\backslash\mathcal{R} \cup \mathcal{F}) - \text{Cost}(S, \mathcal{P}) \geq -\frac{\varepsilon}{\Gamma k^2}\text{Cost}(S, \mathcal{P})$$
$$\geq -\frac{\varepsilon}{k^2}$$
$$\geq -\frac{\varepsilon}{k^2}\text{Cost}(S, \mathcal{C}).$$

Thus

$$-\frac{\varepsilon}{k^2}\text{Cost}(S,\mathcal{C}) \leq \sum_{x \in N(\mathcal{F})} (O_x - A_x) + \sum_{\mathcal{U}}(O_x - A_x) + \sum_{\mathcal{V}} 2O_x.$$

$\square$

**Lemma E.3.** *If* $|\mathcal{F}| = |\mathcal{R}| > t$, $\mathcal{R}$ *has exactly one positive degree center, and* $\mathcal{R}$ *don't capture any center outside* $\mathcal{F}$, *the following inequality holds:*

$$-\frac{\varepsilon}{k}Cost(S,\mathcal{C}) \leq \sum_{x \in N(\mathcal{F})} (O_x - A_x) + \left(1 + \frac{1}{t}\right)\left(\sum_{\mathcal{U}}(O_x - A_x) + \sum_{\mathcal{V}} 2O_x\right).$$

*Proof.* We just repeat the proof of Lemma D.4, but substitute Lemma E.2 with Lemma D.3. We use Lemma E.2 $m(m - 1)$ times and add them together, where $m = |\mathcal{F}|$. Hence we get

$$\gamma\text{Cost}(S,\mathcal{C}) \leq m \sum_{x \in N(\mathcal{F})} (O_x - A_x) + (m - 1)\left(\sum_{\mathcal{U}}(O_x - A_x) + \sum_{\mathcal{V}} 2O_x\right),$$

where $\gamma = -\frac{\varepsilon m(m-1)}{k^2}$.

We divide $m$ on both sides. Since we have proved that $\frac{m-1}{m} \leq 1 + \frac{1}{t}$ and $\left(\sum_{\mathcal{U}}(O_x - A_x) + \sum_{\mathcal{V}} 2O_x\right) \geq 0$, we get

$$\frac{\gamma}{m}\text{Cost}(S,\mathcal{C}) \leq \sum_{x \in N(\mathcal{F})} (O_x - A_x) + \left(1 + \frac{1}{t}\right)\left(\sum_{\mathcal{U}}(O_x - A_x) + \sum_{\mathcal{V}} 2O_x\right).$$

Since $m = |\mathcal{F}| \leq k$, we have

$$\frac{\gamma}{m} = -\frac{\varepsilon(m-1)}{k^2} \geq -\frac{\varepsilon}{k}.$$

Hence

$$-\frac{\varepsilon}{k}\text{Cost}(S,\mathcal{C}) \leq \sum_{x \in N(\mathcal{F})} (O_x - A_x) + \left(1 + \frac{1}{t}\right)\left(\sum_{\mathcal{U}}(O_x - A_x) + \sum_{\mathcal{V}} 2O_x\right).$$

$\square$

**Lemma E.4.**

$$-\varepsilon Cost(S,\mathcal{C}) + \sum_{x \in N(\mathcal{C}_C)} A_x \leq \gamma_1 \sum_{x \in N(\mathcal{C}_C)} O_x + \gamma_2 \sum_{x \in N(\mathcal{C}_E)} (O_x - A_x),$$

*where $\gamma_1 = 3 + \frac{2}{t}$, and $\gamma_2 = 1 + \frac{1}{t}$.*

*Proof.* We repeat the proof of Lemma D.5, but substitute Lemma E.2 and Lemma E.3 with Lemma D.3 and Lemma D.4. Since we have the partition pair $\{(\mathcal{F}_i, \mathcal{R}_i)\}_{i=1}^l$, and we take the inequality for each pair and add them together, we get

$$-\frac{\varepsilon \cdot l}{k}\text{Cost}(S,\mathcal{C}) + \sum_{x \in N(\mathcal{C}_C)} A_x \leq \gamma_1 \sum_{x \in N(\mathcal{C}_C)} O_x + \gamma_2 \sum_{x \in N(\mathcal{C}_E)} (O_x - A_x).$$

Since $\{(\mathcal{F}_i, \mathcal{R}_i)\}_{i=1}^l$ is a partition of $(\mathcal{C}_C, \mathcal{B})$, we have $l \leq k$. Then we get

$$-\varepsilon\text{Cost}(S,\mathcal{C}) + \sum_{x \in N(\mathcal{C}_C)} A_x \leq \gamma_1 \sum_{x \in N(\mathcal{C}_C)} O_x + \gamma_2 \sum_{x \in N(\mathcal{C}_E)} (O_x - A_x).$$

$\square$

Finally, we demonstrate Lemma 5.1.

*Proof.* For the portion of the theorem concerned with accuracy, the argument is simply a reiteration of Lemma 4.1. In the case of the $k$-median, the framework remains the same, but Lemma E.2, Lemma E.3, and Lemma E.4 are substituted with Lemma D.3, Lemma D.4, and Lemma D.5, respectively.

Then we get

$$-\varepsilon\text{Cost}(S,\mathcal{C}) + \text{Cost}(S,\mathcal{P}) \leq (1+\varepsilon)(S,\mathcal{C}),$$

which is equivalent to

$$\text{Cost}(S,\mathcal{P}) \leq (1+2\varepsilon)(S,\mathcal{C}).$$

The proof for the $(k, z)$-clustering scenario is excluded since it closely resembles that of the $k$-median case.

Then, we will prove the portion of the theorem concerned with run time. The case for $|S| \leq k$ is just trivial. For $|S| > k$, we have shown in the proof of Lemma 3.3 that $\text{Cost}(S,\mathcal{C}) \geq \frac{1}{2^z}$. We begin our local search with $\text{Cost}(S, \mathcal{A} \cup \mathcal{B}) = \Gamma$. Since we improve the cost of our center set with a factor at least $1 - \frac{\varepsilon}{\Gamma k^2}$, we can swap for at most $r$ rounds, where

$$r = \log_{1-\frac{\varepsilon}{\Gamma k^2}} \frac{1}{2^z \Gamma} = \frac{\log\left(\frac{1}{2^z \Gamma}\right)}{\log\left(1 - \frac{\varepsilon}{\Gamma k^2}\right)} = \mathcal{O}(\frac{k^2}{\varepsilon}).$$

$\square$

### E.2 DISCRETE HEAVY SKEW AND CONTINUOUS HEAVY SKEW

Finally, we will prove Theorem 5.2 and Theorem 5.3.

**Theorem E.5.** *Let $X$ be a set of $n$ data points, and let $T$ be a set of potential centers such that $|T| = \mathsf{poly}(n)$. Given any $\varepsilon > 0$, DISCRETEHEAVYSKEW returns a $(1 + \varepsilon)$-approximation $\mathcal{P}$ in $(nk/\varepsilon)^{\mathcal{O}(s+1/\varepsilon)}$ time for discrete $(k, z)$-clustering as long as $X$ is $(s, 1 - \varepsilon^{z+1})$-skewed. Furthermore, for $z = 1$, $X$ only needs to be $(s, 1 - \varepsilon)$-skewed.*

*Proof.* If $|X| = k$ and $X \subset T$, the problem is trivial since the optimal solution is just $X$, and the optimal cost is just $0$.

Otherwise, we will run FASTLOCALSEARCH$(X, T, \frac{\varepsilon}{2}, \mathcal{A}, k, s)$ for all possible $\mathcal{A}$, and return the one with cheapest cost. By Lemma 5.1, we know that FASTLOCALSEARCH$(X, T, \frac{\varepsilon}{2}, \mathcal{C}_E, k, s)$ returns a set $\mathcal{P}'$ with $\mathrm{Cost}(S, \mathcal{P}') \leq (1 + \varepsilon) \mathrm{Cost}(S, \mathcal{C})$, where $\mathcal{C}$ is the optimal solution for the clustering on $T$. Hence, we prove the accuracy claim of the theorem.

If $|X| = k$ and $X \subset T$, then naturally, the running time is polynomial.

Otherwise, we run FASTLOCALSEARCH$(X, T, \frac{\varepsilon}{2}, \mathcal{A}, k, s)$ for all possible $\mathcal{A}$. Since $\mathcal{A} \in T^s$ and $|T| = \mathsf{poly}(n)$, we will repeat FASTLOCALSEARCH$(X, T, \frac{\varepsilon}{2}, \mathcal{A}, k, s)$ for $2^{\mathcal{O}(s \log n)}$ times.

For every time we run FASTLOCALSEARCH$(X, T, \frac{\varepsilon}{2}, \mathcal{A}, k, s)$, by Lemma 5.1, we will terminate after no more than $\mathcal{O}(\frac{k^2}{\varepsilon})$ swaps.

For every swap, we need to check whether the exists a swap meets our condition. For the worst case, we may check every possible swapping. Since we swap for $t$ centers, it takes $|T|^t = 2^{\mathcal{O}(\frac{1}{\varepsilon} \log n)}$ running time.

By multiplying the three terms together, we get the total run time $2^{\mathcal{O}((s + \frac{1}{\varepsilon}) \log n)} \cdot \frac{k^2}{\varepsilon} = (nk/\varepsilon)^{\mathcal{O}(s+1/\varepsilon)}$.

For Zipfian data set with exponent $p > 1$, by Lemma 2.3, $s = \mathcal{O}(1/\varepsilon^{(z+1)/(p-1)})$. Therefore, we complete our proof. $\qquad\square$

Next, we establish Theorem 5.3.

**Theorem E.6.** *Let $X$ be a set of $n$ data points. Given any $\varepsilon > 0$, CONTINUOUSHEAVYSKEW returns a $(1+\varepsilon)$-approximation $\mathcal{P}$ in $\tilde{\mathcal{O}}(nk) + (k \log n)^{\tilde{\mathcal{O}}(s+1/\varepsilon)}$ time for continuous $(k, z)$-clustering with probability at least $0.97$, as long as $X$ is $(s, 1 - \varepsilon^{z+1})$-skewed. Furthermore, for $z = 1$, $X$ only needs to be $(s, 1 - \varepsilon)$-skewed.*

*Proof.* If $|X| = k$, the problem is trivial, as the optimal solution is just $X$ and the optimal cost is just $0$.

In the case $|X| > k$, we will execute CORESETCONSTRUCTION$(X, \varepsilon, n, k, \Delta)$ to form a coreset $S$. According to Lemma 3.2, when $\mu > \frac{\gamma dk}{\varepsilon^3} \log(n\Delta)$, there is at least a $0.97$ probability that $S$ is an $\frac{\varepsilon}{8}$-coreset of $X$, and $S$ is $(s, 1 - \varepsilon)$-skewed. Subsequently, we run CENTERNET$(S, \frac{\varepsilon}{4}, \Delta)$ to obtain $T$. By Lemma 3.3, the optimal solution $\mathcal{C}^*$ for discrete $(k, z)$-clustering on $T$ serves as a $\left(1 + \frac{\varepsilon}{4}\right)$-approximation of the optimal solution $\mathcal{C}$ for continuous $(k, z)$-clustering on $S$. Finally, we carry out DISCRETEHEAVYSKEW$(X, T, \frac{\varepsilon}{4}, k, s)$ to produce a $(1 + \frac{\varepsilon}{4})$-approximation for the discrete $(k, z)$-clustering on $T$. Therefore

$$
\begin{aligned}
\mathrm{Cost}(X, \mathcal{P}) &\leq \left(1 + \frac{\varepsilon}{8}\right) \mathrm{Cost}(S, \mathcal{P}) \\
&\leq \left(1 + \frac{\varepsilon}{8}\right) \cdot \left(1 + \frac{\varepsilon}{4}\right) \mathrm{Cost}(S, \mathcal{C}^*) \\
&\leq \left(1 + \frac{\varepsilon}{8}\right) \cdot \left(1 + \frac{\varepsilon}{4}\right)^2 \mathrm{Cost}(S, \mathcal{C}) \\
&\leq \left(1 + \frac{\varepsilon}{8}\right)^2 \cdot \left(1 + \frac{\varepsilon}{4}\right)^2 \mathrm{Cost}(X, \mathcal{C}) \\
&\leq (1 + \varepsilon) \mathrm{Cost}(X, \mathcal{C}).
\end{aligned}
$$

For running time, Bhattacharya et al. (2023) shows that sensitivity sampling can be completed in $\tilde{\mathcal{O}}(nk)$ time.

For the construction of $T$, the run time is just the size of $|T|$. By Lemma 3.3, $|T| = |S| \cdot 2^{\mathcal{O}(d \log \frac{1}{\varepsilon} \log \log(\frac{k\Delta}{\varepsilon}))} = (k \log n)^{\mathcal{O}(d \text{polylog}(1/\varepsilon))}$.

Then we run FASTLOCALSEARCH$(X, T, \frac{\varepsilon}{4}, \mathcal{A}, k, s)$ for all possible $\mathcal{A}$. Since $\mathcal{A} \in T^s$, we repeat FASTLOCALSEARCH$(X, T, \frac{\varepsilon}{4}, \mathcal{A}, k, s)$ for $|T|^s$ times. For every time we run FASTLOCALSEARCH$(X, T, \frac{\varepsilon}{4}, \mathcal{A}, k, s)$, by Lemma 5.1, we will terminate after no more than $\frac{k^2}{\varepsilon}$poly$(|S|)$ swaps. For every swap, we need to check whether the swap meets our condition. For the worst case, we may check every possible swapping. Since we swap for $t$ centers, it takes $|T|^t = |T|^{\mathcal{O}(1/\varepsilon)}$ running time. Multiplying these three terms together, we get the running time for FASTLOCALSEARCH is $\frac{k^2}{\varepsilon}$poly$(|S|) \cdot |T|^{\mathcal{O}(s+/\varepsilon)} = (k \log n)^{\tilde{\mathcal{O}}(d(s+1/\varepsilon))}$.

By adding the running time for every part of the algorithm, the total running time is $\tilde{\mathcal{O}}(nk) + (k \log n)^{\tilde{\mathcal{O}}(d(s+1/\varepsilon))}$. If we assume $d$ as a constant, it would be $\tilde{\mathcal{O}}(nk) + (k \log n)^{\tilde{\mathcal{O}}(s+1/\varepsilon)}$. For a large $d$, a dimension reduction technique introduced by Makarychev et al. (2019) can be used. It reduce $d$ to $\mathcal{O}(\frac{\log \frac{k}{\varepsilon}}{\varepsilon^2})$, which makes $|T| = |S| \cdot 2^{\mathcal{O}(d \log \frac{1}{\varepsilon} \log \log(\frac{k\Delta}{\varepsilon}))} = (k \log n)^{\tilde{\mathcal{O}}(1/\varepsilon^2)}$. Then the running time for the algorithm will be $\tilde{\mathcal{O}}(nk) + (k \log n)^{\tilde{\mathcal{O}}(\varepsilon^{-2}(s+1/\varepsilon))}$. □

# F SUPPLEMENTARY FOR SENSITIVITY EVALUATION AND DIMENSION REDUCTION

As a widely used protocol, several studies propose algorithms to evaluate the sensitivity of a point in a short run time. For instance, Algorithm 1 proposed by Draganov et al. (2024) computes the sensitivity of all points in the dataset and returns a coreset by sensitivity sampling with $\tilde{\mathcal{O}}(nd \log n\Delta)$ run time. Although Draganov et al. (2024) only discuss the case that $z = 1$ and 2, their method works for general $z$.

---

**Algorithm 7** FASTCORESET$(X, k, \varepsilon, m)$

---

**Require:** Dataset $X$, number of cluster $k$, precision parameter $\varepsilon$, target size $m$
**Ensure:** A weighted set $S$
 1: Use a Johnson-Lindenstrauss embedding to embed $\tilde{X}$ of $X$ into $d' = \mathcal{O}(\log k)$ dimensions
 2: Find approximate solution $\tilde{\mathcal{C}} = \{\tilde{c}_1, \cdots, \tilde{c}_k\}$ on $\tilde{X}$ and assignment $\tilde{\sigma} : \tilde{X} \to \tilde{C}$ by FASTKMEANS++
 3: Let $\mathcal{C}_i = \tilde{\sigma}^{-1}(\tilde{c}_i)$. Compute the $(1, z)$-clustering solution $c_i$ of each $\mathcal{C}_i$ in $\mathbb{R}^d$
 4: For each point $x \in \mathcal{C}_i$ define $s(x) = \frac{\text{dist}^z(x, c_i)}{\text{Cost}(\mathcal{C}, c_i)} + \frac{1}{|\mathcal{C}_i|}$.
 5: Compute a set $S$ of $m$ points randomly sampled from $X$ proportionate to $s(x)$.
 6: For each $\mathcal{C}_i$, define $|\hat{\mathcal{C}}_i|$ the estimated weight of $\mathcal{C}_i$ by $S$, namely $|\hat{\mathcal{C}}_i| = \sum_{x \in \mathcal{C}_i \cap S} \frac{\sum_{x' \in S} s(x')}{s(x)m}$.
 7: **return** The coreset $S$, with weight $w(x) = \frac{\sum_{x; \in S} s(x')}{s(x)m} \left( (1+\varepsilon)|\mathcal{C}_i| - |\hat{\mathcal{C}}_i| \right)$.

---

FASTKMEANS++ is an algorithm proposed by Cohen-Addad et al. (2020).

**Theorem F.1.** *There exists an algorithm, cf. algorithm 1 in Draganov et al. (2024), which computes the sensitivity of all points in a dataset $X$ and returns a coreset of $X$ for $(k, z)$-clustering by sensitivity sampling with $\tilde{\mathcal{O}}(nd \log(n\Delta))$ run time.*

To avoid the exponential dependency on $d$, we can apply Johnson–Lindenstrauss to project the coreset $S$ into $\pi(S) \subset \mathbb{R}^{d'}$, where $d' = \mathcal{O}(\frac{\log \frac{k}{\varepsilon}}{\varepsilon^2})$, and apply our algorithm to find a $(1 + \varepsilon)$-approximation for $\pi(S)$. The $(1 + \mathcal{O}(\varepsilon))$-approximation for $\pi(S)$ induces a cluster partition $\{A_1, A_2, \cdots, A_k\}$ of $S$, which is a good approximation of the optimal partition. Then we can find the solution $c_i$ for the $(1, z)$-clustering for each $A_i$, and $\mathcal{C} = \{c_1, c_2, \cdots, c_k\}$ would be a $(1 + \mathcal{O}(\varepsilon))$-approximation for the $(k, z)$-clustering on $S$. poly$(n, d, k)$ time is needed to generate $S$, and the size

of the center net would be $|T| = \text{poly}(n, k)$, which means that it takes $\text{poly}(n, k)$ time and, finally, it takes $\text{poly}(n, k, d)$ time to solve the $(1, z)$-clustering for each $A_i$ since it is a convex optimization. Therefore, the total run time is $\text{poly}(n, k, d)$.

**Theorem F.2.** *Let $X$ be a set of $n$ data points. There exists an algorithm that, given any $\varepsilon > 0$, for continuous $(k, z)$-clustering, in $\tilde{\mathcal{O}}(dnk) + (dk \log n)^{\tilde{\mathcal{O}}(\frac{1}{\varepsilon^2}(s + \frac{1}{\varepsilon}))}$ time returns a $(1 + \varepsilon)$-approximation $\mathcal{P}$ with probability at least $0.97$ as long as $X$ is $(s, 1 - \varepsilon^{z+1})$-skewed. Furthermore, for $z = 1$, $X$ only needs to be $(s, 1 - \varepsilon)$-skewed.*

We recall the theorem in Makarychev et al. (2019).

**Theorem F.3** (Theorem 1.3 in Makarychev et al. (2019))**.** *There exists a family of random maps $\pi_{m,d} : \mathbb{R}^d \to \mathbb{R}^{d'}$ that for every $m \geq 1, \varepsilon, \delta \in (0, \frac{1}{4})$ and $z \geq 1$, the following holds. For any $x \in \mathbb{R}^d$ we have*

$$\Pr_{\pi \sim \pi_{m,d}} [\|\pi(x)\| \approx_{1+\varepsilon} \|x\|] \geq 1 - \delta$$

*and for every finite $X \subset \mathbb{R}^d$ we have*

$$\Pr_{\pi \sim \pi_{m,d}} [Cost_z \mathcal{A} \approx_{1+\varepsilon} Cost_z \pi(\mathcal{A}) \text{ for all partitions } \mathcal{A} = \{A_1, A_2, \cdots, A_k\} \text{ of } X] \geq 1 - \delta,$$

*where*

$$d' = \mathcal{O}(\frac{z^4 \cdot \log \frac{k}{\varepsilon\delta}}{\varepsilon^2})$$

*and*

$$Cost_z \mathcal{A} = \sum_{i=1}^{k} \min_{u_i \in \mathbb{R}^d} \sum_{x \in A_i} dist(x - u_i)^z.$$

Now we prove Theorem F.2.

*Proof.* First, applying CORESETCONSTRUCTION, we can get a coreset $S$ with size $\mathcal{O}(\frac{dk^2}{\varepsilon^3} \log(n\Delta))$. By Theorem F.1, we can generate $S$ in $\tilde{\mathcal{O}}(nd \log(n\Delta))$ time.

Second, we use $\pi$ to project $S$ to $\mathbb{R}^{d'}$ for $d' = \mathcal{O}(\frac{z^4 \cdot \log \frac{k}{\varepsilon\delta}}{\varepsilon^2})$. Then we apply CENTERNET and DISCRETEHEAVYSKEW to find a $(1+\varepsilon)$-approximation of the optimal solution on $\pi(S)$ for $(k, z)$-clustering. Assume $\pi(\mathcal{A}) = \{\pi(A_1), \pi(A_2), \cdots, \pi(A_k)\}$ to be the partition of $\pi(S)$ corresponding to this solution. We claim that $\mathcal{A}$ gives a $(1 + \mathcal{O}(\varepsilon))$-approximation of $S$.

Assume $\mathcal{B} = \{B_1, B_2, \cdots, B_k\}$ to be the partition of $S$ corresponding to the optimal solution for $(k, z)$-clustering on $S$, and $\mathcal{D} = \{D_1, D_2, \cdots, D_k\}$ to be the partition of $\pi(S)$ corresponding to the optimal solution for $(k, z)$-clustering on $\pi(S)$. By Theorem F.3, $Cost_z \mathcal{A} \leq (1+\varepsilon)Cost_z \pi(\mathcal{A})$. Since $Cost_z \pi(\mathcal{A})$ is a $(1 + \varepsilon)$-approximation of $Cost_z \mathcal{D}$, and $\mathcal{D}$ is the optimal solution of $\pi(S)$ for $(k, z)$-clustering, therefore

$$Cost_z \mathcal{A} \leq (1+\varepsilon)Cost_z \pi(\mathcal{A}) \leq (1+\varepsilon)^2 Cost_z \mathcal{D} \leq (1+\varepsilon)^2 Cost_z \pi(\mathcal{B}) \leq (1+\varepsilon)^3 Cost_z \mathcal{B}.$$

Let $\mathcal{C} = \{c_1, c_2, \cdots, c_k\}$, where $c_i = \arg\min_{c \in \mathbb{R}^d} \text{Cost}(A_i, c)$. Then $\text{Cost}(S, \mathcal{C}) = Cost_z \mathcal{A} \leq (1 + \mathcal{O}(\varepsilon)) Cost_z \mathcal{B} = \text{Cost}(S, \mathcal{C}_{\text{OPT}})$. Since $S$ is a $(1 + \varepsilon)$-coreset of $X$, $\mathcal{C}$ would be a $(1 + \mathcal{O}(\varepsilon))$-approximation for $(k, z)$-clustering on $X$.

Fortunately, although $(k, z)$-clustering is APX-hard, it is possible to find a $(1+\varepsilon)$-approximation of $c_i$ in polynomial time. In fact, the problem reduces to a $(1, z)$-clustering when we look for $c_i$. We can apply Weiszfeld's algorithm (Weiszfeld, 1937) to find a $(1+\varepsilon)$-approximation of $c_i$ when $z = 1$. When $z > 1$, the problem becomes a convex optimization since the cost function is convex. Since the cost function is also differentiable, we can use gradient descent to find a $(1 + \varepsilon)$-approximation of $c_i$. Therefore, we can find a $(1 + \varepsilon)$-approximation of $c_i$ in $\mathcal{O}(nd \log \frac{1}{\varepsilon})$ time.

Since $|S| = \mathcal{O}(\frac{dk^2}{\varepsilon^3} \log(n\Delta))$, thus the size of center net $T$ would be

$$|T| = |S| + (\log \Delta - \log \left(\frac{\varepsilon}{kW}\right) + 2z + 2) \cdot |S| \cdot 2^{\mathcal{O}(d' \log(\frac{1}{\varepsilon}))}$$

Table 3: Skewness of dataset in cli (2019)

|  | $p = 50\%$ | $p = 75\%$ | $p = 90\%$ | $p = 95\%$ |
|---|---|---|---|---|
| $k = 8$ | 12.5% | 12.5% | 12.5% | 12.5% |
| $k = 16$ | 6.25% | 6.25% | 6.25% | 6.25% |
| $k = 32$ | 6.25% | 9.375% | 12.5% | 12.5% |
| $k = 64$ | 1.563% | 1.563% | 3.125% | 3.125% |
| $k = 128$ | 0.781% | 1.563% | 1.563% | 1.563% |

Table 4: Skewness of dataset in cli (2019) when $k \in [80, 160]$

|  | $p = 50\%$ | $p = 75\%$ | $p = 90\%$ | $p = 95\%$ |
|---|---|---|---|---|
| $k = 80$ | 1.25% | 1.25% | 2.5% | 3.75% |
| $k = 100$ | 1.0% | 2.0% | 2.0% | 2.0% |
| $k = 120$ | 0.833% | 1.667% | 1.667% | 1.667% |
| $k = 140$ | 0.714% | 1.429% | 1.429% | 1.429% |
| $k = 160$ | 0.625% | 0.625% | 0.625% | 13.125% |

according to the proof of Lemma C.12, where $W = \text{poly}(n)$ is the maximum weight of $S$. Since $d' = \mathcal{O}(\frac{z^4 \cdot \log \frac{k}{\varepsilon \delta}}{\varepsilon^2})$, thus

$$|T| = |S| + \mathcal{O}(\log(n\Delta) + \log \frac{1}{\varepsilon}) \cdot |S| \cdot 2^{\mathcal{O}(\frac{z^4 \cdot \log \frac{k}{\varepsilon \delta}}{\varepsilon^2})} = 2^{\mathcal{O}(\log d + \log k + \log \log(n\Delta) + \frac{1}{\varepsilon^2} \text{polylog}(\frac{1}{\varepsilon}))}.$$

Therefore, we can run DISCRETEHEAVYSKEW on $T$ to find a $(1 + \varepsilon)$-approximation $\mathcal{A}$ of $\pi(S)$ in $(dk \log n)^{\tilde{\mathcal{O}}(\frac{1}{\varepsilon^2}(s + \frac{1}{\varepsilon}))}$, and find a $(1 + \varepsilon)$-approximation solution to $\mathcal{A}$ in $\mathcal{O}(ndk \log \frac{1}{\varepsilon})$ time. Thus we can find a $(1 + \mathcal{O}(\varepsilon))$-approximation to $X$ in $\tilde{\mathcal{O}}(dnk) + (dk \log n)^{\tilde{\mathcal{O}}(\frac{1}{\varepsilon^2}(s + \frac{1}{\varepsilon}))}$ time. $\qquad\square$

# G SUPPLEMENTARY EXPERIMENTS

## G.1 INSTANCE FOR DATASET WITH HEAVY SKEWNESS

The run time of our algorithm depends on the skewness of the dataset. Due to the APX-hardness, there does not exist any algorithm that is fast for any datasets. Therefore, our algorithm focuses on performance on specific datasets that have heavy skewness only. We will display some datasets with heavy skewness in real world.

cli (2019) offers a dataset contains information on the clickstream of an online store that offers clothing for pregnant women, which has 165474 instances. We show the skewness of this dataset in Table 3. The table illustrates the contribution of the most expensive clusters to the total cost in a k-means clustering solution. Each row corresponds to a value of $k$, the number of clusters. Each column represents a threshold $p$, which denotes a percentage of the total cost (e.g., 50%, 70%, etc.). The value in the cell in the row $k$ and the column $p$ indicates the proportion of clusters (as a percentage of $k$) that contributes at least $p$ of the total cost. For instance: A value of 12.5% in the cell in row $k = 8$ and column $p = 95\%$ means that the 12.5% most expensive clusters (1 clusters out of 8) contribute at least 95% to the total cost. This table highlights the skewness of the dataset, demonstrating that a small subset of clusters can dominate the total cost.

The dataset in cli (2019) has an extremely high skewness when $k \in [80, 160]$. We further show its skewness when $k \in [80, 160]$ in Table 4

gen (2020) is another dataset with a heavy skewness. The dataset attributes first names to genders and has 147270 instances. We disply its skewness in Table 5 by the same way as Table 3 and Table 4.

At last, we display the skewness of Exa20. The dataset comprises 399 instances and 4 features. This data set includes demographic information on 4 groups of saliva samples (COPD, asthma, infection, HC) collected as part of the joint research project Exasens. Since this dataset has a relatively small

Table 5: Skewness of dataset in gen (2020)

|  | $p = 50\%$ | $p = 75\%$ | $p = 90\%$ | $p = 95\%$ | $p = 99\%$ |
|---|---|---|---|---|---|
| $k = 5000$ | 9.66% | 19.2% | 28.6% | 33.5% | 40.28% |
| $k = 6000$ | 8.05% | 16.1% | 24.2% | 28.85% | 34.6% |
| $k = 7000$ | 6.4% | 13.357% | 19.857% | 23.671% | 29.514% |
| $k = 8000$ | 4.938% | 9.913% | 16.113% | 18.95% | 21.25% |
| $k = 9000$ | 3.756% | 7.022% | 9.356% | 10.144% | 10.767% |
| $k = 10000$ | 0.28% | 0.43% | 0.52% | 0.55% | 0.58% |

Table 6: Skewness of dataset in Exa20

| $p$ | 50% | 75% | 90% | 95% | 99% |
|---|---|---|---|---|---|
| $k = 4$ | 1 | 2 | 2 | 2 | 3 |
| $k = 5$ | 1 | 2 | 2 | 2 | 3 |
| $k = 6$ | 2 | 2 | 3 | 3 | 4 |
| $k = 7$ | 2 | 3 | 4 | 4 | 5 |
| $k = 8$ | 2 | 3 | 5 | 6 | 6 |
| $k = 9$ | 2 | 4 | 6 | 7 | 7 |
| $k = 10$ | 3 | 5 | 7 | 8 | 8 |

size, we will use relatively small $k$. Therefore, we will display the exact number of clusters that contribute more than specific portion of total cost in Table 5, rather than disply the percentage in Table 3, Table 4, Table 5.

## G.2 COMPARISON WITH LOCAL SEARCH

### G.2.1 SYNTHETIC DATA

Figure 5: Comparison between local search and our algorithm for $k$-means

Figure 6: Comparison between local search and our algorithm for $k$-medoids

Our experiments illustrate an improvement range for $k$-means from $11.54\%$ at $k = 4$ for the minimum metric to $54.87\%$ at $k = 10$ for the median metric, and for $k$-medoids from $6.06\%$ at $k = 5$ for the minimum metric to $31.86\%$ at $k = 7$ for the average metric. This overall enhancement underscores the superior performance of our algorithm in terms of accuracy when compared to local search across average, minimum, and median metrics. Furthermore, the notable improvement observed in the average and median metric implies a higher variability in local search when evaluated on synthetic data, whereas our algorithm demonstrates significantly lower variance.

Table 7: Improvement rate for $k$-means and $k$-medoids on synthetic data

| $k$ | $k$-means (%) | | | $k$-medoids (%) | | |
|---|---|---|---|---|---|---|
| | **Avg** | **Min** | **Median** | **Avg** | **Min** | **Median** |
| **4** | 24.85 | 11.54 | 25.07 | 16.40 | 8.14 | 16.98 |
| **5** | 31.88 | 24.59 | 29.48 | 25.15 | 6.06 | 10.93 |
| **6** | 45.64 | 37.95 | 41.42 | 29.28 | 17.80 | 19.76 |
| **7** | 37.10 | 29.61 | 35.79 | 31.86 | 16.15 | 26.94 |
| **8** | 39.39 | 16.65 | 41.34 | 30.83 | 22.13 | 30.65 |
| **9** | 45.08 | 22.91 | 46.45 | 20.64 | 17.16 | 21.5 |
| **10** | 53.07 | 32.52 | 54.87 | 26.64 | 15.18 | 26.82 |

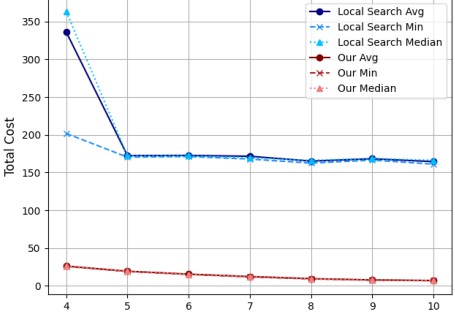

Figure 7: Comparison between Lloyd heuristic and our algorithm for $k$-means

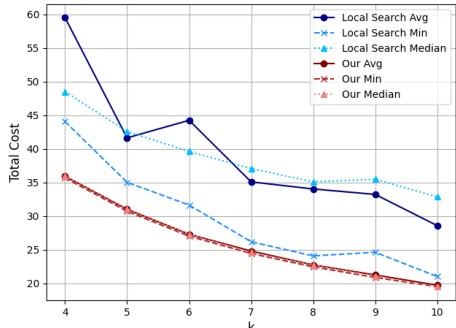

Figure 8: Comparison between KMedoids and our algorithm for $k$-medoids

### G.2.2 REAL WORLD DATA

Our experimental results demonstrate an enhancement range for $k$-means from $87.23\%$ at $k = 4$ for the minimum metric up to $95.77\%$ at $k = 10$ for the median metric, and for $k$-medoids from $6.63\%$ at $k = 7$ for the minimum metric to $40.60\%$ at $k = 10$ for the median metric. This overall improvement highlights the superior accuracy performance of our algorithm relative to local search, across various metrics including average, minimum, and median. Additionally, the observed substantial improvement in the average and median metric suggests greater variability in local search when tested on real world data, while our algorithm displays considerably lower variance.

Table 8: Improvement rate for $k$-means and $k$-medoids on real world data

| $k$ | $k$-means (%) | | | $k$-medoids (%) | | |
|---|---|---|---|---|---|---|
| | Avg | Min | Median | Avg | Min | Median |
| 4 | 92.20 | 87.23 | 92.81 | 39.63 | 19.01 | 25.95 |
| 5 | 88.75 | 88.81 | 88.73 | 25.49 | 12.28 | 27.00 |
| 6 | 91.02 | 91.11 | 91.15 | 38.30 | 14.64 | 31.04 |
| 7 | 92.79 | 92.94 | 92.68 | 29.32 | 6.63 | 33.63 |
| 8 | 94.20 | 94.29 | 94.25 | 33.18 | 6.77 | 35.20 |
| 9 | 95.30 | 95.44 | 95.29 | 36.05 | 15.21 | 40.52 |
| 10 | 95.69 | 95.72 | 95.77 | 31.00 | 7.19 | 40.60 |

