# OpenReview forum: "Clustering on Skewed Cost Distributions"
_ICLR.cc/2025/Conference — Submitted to ICLR 2025_

### Official Review · Reviewer_gzCW · 2024-10-25

**Soundness:** 3
**Presentation:** 3
**Contribution:** 2
**Rating:** 5
**Confidence:** 4

**Summary:**

The paper proposes a method for obtaining a $(1 + \varepsilon)$-approximation to the $(k, z)$-clustering problem. This is known to be APX-hard and the authors therefore constrain their focus to the case where a dataset exhibits strong skewness (some clusters account for a larger portion of the cost than others). In this setting, the authors show that one can obtain an a $(1 + \varepsilon)$-approximation in time $O( poly(k, n)^{1/ \varepsilon} )$. They conclude by experimentally verifying that this method indeed finds a solution of smaller cost than the standard $k$-means and $k$-median techniques.

**Strengths:**

This paper has several clear strengths.

First, the technical matter itself is well-written and easy to follow. The algorithms are clearly laid out and the overview of the proof is clear. I particularly appreciate that the related work is woven into the narrative rather than appearing as a brick wall on one of the pages of the paper.

Second, the theoretical ideas are interesting. The authors show that one can find the approximate solution in the skewed setting by first finding the optimal solutions for the "heavy" clusters and subsequently adapting a local-search algorithm over the remaining, "lighter" clusters. It is intuitively expressed why this would not work if done in the naive setting due to (1) an accumulation of errors and (2) a need to adjust the local search for when some centers (those handling the heavy clusters) must stay in place. Although these arguments are subtle, their description is clear and I find them convincing.

Third, the coreset application almost feels like a separate theoretical contribution. The notion that oversampling according to sensitivities is sufficient to approximate the heavy clusters' cost is interesting and I could see applications of this to other problems.

**Weaknesses:**

This feels like a paper written by people in the theoretical CS community which has been adapted for ICLR by incorporating a few experiments. I am not opposed to this in general -- many valuable insights can be gained by the exchange between these communities. However, I believe that for this paper to be a fit for ICLR, the authors should show that their algorithm addresses an actual need in the machine learning community beyond being theoretically interesting.

### Experiments

I believe that the experiments the authors have chosen are, unfortunately, the least interesting ones and do not highlight why one would use the proposed algorithm. The included experiments simply show that the authors' approximation achieves lower cost than the scikit-learn implementation of $k$-means and $k$-medioid. This falls into the category of "verifying the theory".

However, this experiment is insufficient to justify using the algorithms proposed by the authors. Namely, the authors have not convinced me that there is a good reason to use the proposed techniques over standard k-means++ -> Lloyd's algorithm. The following are a few themes which would help address that concern:
 1. The authors' primary contribution is a faster algorithm for obtaining a $(1 + \varepsilon)$-approximation but they have not shown how fast/slow the algorithm is in practice. How long does the algorithm take to run? How does this change as a function of $n$, $k$, or $\varepsilon$? At what dataset-size does it become infeasible to run? The authors only chose a real-world dataset of 400 data samples -- much smaller than would appear in any realistic scenario that the standard ICLR attendee would be operating in.
 2. The comparison to scikit-learn's clustering lacks depth. Yes -- the authors' $(1+\varepsilon)$-approximation is (as expected) better than the $k$-means++ $\log k$ one. But the scikit-learn algorithm is, I expect, significantly faster. Thus, a fairer comparison would be to time the authors' algorithm on a series of datasets and then repeatedly run the scikit-learn algorithm until the time expires, keeping the minimum cost over these runs. This would actually suggest that the authors' algorithm is a worthwhile use of the runtime in practice. An analysis of this over n would be necessary to show that the authors' proposed algorithm is consistently useful over scikit-learn's kmeans.
 3. The choice of datasets feels insufficient. First -- it is not even verified whether the chosen Exasens dataset is heavily skewed. I understand that testing for the skewness technically requires the optimal solution, but I'm sure it can be checked approximately (indeed, if the skewness cannot be verified then it is unclear when one could even apply the proposed algorithms in practice). So is there even a guarantee that the algorithm should work here? Second, why was this the only chosen dataset? In the absence of evidence to the contrary, one might conclude that the Exasens dataset is the only one where there is a significant difference between the authors' proposed algorithm and the scikit-learn approaches.

### Regarding the scope

This point is a matter of opinion and I am willing to change my mind: although I find this paper to be interesting and well-written, it feels like it's on the outer edges of topics which are relevant to the ICLR community.

That is, I am unconvinced that there is a practical need for approximating the k-means/k-median solution for skewed datasets in faster time than was previously attainable. If such settings do exist, I am unconvinced that within them one would need a solution which is better than the quality of the standard k-means++ -> Lloyd's usecase. Lastly, if one does require a solution of such quality, then the authors' experiments have not convinced me that their proposed algorithm performs better in practice than running scikit-learn's k-means method $\sim 100$ times and picking the best set of centers from that set.

Thus, this paper feels like its applications are primarily geared towards the theoretical computer science community and these results should appear in a theoretical computer science conference rather than at ICLR.

### Potential connection to cluster separability literature

Perhaps I'm misunderstanding something, but I get the sense that the skewness notions described by the authors inherently lead to separability between clusters. Namely, in order for a dataset to satisfy the Zipfian distribution, a heavy cluster (with center $c_h$) in the optimal solution must have exponentially higher cost than some subsequent light cluster (with center $c_\ell$). Suppose now that we remove center $c_\ell$ and instead place two centers on $N(c_h)$. $c_h$ was the heavy cluster, so this should induce a large cost-decrease. However, this new partition is by definition not the optimal solution. Thus, the points $N(c_\ell)$ must now have very large cost in this new partition to offset the cost-decrease we obtained. By extension, we can conclude that the variance within $N(c_\ell)$ is small compared to this cluster's distance to the rest of the data. This seems extremely similar to the proximity conditions described in [2].

Based on the above logic, it feels like a theme of related work is missing. Namely, it is known that one can obtain a $(1 + \varepsilon)$ approximation when the data is sufficiently separable [1]. Thus, it would be helpful to highlight when one might expect the skewness to come into play without simultaneously incurring separability conditions under which the optimal solution can be approximated using known techniques.

[1]: Ostrovsky, Rabani, Schulman, Swamy. The Effectiveness of Lloyd-Type Methods for the k-Means Problem.

[2]: Kumar, Kannan. Clustering with Spectral Norm and the k-means Algorithm.

**Questions:**

The questions are in line with the above discussion and ordered from most to least important:
1. Can the authors comment on when the skewness conditions would not lead to separability between the clusters? Specifically, when does the skewness not induce the proximity conditions described in [2]?
2. Could the authors run a comparison of their algorithm vs scikit-learn when both are given the same amount of time to find the solution and then report the minimum of the scikit-learn solutions? I would expect that the solutions are of equal quality in this setting but look forward to being proven wrong. Similarly, could the authors report runtimes of their algorithm vs. scikit-learn on real-world datasets of varying size?
3. Can the authors describe the practical settings where the improved approximation should be required? Specifically, can the authors point to literature from ICLR/ICML/Neurips in the last decade or so where the datasets exhibit the relevant skewness and the improved accuracy was necessary?
4. It would go a long way to show this on additional datasets which satisfy the skewness requirements. Does the difference in solution quality between the $(1 + \varepsilon)$ approximation and the scikit-learn solution remain on other datasets?

---

> ### Author Response · Authors · 2024-11-21
> **Response 1 to Reviewer gzCW**
>
> > The authors' primary contribution is a faster algorithm for obtaining a $(1 + \varepsilon)$-approximation but they have not shown how fast/slow the algorithm is in practice. How long does the algorithm take to run? How does this change as a function of $n$, $k$, or $\varepsilon$? At what dataset-size does it become infeasible to run? The authors only chose a real-world dataset of 400 data samples -- much smaller than would appear in any realistic scenario that the standard ICLR attendee would be operating in.
>
> > The comparison to scikit-learn's clustering lacks depth. Yes -- the authors' $(1+\varepsilon)$-approximation is (as expected) better than the $k$-means++ $\log k$ one. But the scikit-learn algorithm is, I expect, significantly faster. Thus, a fairer comparison would be to time the authors' algorithm on a series of datasets and then repeatedly run the scikit-learn algorithm until the time expires, keeping the minimum cost over these runs. This would actually suggest that the authors' algorithm is a worthwhile use of the runtime in practice. An analysis of this over n would be necessary to show that the authors' proposed algorithm is consistently useful over scikit-learn's kmeans.
>
> Runtime is a crucial metric for illustrating the trade-off between clustering quality and algorithmic efficiency. However, the algorithms with polynomial runtime, eg. Lloyd's heuristic, return a result which is not a $1+\varepsilon$ approximation. On the other hand, the algorithm with $1+\varepsilon$ approximation guarantee, eg. Feldman's algorithm, all have exponential runtime due to the APX-hardness of $(k, z)$-clustering. We do not compare our algorithm with the algorithms like Feldman's one because these algorithms are infeasible for experiments due to their exponential runtime, while we do not compare the runtime with other algorithm like Lloyd's heuristic because these algorithm is faster than our algorithm but can only promise a constant or even worse approximation. Although our algorithm achieves a polynomial runtime, its speed is still not very optimal compared to the popular practical algorithms. We may consider further improve the speed in the future research.
>
> > The choice of datasets feels insufficient. First -- it is not even verified whether the chosen Exasens dataset is heavily skewed. I understand that testing for the skewness technically requires the optimal solution, but I'm sure it can be checked approximately (indeed, if the skewness cannot be verified then it is unclear when one could even apply the proposed algorithms in practice). So is there even a guarantee that the algorithm should work here? Second, why was this the only chosen dataset? In the absence of evidence to the contrary, one might conclude that the Exasens dataset is the only one where there is a significant difference between the authors' proposed algorithm and the scikit-learn approaches.
>
> We have included an additional section in the appendix demonstrating that real-world datasets with significant skewness do exist, which includes Exasens dataset and other datasets. For instance, the dataset of clickstream data for online shopping has $n = 165474$ samples. When $k = 140$, $0.714\%$ most expensive cluster contributes more then $50\%$ of the total cost, and $1.429\%$ most expensive cluster contributes more then $95\%$ of the total cost. In other words, when $k = 140$, it is $(1, 0.5)$-skewed and $(2, 0.99)$-skewed. Thus, our algorithm can outperform algorithms with exponential dependency on $k$ for certain datasets in practice. Consequently, our algorithm can outperform standard algorithms with exponential dependency on $k$ for such datasets in practice.

---

> > ### Author Response · Authors · 2024-11-21
> > **Response 2 to Reviewer gzCW**
> >
> > > Perhaps I'm misunderstanding something, but I get the sense that the skewness notions described by the authors inherently lead to separability between clusters. Namely, in order for a dataset to satisfy the Zipfian distribution, a heavy cluster (with center $c_h$) in the optimal solution must have exponentially higher cost than some subsequent light cluster (with center $c_\ell$). Suppose now that we remove center $c_\ell$ and instead place two centers on $N(c_h)$. $c_h$ was the heavy cluster, so this should induce a large cost-decrease. However, this new partition is by definition not the optimal solution. Thus, the points $N(c_\ell)$ must now have very large cost in this new partition to offset the cost-decrease we obtained. By extension, we can conclude that the variance within $N(c_\ell)$ is small compared to this cluster's distance to the rest of the data. This seems extremely similar to the proximity conditions described in [2].
> >
> > > Based on the above logic, it feels like a theme of related work is missing. Namely, it is known that one can obtain a $(1 + \varepsilon)$ approximation when the data is sufficiently separable [1]. Thus, it would be helpful to highlight when one might expect the skewness to come into play without simultaneously incurring separability conditions under which the optimal solution can be approximated using known techniques.
> >
> > > Can the authors comment on when the skewness conditions would not lead to separability between the clusters? Specifically, when does the skewness not induce the proximity conditions described in [2]?
> >
> > In our study, we explored the relationship between heavy skewness and separability, as intuition suggests that replacing the center of the most expensive cluster with two centers would significantly reduce the cost and potentially induce separability. However, we found no formal guarantee of this relationship, as there is no assurance of how much the cost would decrease through such an action. Therefore, we believe that the connection between skewness and separability remains an open question.
> >
> > > Could the authors run a comparison of their algorithm vs scikit-learn when both are given the same amount of time to find the solution and then report the minimum of the scikit-learn solutions? I would expect that the solutions are of equal quality in this setting but look forward to being proven wrong. Similarly, could the authors report runtimes of their algorithm vs. scikit-learn on real-world datasets of varying size?
> >
> > The runtime of our algorithm is not optimal for practical use. Popular algorithms, such as Lloyd's heuristic, can perform well given sufficient time to solve the problem. While our results currently hold more theoretical than practical value, we believe there is potential for future research to develop increasingly faster algorithms.
> >
> > > Can the authors describe the practical settings where the improved approximation should be required? Specifically, can the authors point to literature from ICLR/ICML/Neurips in the last decade or so where the datasets exhibit the relevant skewness and the improved accuracy was necessary?
> >
> > We also supply extra section in the appendix that exhibit that there exists some datasets with quite heavy skewness in real world. Thus, our algorithm can outperform algorithms with exponential dependency on $k$ for certain datasets in practice.
> >
> > > It would go a long way to show this on additional datasets which satisfy the skewness requirements. Does the difference in solution quality between the $(1 + \varepsilon)$ approximation and the scikit-learn solution remain on other datasets?
> >
> > The difference in solution quality differs for the skewness of the dataset. The difference is larger for the dataset with heavier skewness.

---

> ### Comment · Reviewer_gzCW · 2024-11-22
>
> I hope it's okay that I will respond to both of the authors' comments in this one response.
>
> Unfortunately, the author's responses to the review have not really addressed my concerns. Here are the relevant thoughts:
>
> - Regarding the authors' response on the runtime topic: yes -- I understand that one would not be able to produce a $(1 + \varepsilon)$ approx in the general setting. However, this is not my point.
> Let's suppose we were in the authors' setting: I am a user who wants to run k-means clustering on a skewed dataset and I require a $(1 + \varepsilon)$ approximation (let's even ignore the fact that this setting is contrived and likely does not appear "in the wild"). Then in this setting, would I use the authors' algorithm? As it stands now, I am unconvinced that I would. To be convinced, I would need to understand the following sorts of ideas:
>    - How feasible is the author's algorithm to run on datasets with $n > 1K$?
>    - Would I obtain worse results in practice if I ran k-means++ into Lloyd's repeatedly for the amount of time the authors' algorithms runs for and chose the best clustering? The "min" line in Figure 3 suggests that this would perform well in practice.
>    - At which point does the algorithm by Feldman et al. become impractical? The authors use $k \sim 5$ on the exasens dataset. Then wouldn't the algorithm by Feldman et al. be fine in this setting too? Their exponential dependency is of the form $2^{O(k/ \varepsilon)}$, so I expect that it should be fine at $k=5$...
>    - Also, couldn't I use the algorithm by Feldman et. al with fast-coreset methods to obtain a $(1 + \varepsilon)$ approx for values of k in the range of, say, 10? For example, I could obtain a coreset with size independent of $n$ in $O(nd + nk)$ time [1] and then run Feldman's algorithm over this coreset. Since Feldman's algorithm only exponentially depends on k and epsilon, I do not foresee this being much slower than the authors' method for datasets with reasonable values of $k$ (the standard clustering tasks such as cover-type, MNIST, etc. all have $ k \leq 10$).
> - Although the authors find 2 datasets which exhibit skewness, these are unfortunately irrelevant to the paper due to having $n > 100K$ and $k \sim 100$ -- both much too large for their algorithm to be of practical use for any reasonable value of $\varepsilon$.
>    - It is also unclear to me how the authors even test for the skewness. Doesn't the skewness get evaluated against an optimal solution? If the authors are evaluating the skewness by running k-means++ or Lloyd's, then this suggests that the downstream use of the authors' algorithm is going to lose the theoretical guarantees, no?
> - Furthermore, the discussion around skewness vs. separability still feels incomplete to me. If I place two optimal centers on a "heavy" cluster, one could bound how the cost of that set of points decreases. Since this is not an optimal solution, the "lighter" clusters must have variance/cost that depends on this bound. As a result, some element of skewness -> separability seems inevitable.
> - Lastly, as the authors admit, their algorithm is not "optimal" for practical use. Thus, it is a theoretical algorithm for a problem that feels specific to TCS settings. This is completely fine (and interesting!). But it does therefore feel like this paper belongs in a TCS conference rather than at ICLR.
>
> [1] https://arxiv.org/abs/2210.08361

---

> ### Author Response · Authors · 2024-11-25
>
> Thank you for the continued correspondence. We offer some responses to your questions and would be happy to discuss additional concerns.
>
> > Regarding the authors' response on the runtime topic: yes -- I understand that one would not be able to produce a $(1+\varepsilon)$ approx in the general setting. However, this is not my point. Let's suppose we were in the authors' setting: I am a user who wants to run k-means clustering on a skewed dataset and I require a $(1+\varepsilon)$ approximation (let's even ignore the fact that this setting is contrived and likely does not appear "in the wild"). Then in this setting, would I use the authors' algorithm?
>
> We emphasize that our techniques are more general than $k$-means. For example, our approach can handle general $(k,z)$-clustering such as $k$-median, for which kmeans++ and Lloyd's algorithm would not work. Another example is the $k$-medoids problem when the centers must be placed at one of the data points, which is more appropriate for datasets that contain noise or outliers.
>
> We compared the performance of our naive algorithm with *KMedoids* from *scikit-learn-extra*, given the **same amount of runtime** for each algorithm. Our experimental results demonstrate a difference between our algorithm and *KMedoids* from $1.10 \%$ at $k = 4$ up to of $5.89 \%$ at $k = 7$. In another setting, the difference was as small as $0.8\%$. Thus our proof-of-concept implementation is surprisingly competitive with the highly optimized *scikit* library on a standard dataset. We hope that either simple optimizations or more skewed datasets can further complement our theoretical results.
>
> > How feasible is the author's algorithm to run on datasets with $n > 1K$?
>
> Our algorithm is quite feasible for $n > 1K$ and in fact, one of the datasets used in our experiments has a size of $n = 6000$.
>
> > Would I obtain worse results in practice if I ran k-means++ into Lloyd's repeatedly for the amount of time the authors' algorithms runs for and chose the best clustering? The "min" line in Figure 3 suggests that this would perform well in practice.
>
> Right, Lloyd's algorithm with $k$-means++ is quite popular in practice and indeed performs well on many "real-world" datasets that exhibit "average-case" behavior. Indeed, we performed a number of additional experiments and can confirm that on standard datasets that are not highly skewed, k-means++ into Lloyd's performs better than our algorithm. However, it is important to note that Lloyd's algorithm is nevertheless a heuristic, and there are "worst-case" datasets in which it fails catastrophically, e.g., datasets with highly elongated or irregularly shaped clusters where the algorithm may converge to poor local minima, or datasets with overlapping clusters where the choice of initial centroids significantly impacts the results.
>
> > At which point does the algorithm by Feldman et al. become impractical? The authors use $k \sim 5$ on the exasens dataset. Then wouldn't the algorithm by Feldman et al. be fine in this setting too? Their exponential dependency is of the form $2^{O(k/ \varepsilon)}$, so I expect that it should be fine at $k=5$...
>
> > Also, couldn't I use the algorithm by Feldman et. al with fast-coreset methods to obtain a $(1 + \varepsilon)$ approx for values of k in the range of, say, 10? For example, I could obtain a coreset with size independent of $n$ in $O(nd + nk)$ time [1] and then run Feldman's algorithm over this coreset. Since Feldman's algorithm only exponentially depends on k and epsilon, I do not foresee this being much slower than the authors' method for datasets with reasonable values of $k$ (the standard clustering tasks such as cover-type, MNIST, etc. all have $ k \leq 10$).
>
> Unfortunately, due to the time constraints, we have not yet been able to implement a version of Feldman's algorithm that completes on our datasets with $k=5$. One difficulty is that the additional factor of $k$ is in the exponent, rather than the polynomial. In fact, our algorithm is still feasible for $n = 6000$ and $k = 20$, which may be significantly more challenging for Feldman's algorithm. Morevoer, we remark that Feldman's algorithm only works for $k$-means, but our algorithm works for general $(k, z)$-clustering, including both $k$-median and $k$-medoids.

---

> > ### Comment · Reviewer_gzCW · 2024-11-25
> >
> > Thanks for the updates! Below are some thoughts in no particular order:
> >
> > - To be clear, kmeans++ *does* work for k-median. The log-k approximation remains the same as can be verified in the original paper [1].
> > - Thank you for the updated experiments. I do have to say that I'm a bit concerned about the fact that the skewness has to be measured with Lloyd's algorithm. The remaining experimental results are interesting and I appreciate the authors taking the time to run them.
> > - I disagree that replacing the heaviest center by two centers will reduce the cost by a fixed constant. Take, for example, the dataset of n points in 2-dimensional space and let k=2. For this dataset, let one point be an extreme outlier at (10^8, 0) while the remaining n-1 points are all distributed normally around the origin. Clearly, the k=2 clustering will put one center at the origin and another at the outlier. Thus, the entire cost comes from the heaviest cluster at the origin. If we now replace this heavy cluster with 2 clusters, our cost decreases multiplicatively (roughly by the order of 2).
> >
> >
> >
> >
> > [1]: Arthur, Vassilvitskii; "kmeans++: The Advantages of Careful Seeding"

---

> > > ### Author Response · Authors · 2024-11-25
> > >
> > > Again, thank you for following up, we really appreciate the interest and the opportunity to continue discussing concerns about our work!
> > >
> > > > To be clear, kmeans++ does work for k-median. The log-k approximation remains the same as can be verified in the original paper [1].
> > >
> > > Yes, we agree that the initial seeding still works for k-median, and furthermore that Lloyd's algorithm can be modified to choose the $(1,z)$-center of the points assigned to the cluster, rather than the centroid. This calculation can be approximated by some stochastic optimization process, but we hope that there is potential benefit in having an additional unified framework such as ours.
> > >
> > > > I do have to say that I'm a bit concerned about the fact that the skewness has to be measured with Lloyd's algorithm.
> > >
> > > Yes, we agree. It would certainly be more ideal if either 1) skewness did not empirically need to be measured by an approximation algorithm or 2) the theoretical guarantees also hold for approximately optimal solutions that exhibit skewness.
> > >
> > > > I disagree that replacing the heaviest center by two centers will reduce the cost by a fixed constant. Take, for example, the dataset of n points in 2-dimensional space and let k=2. For this dataset, let one point be an extreme outlier at (10^8, 0) while the remaining n-1 points are all distributed normally around the origin. Clearly, the k=2 clustering will put one center at the origin and another at the outlier. Thus, the entire cost comes from the heaviest cluster at the origin. If we now replace this heavy cluster with 2 clusters, our cost decreases multiplicatively (roughly by the order of 2).
> > >
> > > Ah sorry for the confusion, we meant the cost would improve **multiplicatively** by some constant in the worst case, whereas the existing PTAS for separability require the cost to improve **multiplicatively** by some function of $\frac{1}{\varepsilon}$.

---

> ### Author Response · Authors · 2024-11-25
>
> > Although the authors find 2 datasets which exhibit skewness, these are unfortunately irrelevant to the paper due to having $n > 100K$ and $k \sim 100$ -- both much too large for their algorithm to be of practical use for any reasonable value of $\varepsilon$.
>
> Yes, we tested roughly 20 large real-world datasets to evaluate their skewness parameters and found that these two datasets exhibit significantly heavy skewness. Since our algorithm is feasible for our synthetic dataset with size $n=6000$, we hope that our algorithm can handle substantially larger datasets after optimizing the code, since the dominating term in our runtime is $(k\log n)^{\tilde{O}(s+1/\varepsilon)}$ as compared to the previous PTAS with runtime $2^{\tilde{O}(k/\varepsilon)}$.
>
> > It is also unclear to me how the authors even test for the skewness. Doesn't the skewness get evaluated against an optimal solution? If the authors are evaluating the skewness by running k-means++ or Lloyd's, then this suggests that the downstream use of the authors' algorithm is going to lose the theoretical guarantees, no?
>
> Yes, we use Lloyd's heuristic with $k$-means++ to test the skewness of the dataset. We agree this does not precisely measure the skewness of a dataset since it does not return the optimal solution. However, it can still reflect the dataset's skenewss to some extent. Since we define the skewness of a dataset by two parameters $s$ and $\varepsilon$, for instance, a dataset can be both $(1, 0.7)$-skewed and $(3, 0.95)$-skewed, which means there is a trade-off between $s$ and the accuracy $\varepsilon$. Therefore in implementation, we can fix values of $s$ and $t$, which is the parameter for the number of swaps, and the skewness will affect the accuracy for the final result.
>
> > Furthermore, the discussion around skewness vs. separability still feels incomplete to me. If I place two optimal centers on a "heavy" cluster, one could bound how the cost of that set of points decreases. Since this is not an optimal solution, the "lighter" clusters must have variance/cost that depends on this bound. As a result, some element of skewness -> separability seems inevitable.
>
> Ah yes, that's a great observation that we missed in the previous exchange, thanks for pointing this out! Indeed it does seem that skewness implies some amount of separability. However, it should be noted that replacing an optimal center in the "heaviest" cluster with two optimal centers is only guaranteed to improve the clustering cost by at most some fixed constant. By comparison, existing results such as [ABS10, BMO+11] require the clustering cost to improve by some fixed function of $\frac{1}{\varepsilon}$ to achieve a PTAS. We will add this to the discussion in the next version of the manuscript. Thanks again for bringing this up!
>
> [ABS10] Pranjal Awasthi, Avrim Blum, Or Sheffet: Stability Yields a PTAS for k-Median and k-Means Clustering. FOCS 2010: 309-318
>
> [BMO+11] Vladimir Braverman, Adam Meyerson, Rafail Ostrovsky, Alan Roytman, Michael Shindler, Brian Tagiku: Streaming k-means on Well-Clusterable Data. SODA 2011: 26-40

---

### Official Review · Reviewer_eDaF · 2024-10-29

**Soundness:** 3
**Presentation:** 4
**Contribution:** 3
**Rating:** 8
**Confidence:** 4

**Summary:**

The paper presents algorithms for k-means, k-median etc that achieve an (1+eps)-approx. The authors introduce the notion of (s,1-\eps)-skewness. A dataset has this skewness if there exists an optimal clustering that using s \leq k cluster-points that accounts for 1-\eps of the cost.
In that case they achieve a runtime that can be even polynomial assuming s is a constant or close to one. Note that the problem in general is APX-hard meaning a (1+eps) approximation is not possible in polynomial time assuming P != NP.

The authors study different versions (continuous and discrete clustering) and prove rigorously their results. They supplement their theory results with some experimental results that show that often their clustering is better than commonly used algorithms.

**Strengths:**

Interesting model

Strong results

Non-trivial math

Experimental results show that the obtained results are actually good

Well-written

**Weaknesses:**

The crucial assumption is not validated: it would be nice to have some experiments (for small datasets) that suggest that for small s one can actually get (s,1+eps) skewness (for real-world-data).

The run-time of the algorithms in the experiments is not discussed.

The results are very bad for large s, this needs to be highlighted. A better comparison with the SOA would be good here.

Other baseline algorithm could have been used in the experiments.

**Questions:**

What was the result in the experiments you got for other baseline algorithms?

---

> ### Author Response · Authors · 2024-11-21
> **Response to Reviewer eDaF**
>
> > The crucial assumption is not validated: it would be nice to have some experiments (for small datasets) that suggest that for small s one can actually get (s,1+eps) skewness (for real-world-data).
>
> We have included an additional section in the appendix that exhibit that there exists some datasets with quite heavy skewness in real world. Thus, our algorithm can outperform algorithms with exponential dependency on $k$ for certain datasets in practice.
>
> > The runtime of the algorithms in the experiments is not discussed.
>
> Runtime is a crucial metric for illustrating the trade-off between clustering quality and algorithmic efficiency. However, the algorithms with polynomial runtime, eg. Lloyd's heuristic, return a result which is not a $1+\varepsilon$ approximation. On the other hand, the algorithm with $1+\varepsilon$ approximation guarantee, eg. Feldman's algorithm, all have exponential runtime due to the APX-hardness of $(k, z)$-clustering. We do not compare our algorithm with the algorithms like Feldman's one because these algorithms are infeasible for experiments due to their exponential runtime, while we do not compare the runtime with other algorithm like Lloyd's heuristic because these algorithms are faster than our algorithm but can only promise a constant or even worse approximation.
>
> > The results are very bad for large s, this needs to be highlighted. A better comparison with the SOA would be good here.
>
> The state-of-the-art algorithm in theoreitcal side is Fledman's algorithm we mentioned in the paper. Their algorithm has a polynomial dependency in $n$ but an exponential dependency in $k$. The state-of-the-art algorithm in practical side is Lloyd's heuristic, which has a worse accuracy guarantee. We run experiments in our paper to compare its accuracy with our algorithm.
>
> > Other baseline algorithm could have been used in the experiments.
>
> We have supplied an extra section in the appendix to compare our algorithm with standard local search. We apply the experiments with the same datasets and parameters as the experiments we run for Lloyd's heuristic.
>
> For synthetic data, our experiments illustrate an improvement range for $k$-means from $11.54$% at $k=4$ for the minimum metric to $54.87$% at $k=10$ for the median metric, and for $k$-medoids from $6.06$% at $k=5$ for the minimum metric to $31.86$% at $k=7$ for the average metric. For real data, our experimental results demonstrate an enhancement range for $k$-means from $87.23$% at $k=4$ for the minimum metric up to $95.77$% at $k=10$ for the median metric, and for $k$-medoids from $6.63$% at $k=7$ for the minimum metric to $40.60$% at $k=10$ for the median metric. This overall improvement highlights the superior accuracy performance of our algorithm relative to local search, across various metrics including average, minimum, and median.

---

> > ### Comment · Reviewer_eDaF · 2024-11-22
> >
> > I had a look at the new results and I have to say I really appreciate these new results. However, they seem to confirm my fear that s needs to be linear in k for strong results. Nonetheless, I still think it's a very nice paper and I've updated my score to accept the paper (although I'm actual score is more of 7.7). Should it not make the cut, then I recommend finding (synthetic) datasets where a sublinear s is fine.

---

### Official Review · Reviewer_xMdu · 2024-11-03

**Soundness:** 3
**Presentation:** 3
**Contribution:** 2
**Rating:** 5
**Confidence:** 4

**Summary:**

This paper studies the popular $(k, z)$-clustering problem in both discrete and continuous settings. The $(k, z)$-clustering problem has been widely studied in the field of machine learning, where the goal is to identify a set of centers with size at most $k$ while minimizing the sum of the distances from the data points to the centers selected. For the $(k, z)$-clustering problem, several theoretical lower bounds for approximation guarantees have been established, where one cannot even achieve a PATS within polynomial running time (i.e., the $(k, z)$-clustering problem is APX-hard). To break the theoretical lower bound and further narrow the gap between theory and practice, this paper introduces the notion of "skewness" for the clustering problem. Intuitively speaking, a give clustering dataset is skew if the clustering cost of several optimal clusters take a large fraction of the overall optimal clustering cost. To measure the skewness of a given datasets, this paper proposes the notion of $(s, 1-\epsilon)$-skewed dataset, where the clustering cost of the top $s$ optimal clusters take a fraction of at least $(1-\epsilon)$ of the optimal clustering cost.  Based on the proposed notion of skewed dataset, this paper proposes coreset and potential set construction methods. The coreset construction method can reduce the number of data points significantly while maintaining the skewness of the given dataset while the potential set construction method reduces the number of potential centers for enumeration based on $O(\epsilon)$-net construction method. Then, this paper proposes new local search methods such that the proposed methods can achieve $(1+\epsilon)$-approximation for any given dataset satisfying the skewness properties. The high-level idea behind the local search methods is to handle skewed optimal clusters and other optimal clusters separately. By enumerating for the (approximately) optimal clustering centers for the $s$ skewed optimal clusters and fixed them, the proposed local search algorithm only needs to achieve a convergence on the remaining $k-s$ clustering centers using exhaustive local search strategies. For discrete setting, the proposed local search algorithm should enumerate for all the subsets of the given dataset with sizes $s$ to find the optimal clustering centers. However, in continuous settings, coreset and potential center set construction methods can be applied to reduce the enumerations to accelerate the local search process. Based on the techniques above, this paper obtains PTAS for the $(k, z)$-clustering problem in discrete settings with running time $(nk/\epsilon)^{O(s+1/\epsilon)}$, where $s$ is the number of skewed optimal clusters. For continuous settings, an improved running time of $\tilde{O}(nk) + (klogn)^{\tilde{O}(s+1/\epsilon)}$ can be obtained. Empirical experiments show that the proposed local search methods achieve better performances compared with Lloyd-type clustering algorithms.

**Strengths:**

1. This paper establishes PTAS for the $(k, z)$-clustering problem by leveraging specific data distribution assumptions on the skewness of the data points. This advancement in approximation is a clear advantage of the proposed methods, marking a significant contribution to the field of clustering and machine learning.

2. The proposed data reduction and local search techniques are new and interesting, offering potential enhancements to the well-studied local search methods with tight theoretical lower bound of $(9+\epsilon)$.

3. The proposed data distribution assumption is shown to be practical, as data points generated from a Zipfian distribution naturally exhibit the skewness property.

**Weaknesses:**

1. While the introduced concept of skewness facilitates the design of approximation algorithms capable of achieving $(1+\epsilon)$-approximation guarantees for clustering quality, the time complexity of the proposed local search-based methods can grow significantly if the number $s$ of skewed optimal clusters is large. Furthermore, there are several existing local search algorithms that can be executed in linear running time with respect to the data size (such as those in [1]-[3]). Consequently, the proposed methods in this paper may have running time considerably slower than those linear-time local search methods even when there are only a few skewed optimal clusters. In practical applications, the number of skewed optimal clusters may indeed be large. The main advantage of this paper is the approximation guarantees on clustering quality. However, to achieve better clustering quality guarantee, a significant sacrifice is made in terms of time complexity. This paper lacks a detailed examination of cases with numerous skewed clusters and instead relies on exhaustive enumeration for these clusters, which may limit the scalability of the approach for large-scale clustering instances. This paper also lacks discussions on the trade-off between  the time complexities and the theoretical clustering quality guarantees. It is recommended that the authors should include a more detailed analysis or discussion of how the algorithm's performance changes as the number of skewed clusters increases.

2. In the experimental section, this paper primarily compares the proposed local search methods with existing Lloyd-type clustering algorithms, but lacks comparisons with other existing local search methods. Previous work has shown that while these local search algorithms may not achieve strong theoretical approximation guarantees on clustering quality (e.g., $(1+\epsilon)$-approximation), their empirical performance often yields clustering costs close to the optimal. Adding comparisons between the proposed methods and existing local search algorithms would provide a more comprehensive evaluation of their practical effectiveness. It is recommended that the authors should add more comparisons between the proposed local search algorithms and the $(9+\epsilon)$-approximation multi-swap local search algorithm and other sampling-based local search algorithms in [1]-[3].

3. To verify the effectiveness of the proposed method, this paper should compare the proposed method with other clustering algorithms that can achieve near-optimal clustering costs (such as the branch and bound method given in [4]).

4. The experiments lack a comparison of the running times across different clustering algorithms. Running time is a crucial metric for illustrating the trade-off between clustering quality and algorithmic efficiency, and its inclusion would provide a more balanced evaluation of the proposed methods.


[1] Beretta L, Cohen-Addad V, Lattanzi S, et al. Multi-swap k-means++[C]//Proceedings of the 37th International Conference on Neural Information Processing Systems. 2023: 26069-26091.

[2] Fan C, Li P, Li X. LSDS++: Dual sampling for accelerated k-means++[C]//International Conference on Machine Learning. PMLR, 2023: 9640-9649.

[3] Lattanzi S, Sohler C. A better k-means++ algorithm via local search[C]//International Conference on Machine Learning. PMLR, 2019: 3662-3671.

[4] Ren J, Hua K, Cao Y. Global optimal K-medoids clustering of one million samples[C]//Proceedings of the 36th International Conference on Neural Information Processing Systems. 2022: 982-994.

**Questions:**

Part of the Questions have been given in the Weakness part. Below are some further questions.

1. Does the proposed PTAS framework rely heavily on the local search strategy? For $s$ fixed skewed clusters, is it possible to use other clustering techniques (such as linear-programming rounding and sampling-based methods) to find the remaining $k-s$ clustering centers to achieve $(1+\epsilon)$-approximation?

2. In experiments, how to choose the number of $s$ for a specific dataset. From the paper, $s$ is fixed as $1$. What are the performances for other choices of $s$?

3. In experiments, the swap size $t$ is also fixed as $1$ instead of $O(1/\epsilon)$ as stated in the algorithm and the theoretical analysis. Please explain.

4. According to the local search algorithms proposed in this paper, the time complexity for finding skewed optimal clusters increase sharply as the number of skewed clusters increase. The authors should add a detailed discussion on the trade-off between the time complexity and the approximation guarantees regarding the number $s$ of skewed optimal clusters. Furthermore, the authors should explore or discuss potential optimizations for handling cases with many skewed clusters, rather than relying solely on exhaustive enumeration.

5. Since the main contribution of this paper is the improvements on clustering quality guarantee, why don't the authors include comparisons against methods like the branch bound and linear programming methods? These methods can give references of optimal clustering costs and how close is the proposed local search methods to the optimal solutions.

6. Adding specific metrics or visualizations would be helpful for comparing running times, such as scaling plots showing how runtime grows with dataset size or number of clusters for different algorithms.

**Details Of Ethics Concerns:**

Since this is mainly a theoretical result, there is no ethical issues that should be considered.

---

> ### Author Response · Authors · 2024-11-21
> **Response 1 to Reviewer xMdu**
>
> > While the introduced concept of skewness facilitates the design of approximation algorithms capable of achieving $(1+\epsilon)$-approximation guarantees for clustering quality, the time complexity of the proposed local search-based methods can grow significantly if the number $s$ of skewed optimal clusters is large. Furthermore, there are several existing local search algorithms that can be executed in linear running time with respect to the data size (such as those in [1]-[3]). Consequently, the proposed methods in this paper may have running time considerably slower than those linear-time local search methods even when there are only a few skewed optimal clusters. In practical applications, the number of skewed optimal clusters may indeed be large. The main advantage of this paper is the approximation guarantees on clustering quality. However, to achieve better clustering quality guarantee, a significant sacrifice is made in terms of time complexity. This paper lacks a detailed examination of cases with numerous skewed clusters and instead relies on exhaustive enumeration for these clusters, which may limit the scalability of the approach for large-scale clustering instances. This paper also lacks discussions on the trade-off between the time complexities and the theoretical clustering quality guarantees. It is recommended that the authors should include a more detailed analysis or discussion of how the algorithm's performance changes as the number of skewed clusters increases.
>
> Although our algorithm does not outperform standard algorithms with exponential dependency on $k$ when the dataset has a large $s$, we emphasize that due to the APX-hardness of the $(k, z)$-clustering problem, no algorithm can universally outperform standard algorithms with exponential dependency on $k$ in the general case.
>
> This limitation motivates our focus on datasets with heavy skewness, where the underlying structure can be leveraged to achieve polynomial runtime in certain settings. To support this, we have ran experiments to evaluate the skewness of various datasets. We have found many datasets have certain extent of skewness. For instance, more than $30$% datasets are at least $(0.3k, 0.5)$-skewed, which means $30$% of the clusters contributes more than a half of the total cost. Although such skewness is not extremely heavy, our algorithm can still provide certain extent of improvement in runtime compared to the standard algorithm with exponential dependacy on $k$.
>
> Furthermore, there are some datasets with even heavier skewness. We have included an additional section in the appendix demonstrating the $10$% most heavily skewed real-world datasets that we have evaluated. For instance, the dataset of clickstream data for online shopping has $n = 165474$ samples. When $k = 140$, $0.714$% most expensive cluster contributes more than $50$% of the total cost, and $1.429$% most expensive cluster contributes more than $95$% of the total cost. In other words, when $k = 140$, the dataset is $(1, 0.5)$-skewed and $(2, 0.99)$-skewed. Consequently, our algorithm can outperform standard algorithms with exponential dependency on $k$ for such datasets in practice.
>
> > To verify the effectiveness of the proposed method, this paper should compare the proposed method with other clustering algorithms that can achieve near-optimal clustering costs (such as the branch and bound method given in [4]).
>
> We have checked [4], but unfortunately, the algorithm in [4] converges to the optimal solution of Lagrangian relaxation of the $k$-medoids problem rather than the optimal solution of $k$-medoids itself. The optimal solution of Lagrangian relaxation of the $k$-medoids problem is a $1/e$ approximation of the optimal solution of k-medoids. Hence the algorithm in [4] is a polynomial time algorithm returning a constant approximation of $k$-medoids rather than a $1+\varepsilon$ approximation. In fact, it is impossible to propose a polynomial time algorithm that returns a $1+\varepsilon$ approximation due to the APX-hardness of $(k, z)$-clustering.

---

> ### Author Response · Authors · 2024-11-21
> **Response 2 to Reviewer xMdu**
>
> > The experiments lack a comparison of the running times across different clustering algorithms. Running time is a crucial metric for illustrating the trade-off between clustering quality and algorithmic efficiency, and its inclusion would provide a more balanced evaluation of the proposed methods.
>
> > Adding specific metrics or visualizations would be helpful for comparing running times, such as scaling plots showing how runtime grows with dataset size or number of clusters for different algorithms.
>
> Runtime is a crucial metric for illustrating the trade-off between clustering quality and algorithmic efficiency. However, the algorithms with polynomial runtime, e.g., Lloyd's heuristic or the ones in [1,2,3,4], return a result which is not a $1+\varepsilon$ approximation. On the other hand, the algorithm with $1+\varepsilon$ approximation guarantee, eg. Feldman's algorithm, all have exponential runtime due to the APX-hardness of $(k, z)$-clustering, and thus are infeasible to perform experimental comparisons. We have included an additional section in Appendix G of the updated version of the manuscript to compare our algorithm with standard local search. We apply the experiments with the same datasets and parameters as the experiments we run for Lloyd's heuristic.
>
> For synthetic data, our experiments illustrate an improvement range for $k$-means from $11.54$% at $k=4$ for the minimum metric to $54.87$% at $k=10$ for the median metric, and for $k$-medoids from $6.06$% at $k=5$ for the minimum metric to $31.86$% at $k=7$ for the average metric. For real data, our experimental results demonstrate an enhancement range for $k$-means from $87.23$% at $k=4$ for the minimum metric up to $95.77$% at $k=10$ for the median metric, and for $k$-medoids from $6.63$% at $k=7$ for the minimum metric to $40.60$% at $k=10$ for the median metric. This overall improvement highlights the superior accuracy performance of our algorithm relative to local search, across various metrics including average, minimum, and median.
>
> > Does the proposed PTAS framework rely heavily on the local search strategy? For $s$ fixed skewed clusters, is it possible to use other clustering techniques (such as linear-programming rounding and sampling-based methods) to find the remaining $k-s$ clustering centers to achieve $(1+\epsilon)$-approximation?
>
> The algorithm of our paper relies heavily on the local search strategy. However, it may be possible to use other clustering techniques (such as linear-programming rounding and sampling-based methods) to find the remaining $k-s$ clustering centers to achieve $(1+\epsilon)$-approximation. We agree this is a good direction for future study.
>
> > In experiments, how to choose the number of $s$ for a specific dataset. From the paper, $s$ is fixed as $1$. What are the performances for other choices of $s$?
>
> Usually, we cannot have an accurate estimation of the specific value of $s$ unless we have the optimal solution. Fortunately, we can still apply our algorithm by using the following possible methods. The first method is to apply some fast algorithm like Lloyd heuristic with k-means++ to evaluate the size of $s$. Although there does not exist accuracy guarantee for such evaluation, it usually performs well in practice. The second method is to fix the value of $s$. Since we define the skewness of a dataset by two parameter $s$ and $\varepsilon$, for instance, a dataset can be both $(1, 0.7)$-skewed and $(3, 0.95)$-skewed, which means we can make trade-off between $s$ and the accuracy $\varepsilon$. Therefore, we can just fix the value of $s$ without evaluating it, and the skewness will just affect the accuracy for the final result.
>
> > In experiments, the swap size $t$ is also fixed as $1$ instead of $O(1/\epsilon)$ as stated in the algorithm and the theoretical analysis. Please explain.
>
> The swap size $t$ is dependent on the accuracy $\varepsilon$. We choose $t = 1$ in the experiment for simplicity. Other choice of $t$ is also feasible, and will induce a trade-off between accuracy and runtime.
>
> > According to the local search algorithms proposed in this paper, the time complexity for finding skewed optimal clusters increase sharply as the number of skewed clusters increase. The authors should add a detailed discussion on the trade-off between the time complexity and the approximation guarantees regarding the number $s$ of skewed optimal clusters. Furthermore, the authors should explore or discuss potential optimizations for handling cases with many skewed clusters, rather than relying solely on exhaustive enumeration.
>
> Our contribution of this paper is to propose an algorithm that can return a $1+\varepsilon$ approximation in polynomial time, which breaks the previous exponential time bound. The brute-force search in the algorithm increases the runtime significantly. We may consider improve the algorithm to reduce the runtime in the future study.

---

> > ### Author Response · Authors · 2024-11-21
> > **Response 3 to Reviewer xMdu**
> >
> > > Since the main contribution of this paper is the improvements on clustering quality guarantee, why don't the authors include comparisons against methods like the branch bound and linear programming methods? These methods can give references of optimal clustering costs and how close is the proposed local search methods to the optimal solutions.
> >
> > These method cannot return a $1+\varepsilon$ approximation and only return a constant approximation for the optimal cost.

---

> ### Comment · Reviewer_xMdu · 2024-11-22
> **Response to the Authors**
>
> After reviewing the authors' responses, my concerns persist.
>
> The key strength of this paper lies in its clustering quality guarantees (i.e., $(1+\epsilon)$). However, strong theoretical guarantees may lose impact if they are not validated through comparisons with existing algorithms that achieve near-optimal clustering costs in practice. While the authors argue that the algorithm by Ren et al. only offers constant approximation, experimental results from Ren et al. demonstrate that their method achieves a mere 0.8% gap to the optimal clustering cost when centers are restricted to the given dataset (the discrete version considered in this paper). Including comparisons with such algorithms would make the results of the proposed PTAS more compelling and convincing.
>
> From a practical perspective, the proposed algorithm also presents significant challenges in implementation. The skewness parameter $s$ and the swap size $t$ are difficult to determine effectively, which complicates its practical applications. Additionally, runtime is a critical factor, especially for large-scale datasets. If the proposed method does not scale well for larger datasets, its practical contribution may be influenced. For smaller datasets, robust heuristic methods (e.g., [1]) can already deliver satisfactory results, potentially undermining the necessity of the proposed approach for these cases.
>
> Overall, while the paper makes  theoretical contributions, its practical value appears limited. I encourage the authors to better articulate the practical implications of their work. Specifically, it would be helpful to clarify:
>
> The largest dataset sizes the algorithm can effectively handle.
> Report the running time for handling small datasets.
> A runtime analysis showing how the algorithm scales with increasing dataset sizes, particularly for small-scale datasets.
> These additions could provide a clearer picture of the algorithm's utility in real-world scenarios.
>
> Reference
> [1] Conrads T, Drexler L, Könen J, et al. Local Search k-means++ with Foresight[J]. arXiv preprint arXiv:2406.02739, 2024.

---

> ### Author Response · Authors · 2024-11-25
>
> Thank you for the follow-up. We appreciate the continued discussion and believe the resulting product will be significantly improved as a result of the review process.
>
> > The key strength of this paper lies in its clustering quality guarantees (i.e., ). However, strong theoretical guarantees may lose impact if they are not validated through comparisons with existing algorithms that achieve near-optimal clustering costs in practice. While the authors argue that the algorithm by Ren et al. only offers constant approximation, experimental results from Ren et al. demonstrate that their method achieves a mere 0.8% gap to the optimal clustering cost when centers are restricted to the given dataset (the discrete version considered in this paper). Including comparisons with such algorithms would make the results of the proposed PTAS more compelling and convincing.
>
> > Additionally, runtime is a critical factor, especially for large-scale datasets. If the proposed method does not scale well for larger datasets, its practical contribution may be influenced. For smaller datasets, robust heuristic methods (e.g., [1]) can already deliver satisfactory results, potentially undermining the necessity of the proposed approach for these cases.
>
> Thanks for the suggestion, we performed additional experiments comparing the performance of our naive algorithm with *KMedoids* from *scikit-learn-extra*, given the **same amount of runtime** for each algorithm. Our results showed that our simple proof-of-concept was surprisingly competitive with the highly optimized *scikit* library on a standard dataset, also achieving a $0.8$% gap, and in a more comprehensive trial, from $1.10 \%$ at $k = 4$ up to of $5.89 \%$ at $k = 7$.  Therefore, we hope that either simple optimizations or more skewed datasets can further complement our theoretical results.
> Moreover, our approach presents a unified framework for general $(k,z)$-clustering such as $k$-median, whereas existing implementations require separate libraries for k-median, k-means, and k-medoids.
>
> Since we define the skewness of a dataset by two parameters $s$ and $\varepsilon$, for instance, a dataset can be both $(1, 0.7)$-skewed and $(3, 0.95)$-skewed, which means there is a trade-off between $s$ and the accuracy $\varepsilon$. Therefore in implementation, we can fix values of $s$ and $t$, which is the parameter for the number of swaps, and the skewness will affect the accuracy for the final result.
>
> > The skewness parameter $s$ and the swap size $t$ are difficult to determine effectively, which complicates its practical applications.
>
> > The largest dataset sizes the algorithm can effectively handle. Report the running time for handling small datasets. A runtime analysis showing how the algorithm scales with increasing dataset sizes, particularly for small-scale datasets. These additions could provide a clearer picture of the algorithm's utility in real-world scenarios.
>
> We remark that $s$ and $t$ are input parameters, which may be determined as design choices/hyperparameters, and the skewness will affect the accuracy in the final output. Thus while $s$ and $t$ parameterize the theoretical guarantees, they are not necessary calculations in practice. Hence, we can handle massively large datasets for small input values of $s$ and $t$ (though in all fairness, there is a corresponding degradation in the resulting accuracy).

---

### Official Review · Reviewer_Aqmz · 2024-11-04

**Soundness:** 2
**Presentation:** 2
**Contribution:** 2
**Rating:** 5
**Confidence:** 4

**Summary:**

This paper studies the classic $k$-clustering problem. In this problem, given a dataset of size $n$ and a positive integer $k$, the objective is to identify up to $k$ points in the space as clustering centers while minimizing the connection loss between the data points and the centers selected. For the clustering problem, local search methods are one of the most widely used techniques in practice. However, a challenging issue for the local search algorithms is that the best approximation ratio for the local search methods within polynomial running time is $(9+\epsilon)$, which has also been proved to be tight. To further narrow the theoretical guarantees on approximation ratio for the local search methods, this paper introduces the concept of "skew datasets" and presents a new local search method achieving a $(1 + \epsilon)$-approximation under skewness assumptions. The proposed approach for the continuous space involves three main steps: (1) Coreset Construction; (2) Potential Center Construction; (3) Heavy Skewed Local Search. The first two steps ensure efficient data reduction while maintaining the skewness on the reduced dataset. In the final step, $s$ center points (for finding skewed clusters) are fixed, and local search is performed on the remaining $k - s$ potential center points. Additionally, this paper also conducts empirical experiments on synthetic and real-world datasets to demonstrate the effectiveness of the proposed local search method.

**Strengths:**

The strengths of this paper can be summarized as follows.

1. Under the notion of skewness, the proposed local search method can break the approximation lower bound of $9+\epsilon$, achieving near-optimal approximation guarantees on clustering quality.

2. The skewness of a dataset can hold for several commonly used data distributions, such as Zipfian distribution.

3. Experiments show that the proposed method can achieve better performance compared with other clustering algorithms.

**Weaknesses:**

The weakness of this paper can be summarized as follows.

1. The proposed method relies heavily on the dataset distribution, particularly requiring $s$ (the number of heavily skewed clusters) to be sufficiently small. In practice, this condition might not always hold for many real-world or synthetic datasets.

2. Additionally, the paper does not provide an effective method for calculating or estimating the specific value of $s$ (the number of heavily skewed clusters) for a given dataset, which limits the feasibility of executing the algorithm’s searching process for heavily skewed optimal clustering centers.

3. In the experimental section, the paper lacks a comparison between the proposed algorithm and standard local search methods, and does not analyze influence of different choices for the number of skewed clusters (i.e., the number of $s$).

**Questions:**

Q1. The theoretical results of this paper for Euclidean space heavily relies on the coreset construction methods. With the coreset construction methods, one can also obtain a $(1+\epsilon)$-approximation by executing a weighted PTAS on the coreset constructed without any assumptions on the data distributions (such as the skewness). Can the authors give detailed comparisons between the proposed method in this paper and directly applying a weighted PTAS after the coreset construction?

Q2. The results of this paper heavily rely on the skewness of the given clustering instance. The proposed method needs to enumerate all subsets with sizes smaller than $s$ to find the optimal clustering centers for the heavily skewed clusters, where the running time can be very large if $s$ is large. Directly using brute-force enumerations may lead to exponential running time. How to handle the case when $s$ is large?

Q3. What are the performances of the proposed method compared with the existing local search methods? The authors should add more experimental results to give clearer comparisons with the state-of-the-art local search algorithms.

Other Comments:
- page 3, line 161, it should specifically explain the reasoning for $\Delta = polyn$ and whether it relates to the aspect ratio.

- page 4, line 208, the specific definition of sensitivity is provided, but Algorithm 1 does not specify how to calculate the sensitivity for each $x \in X$. A specific or approximate calculation method should be provided.

- page 15, line 793, the conclusion “with probability at least 0.99” relies on fixed $\epsilon$ and $\gamma$. However, the theorem does not reflect this property. The theorem should specify approximate values for $\epsilon$ and $\gamma$, or rewrite the probability in a form related to $\epsilon$ and $\gamma$. Similar issues exist for probability descriptions in other theorems throughout the paper.

- page 15, line 796, $(\mathbb{R}^d)^k$ is not defined.

---

> ### Author Response · Authors · 2024-11-21
> **Response 1 to Reviewer Aqmz**
>
> > The proposed method relies heavily on the dataset distribution, particularly requiring $s$ (the number of heavily skewed clusters) to be sufficiently small. In practice, this condition might not always hold for many real-world or synthetic datasets.
>
> Although our algorithm does not outperform standard algorithms with exponential dependency on $k$ when the dataset has a large $s$, we emphasize that due to the APX-hardness of the $(k, z)$-clustering problem, no algorithm can universally outperform standard algorithms with exponential dependency on $k$ in the general case.
>
> This limitation motivates our focus on datasets with heavy skewness, where the underlying structure can be leveraged to achieve polynomial runtime in certain settings. To support this, we have ran experiments to evaluate the skewness of various datasets. We have found many datasets have certain extent of skewness. For instance, more than $30$% datasets are at least $(0.3k, 0.5)$-skewed, which means $30$% of the clusters contributes more than a half of the total cost. Although such skewness is not extremely heavy, our algorithm can still provide certain extent of improvement in runtime compared to the standard algorithm with exponential dependacy on $k$.
>
> Furthermore, there are some datasets with even heavier skewness. We have included an additional section in the appendix demonstrating the $10$% most heavily skewed real-world datasets that we have evaluated. For instance, the dataset of clickstream data for online shopping has $n = 165474$ samples. When $k = 140$, $0.714$% most expensive cluster contributes more than $50$% of the total cost, and $1.429$% most expensive cluster contributes more than $95$% of the total cost. In other words, when $k = 140$, the dataset is $(1, 0.5)$-skewed and $(2, 0.99)$-skewed. Consequently, our algorithm can outperform standard algorithms with exponential dependency on $k$ for such datasets in practice.
>
> > Additionally, the paper does not provide an effective method for calculating or estimating the specific value of $s$ (the number of heavily skewed clusters) for a given dataset, which limits the feasibility of executing the algorithm’s searching process for heavily skewed optimal clustering centers.
>
> Usually, we cannot have an accurate estimation of the specific value of $s$ unless we have the optimal solution. Fortunately, we can still apply our algorithm by using the following possible methods. The first method is to apply some fast algorithm like Lloyd heuristic with k-means++ to evaluate the size of $s$. Although there does not exist accuracy guarantee for such evaluation, it usually performs well in practice. The second method is to fix the value of $s$. Since we define the skewness of a dataset by two parameter $s$ and $\varepsilon$, for instance, a dataset can be both $(1, 0.7)$-skewed and $(3, 0.95)$-skewed, which means we can make trade-off between $s$ and the accuracy $\varepsilon$. Therefore, we can just fix the value of $s$ without evaluating it, and the skewness will just affect the accuracy for the final result.
>
> > In the experimental section, the paper lacks a comparison between the proposed algorithm and standard local search methods, and does not analyze influence of different choices for the number of skewed clusters (i.e., the number of $s$).
>
> > What are the performances of the proposed method compared with the existing local search methods? The authors should add more experimental results to give clearer comparisons with the state-of-the-art local search algorithms.
>
> We have included an additional section in the appendix of the updated version of the manuscript to compare our algorithm with standard local search. We apply the experiments with the same datasets and parameters as the experiments we run for Lloyd's heuristic.
>
> For synthetic data, our experiments illustrate an improvement range for $k$-means from $11.54$% at $k=4$ for the minimum metric to $54.87$% at $k=10$ for the median metric, and for $k$-medoids from $6.06$% at $k=5$ for the minimum metric to $31.86$% at $k=7$ for the average metric. For real data, our experimental results demonstrate an enhancement range for $k$-means from $87.23$% at $k=4$ for the minimum metric up to $95.77$% at $k=10$ for the median metric, and for $k$-medoids from $6.63$% at $k=7$ for the minimum metric to $40.60$% at $k=10$ for the median metric. This overall improvement highlights the superior accuracy performance of our algorithm relative to local search, across various metrics including average, minimum, and median.

---

> ### Author Response · Authors · 2024-11-21
> **Response 2 to Reviewer Aqmz**
>
> > The theoretical results of this paper for Euclidean space heavily relies on the coreset construction methods. With the coreset construction methods, one can also obtain a $(1+\epsilon)$-approximation by executing a weighted PTAS on the coreset constructed without any assumptions on the data distributions (such as the skewness). Can the authors give detailed comparisons between the proposed method in this paper and directly applying a weighted PTAS after the coreset construction?
>
> Due to the APX-hardness of $(k, z)$-clustering, it is impossible to avoid the exponential dependency on $k$ in general setting. The existing PTASs for $(k, z)$-clustering, e.g., the Feldman's PTAS for $k$-means, have a polynomial dependency on $n$ rather than on $k$, while our PTAS has a polynomial dependency on both $n$ and $k$. Unfortunately, due to the APX-hardness, our PTAS has a polynomial dependency on both $n$ and $k$ for dataset with heavy skewness only rather than a general dataset.
>
> > The results of this paper heavily rely on the skewness of the given clustering instance. The proposed method needs to enumerate all subsets with sizes smaller than $s$ to find the optimal clustering centers for the heavily skewed clusters, where the running time can be very large if $s$ is large. Directly using brute-force enumerations may lead to exponential running time. How to handle the case when $s$ is large?
>
> Unfortunately, due to the APX-hardness of $(k,z)$-clustering, it is not possible to always avoid exponential runtime. Since we have an approach to efficiently handle small values of $s$, then we cannot hope to avoid exponential runtime when $s$ is large.
>
> > - page 3, line 161, it should specifically explain the reasoning for $\Delta = polyn$ and whether it relates to the aspect ratio.
>
> We assume that each coordinate is an integer in the range $\{1,\ldots,\Delta\}$, so that each coordinate requires $O(\log\Delta)$ bits of space to represent, which is typically a word of storage. Since weighted points also require $O(\log n)$ bits to represent the weight, we also generally use a word of storage for the weights. Hence, a standard assumption is that $\log\Delta=O(\log n)$. We remark that since each coordinate is an integer in the range $\{1,\ldots,\Delta\}$, then $\Delta$ is related to some notions of aspect ratio, in the sense that it is the maximum distance between points in each dimension.
>
> > - page 4, line 208, the specific definition of sensitivity is provided, but Algorithm 1 does not specify how to calculate the sensitivity for each $x \in X$. A specific or approximate calculation method should be provided.
>
> Thanks for the suggestion. We have described an approach to approximate the sensitivity in Appendix G of the updated version of the manuscript.
>
> > - page 15, line 793, the conclusion “with probability at least 0.99” relies on fixed $\epsilon$ and $\gamma$. However, the theorem does not reflect this property. The theorem should specify approximate values for $\epsilon$ and $\gamma$, or rewrite the probability in a form related to $\epsilon$ and $\gamma$. Similar issues exist for probability descriptions in other theorems throughout the paper.
>
> For the statement of the theorem, our proposition is that there exists some fixed constant $\gamma$ such that for any $\varepsilon$, the successful probability is at least 0.99. We have adjusted the statement in our paper to further clarify this definition.
>
> For instance, for the lemma in page 15, it is now "There exists a constant $\gamma > 0$, such that for any $\varepsilon \in (0, \frac{1}{4}]$, the set $S$ returned by $CoresetConstruction (X, \varepsilon, n, k, \Delta)$ is an $\varepsilon$-coreset of $X$ with probability at least $0.99$ if $\mu = \frac{\gamma dk}{\varepsilon^3} \log (n \Delta)$."
>
> > - page 15, line 796, $(\mathbb{R}^d)^k$ is not defined.
>
> We use the notation $C \in (\mathbb{R}^d)^k$ to express the fact that $C$ is a set of $k$ centers in the space $\mathbb{R}^d$.

---

> > ### Comment · Reviewer_Aqmz · 2024-11-25
> > **Response to the Rebuttal**
> >
> > Thank you for the response. Some of my concerns regarding the theoretical analysis have been addressed. However, I still find the proposed method less practical in its experimental aspects due to the inclusion of enumeration steps. I prefer to keep my initial score at this moment

---

### Official Review · Reviewer_8yqZ · 2024-11-04

**Soundness:** 3
**Presentation:** 3
**Contribution:** 2
**Rating:** 5
**Confidence:** 4

**Summary:**

This paper studies the $(k,z)$-clustering problem for data set with high skewness. The following notation of skewness is used: a dataset is $(s, 1-\epsilon)$-skewed if clustering cost of the $s$ highest cost clusters in the optimal $(k,z)$ clustering is at least $(1 - \epsilon)$ of the optimal total clustering cost. The paper presents a PTAS (running time is $(nk/\epsilon)^{O(s + 1/\epsilon)}$) for $(k,z)$ clustering for $(s, 1-\epsilon)$-skewed data set. The main idea is a partial local search scheme to find the $k$ center points. Here the $s$ centers for the high cost clusters are identified using a brute force search. This can be achieved by spreading $\epsilon$-nets around each point in a coreset of the data set,  inducing a set $T$ of points called candidate center points, and consider $s$-subsets of $T$. The paper presents how such a coreset can be constructed using sensitivity based sampling techniques while preserving the skewness. For the rest $k - s$ centers, an iterative multi-swapping local search process is applied.

**Strengths:**

The paper presents solid mathematical arguments, with the approximation quality and time complexity being theoretically verifiable. The proofs provided for the claims are complex and non-trivial. Additionally, the partial local search approach, which exploits the skewness property, is somewhat intriguing.

**Weaknesses:**

The biggest limitation of the result is the assumption that the dimension $d$ is a constant, while in many practical scenarios where clustering is considered, $d$ is very high. In fact, the size of the constructed candidate center points set $T$ exponentially depends on $d$. The algorithm also tries to perform a brute force search on $T$, which further increase the complexity. In line 90 of the paper, the authors mention that, for high $d$ value a technique in  Makarychev et al. (2019) can be used to eliminate the exponential dependance on dimensionality, and the time complexity can be reduced to polylog. But this is never discussed in later part of the paper. Because of such exponential dependance on $d$, I would not consider the proposed algorithm a real PTAS. The assumption of constant $d$ should also be made clear in the abstract.

The notation of skewness used in this paper is another aspect I believe limits the theoretical contribution of the paper. It depends on the cost of clusters in the optimal clustering, which make the condition rather strong. The running time also depends on $s$ exponentially. This means in some situation the algorithm is not substantially better than a standard algorithm that exponentially depends on $k$.

**Questions:**

Can you provide more details about the claim in line 90 that the running time can be improved for high $d$?

The algorithm depends on $s$ exponentially. In real life data set how large $s$ will typically be?

---

> ### Author Response · Authors · 2024-11-21
> **Response to Reviewer 8yqZ**
>
> > The biggest limitation of the result is the assumption that the dimension $d$ is a constant, while in many practical scenarios where clustering is considered, $d$ is very high. In fact, the size of the constructed candidate center points set $T$ exponentially depends on $d$. The algorithm also tries to perform a brute force search on $T$, which further increase the complexity. In line 90 of the paper, the authors mention that, for high $d$ value a technique in Makarychev et al. (2019) can be used to eliminate the exponential dependance on dimensionality, and the time complexity can be reduced to polylog. But this is never discussed in later part of the paper. Because of such exponential dependance on $d$, I would not consider the proposed algorithm a real PTAS. The assumption of constant $d$ should also be made clear in the abstract.
>
> > Can you provide more details about the claim in line 90 that the running time can be improved for high $d$?
>
> We have added a supplementary section in the appendix to offer more details about how to improve the runtime for high $d$. Informally speaking, we can use Johnson–Lindenstrauss to map the coreset $S$ to $\pi(S) \subset \mathbb{R}^{d'}$, where $d' = ~O(1/\varepsilon^2)$, then use our algorithm to get a $(1+\varepsilon)$-approximation of $\pi(S)$. Let $\{ \pi(A_1), \pi(A_2), \cdots, \pi(A_k) \}$ be the clusters corresponding to such approximate solution. Then by Makarychev et al. (2019), $\{ A_1, A_2, \cdots, A_k \}$ would be clusters corresponding a $(1+O(\varepsilon))$-approximation of $S$. We can find an approximate solution of the center of each $A_i$ in polynomial time since it is a $(1, z)$-clustering, which would be a convex optimation problem. Therefore, we finally find a $(1+O(\varepsilon))$-approximation of $S$ with time polynomial in $n, k, d$.
>
> > The notation of skewness used in this paper is another aspect I believe limits the theoretical contribution of the paper. It depends on the cost of clusters in the optimal clustering, which make the condition rather strong. The running time also depends on $s$ exponentially. This means in some situation the algorithm is not substantially better than a standard algorithm that exponentially depends on $k$.
>
> Although our algorithm does not outperform standard algorithms with exponential dependency on $k$ when the dataset has a large $s$, we emphasize that due to the APX-hardness of the $(k, z)$-clustering problem, no algorithm can universally outperform standard algorithms with exponential dependency on $k$ in the general case.
>
> This limitation motivates our focus on datasets with heavy skewness, where the underlying structure can be leveraged to achieve polynomial runtime in certain settings. To support this, we have ran experiments to evaluate the skewness of various datasets. We have found many datasets have certain extent of skewness. For instance, more than $30$% datasets are at least $(0.3k, 0.5)$-skewed, which means $30$% of the clusters contributes more than a half of the total cost. Although such skewness is not extremely heavy, our algorithm can still provide certain extent of improvement in runtime compared to the standard algorithm with exponential dependacy on $k$.
>
> Furthermore, there are some datasets with even heavier skewness. We have included an additional section in the appendix demonstrating the $10$% most heavily skewed real-world datasets that we have evaluated. For instance, the dataset of clickstream data for online shopping has $n = 165474$ samples. When $k = 140$, $0.714$% most expensive cluster contributes more than $50$% of the total cost, and $1.429$% most expensive cluster contributes more than $95$% of the total cost. In other words, when $k = 140$, the dataset is $(1, 0.5)$-skewed and $(2, 0.99)$-skewed. Consequently, our algorithm can outperform standard algorithms with exponential dependency on $k$ for such datasets in practice.
>
> > The algorithm depends on $s$ exponentially. In real life data set how large $s$ will typically be?
>
> The value of $s$ will depend on the structure of the dataset. There exists some datasets with large $s$ close to $k$, but there also exists some datasets with small $s$. We have included additional discussion in Appendix G of the updated version of the manuscript, including a number of real-world datasets that exhibit datasets with quite heavy skewness.

---

> > ### Comment · Reviewer_8yqZ · 2024-11-25
> >
> > I thank the authors for addressing my concerns. However, I still find the overall contribution of the current result limited. Even with JL lemma, the running time would would exponentially depend on $O(1/\epsilon)$, which still is large for exponential term (especially with the constant factor hidden by big-O). Also, it seems even with a data set that is considered "skewed" enough, the value of s is still around the same magnitude of k, so an exponential dependance on s is still undesirable. Overall, the paper presents some theoretical contributions, but not exciting. I would prefer to keep my current score.

---

### Author Response · Authors · 2024-11-21
**Overall response to all reviewers**

We thank to the reviewers for their constructive remarks and valuable feedback. We especcially appreciate the positive comments on the paper, including:

- The paper presents solid mathematical arguments, with the approximation quality and time complexity being theoretically verifiable. (Reviewer 8yqZ)
- The proofs provided for the claims are complex and non-trivial. (Reviewer 8yqZ)
- Additionally, the partial local search approach, which exploits the skewness property, is somewhat intriguing. (Reviewer 8yqZ)
- Under the notion of skewness, the proposed local search method can break the approximation lower bound of $9+\epsilon$, achieving near-optimal approximation guarantees on clustering quality. (Reviewer Aqmz)
- The skewness of a dataset can hold for several commonly used data distributions, such as Zipfian distribution. (Reviewer Aqmz)
- Experiments show that the proposed method can achieve better performance compared with other clustering algorithms. (Reviewer Aqmz)
- This paper establishes PTAS for the $(k,z)$-clustering problem by leveraging specific data distribution assumptions on the skewness of the data points. (Reviewer xMdu)
- This advancement in approximation is a clear advantage of the proposed methods, marking a significant contribution to the field of clustering and machine learning. (Reviewer xMdu)
- The proposed data reduction and local search techniques are new and interesting, offering potential enhancements to the well-studied local search methods with tight theoretical lower bound of $(9+\epsilon)$. (Reviewer xMdu)
- The proposed data distribution assumption is shown to be practical, as data points generated from a Zipfian distribution naturally exhibit the skewness property. (Reviewer xMdu)
- Interesting model (Reviewer eDaF)
- Strong results (Reviewer eDaF)
- Non-trivial math (Reviewer eDaF)
- Experimental results show that the obtained results are actually good (Reviewer eDaF)
- Well-written (Reviewer eDaF)
- The technical matter itself is well-written and easy to follow. (Reviewer gzCW)
- The algorithms are clearly laid out and the overview of the proof is clear. (Reviewer gzCW)
- I particularly appreciate that the related work is woven into the narrative rather than appearing as a brick wall on one of the pages of the paper. (Reviewer gzCW)
- The theoretical ideas are interesting. (Reviewer gzCW)
- The authors show that one can find the approximate solution in the skewed setting by first finding the optimal solutions for the "heavy" clusters and subsequently adapting a local-search algorithm over the remaining, "lighter" clusters. (Reviewer gzCW)
- It is intuitively expressed why this would not work if done in the naive setting due to (1) an accumulation of errors and (2) a need to adjust the local search for when some centers (those handling the heavy clusters) must stay in place. Although these arguments are subtle, their description is clear and I find them convincing. (Reviewer gzCW)
- The coreset application almost feels like a separate theoretical contribution. The notion that oversampling according to sensitivities is sufficient to approximate the heavy clusters' cost is interesting and I could see applications of this to other problems. (Reviewer gzCW)

We have uploaded an updated version of the manuscript, incorporating reviewer feedback, with the changes marked in blue. Of note, we have:
- Additional experiments demonstrating the skewness of various datasets
- Additional experiments comparing our algorithms with various baselines such as local search, on both synthetic and real-world datasets
- New discussion describing how to efficiently approximate the sensitivities
- Expanded the discussion about how to apply existing dimensionality reducion and convex optimization techniques to remove the exponential dependency on the input dimension $d$ in the runtime

Though we have conducted a substantial number of experiments, we would like to highlight that the primary contribution of our work is theoretical and that the primary function of the empirical evaluations is to serve as straightforward proof-of-concept complementing our theoretical guarantees.

Overall, we believe these changes have improved the presentation of the manuscript and we welcome further feedback. Below, we provide detailed responses to each reviewer’s comments.

---

### Meta-Review · Area_Chair_EYov · 2024-12-17

**Metareview:**

This paper introduces a new notion of "skewness" for clustering problems and presents a polynomial-time approximation scheme (PTAS) for the (k,z)-clustering problem on datasets exhibiting this property.

The reviewers appreciate the introduction of the skewness concept and the theoretical contribution of achieving a PTAS for skewed datasets. They acknowledge the rigor of the analysis and the potential for this work to advance the understanding of clustering algorithms.

However, reviewers also express several concerns:

- Practical Relevance: The main concern is the practical relevance of the proposed skewness assumption. It is unclear how often real-world datasets exhibit this property, and the authors provide limited evidence or discussion on this point.
- Comparison with Existing Work: The paper could benefit from a more thorough comparison with existing work on clustering.

Recommendation:

While the paper presents an interesting theoretical result, the reviewers agree that it does not meet the bar for acceptance at ICLR in its current form.

**Additional Comments On Reviewer Discussion:**

The paper has been significantly improved during the review time but it feels still below the acceptance bar

---

### Decision · Program_Chairs · 2025-01-22

Reject